# EDITABLE CONCEPT BOTTLENECK MODELS

## ABSTRACT

Concept Bottleneck Models (CBMs) have garnered much attention for their ability to elucidate the prediction process through a human-understandable concept layer. However, most previous studies focused on cases where the data, including concepts, are clean. In many scenarios, we always need to remove/insert some training data or new concepts from trained CBMs due to different reasons, such as privacy concerns, data mislabelling, spurious concepts, and concept annotation errors. Thus, the challenge of deriving efficient editable CBMs without retraining from scratch persists, particularly in large-scale applications. To address these challenges, we propose Editable Concept Bottleneck Models (ECBMs). Specifically, ECBMs support three different levels of data removal: concept-label-level, concept-level, and data-level. ECBMs enjoy mathematically rigorous closed-form approximations derived from influence functions that obviate the need for re-training. Experimental results demonstrate the efficiency and effectiveness of our ECBMs, affirming their adaptability within the realm of CBMs.

## 1 INTRODUCTION

Modern deep learning models, such as large language models (Zhao et al., 2023; Yang et al., 2024a;b; Xu et al., 2023; Yang et al., 2024c) and large multimodal (Yin et al., 2023; Ali et al., 2024; Cheng et al., 2024), often exhibit intricate non-linear architectures, posing challenges for end-users seeking to comprehend and trust their decisions. This lack of interpretability presents a significant barrier to adoption, particularly in critical domains such as healthcare (Ahmad et al., 2018; Yu et al., 2018) and finance (Cao, 2022), where transparency is paramount. To address this demand, explainable artificial intelligence (XAI) models (Das & Rad, 2020; Hu et al., 2023b;a) have emerged, offering explanations for their behavior and insights into their internal mechanisms. Among these, Concept Bottleneck Models (CBMs) (Koh et al., 2020) have gained prominence for explaining the prediction process of end-to-end AI models. CBMs add a bottleneck layer for placing human-understandable concepts. In the prediction process, CBMs first predict the concept labels using the original input and then predict the final classification label using the predicted concept in the bottleneck layer, which provides a self-explained decision to users.

Existing research on CBMs predominantly addresses two primary concerns: Firstly, CBMs heavily rely on laborious dataset annotation. Researchers have explored solutions to these challenges in unlabeled settings (Oikarinen et al., 2023; Yuksekgonul et al., 2023; Lai et al., 2023). Secondly, the performance of CBMs often lags behind that of original models lacking the concept bottleneck layer, attributed to incomplete information extraction from original data to bottleneck features. Researchers aim to bridge this utility gap (Sheth & Ebrahimi Kahou, 2023; Yuksekgonul et al., 2023; Espinosa Zarlenga et al., 2022). However, few of them considered the adaptivity or editability of CBMs, crucial aspects encompassing annotation errors, data privacy considerations, or concept updates. Actually, these demands are increasingly pertinent in the era of large models. We delineate the editable setting into three key aspects (illustrated in Figure 1):

- *Concept-label-level:* In most scenarios, concept labels are annotated by humans or experts. Thus, it is unavoidable that there are some annotation errors, indicating that there is a need to correct some concept labels in a trained CBM.

- *Concept-level:* In CBMs, the concept set is pre-defined by LLMs or experts. However, in many cases, evolving situations demand concept updates, as evidenced by discoveries such as chronic obstructive pulmonary disease as a risk factor for lung cancer, and doctors have

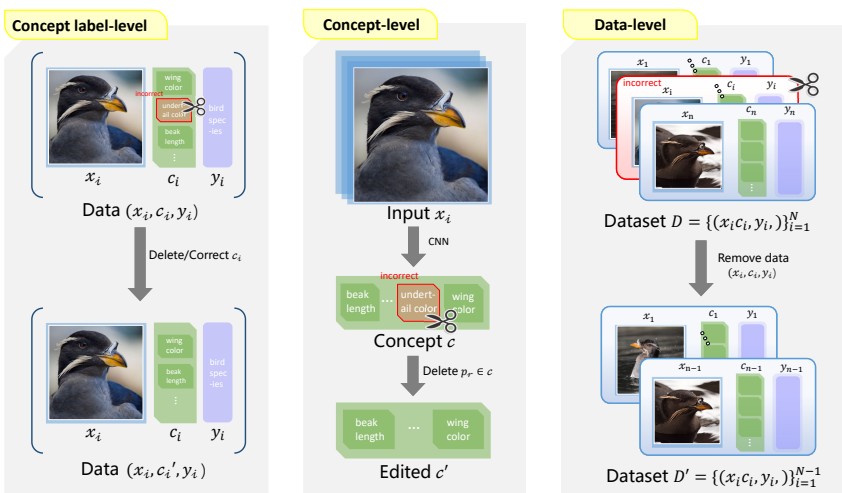

Figure 1: An illustration of Editable Concept Bottleneck Models with three settings.

the requirements to add related concepts. For another example, recent research found a new factor, obesity (Sattar et al., 2020) are risky for severe COVID-19 and factors (e.g., older age, male gender, Asian race) are risk associated with COVID-19 infection (Rozenfeld et al., 2020). On the other hand, one may also want to remove some spurious or unrelated concepts for the task. This demand is even more urgent in some rapidly evolving domains like the pandemic.

- *Data-level:* Data issues can arise in CBMs when training data is erroneous or poisoned. For example, if a doctor identifies a case as erroneous or poisoned, this data sample becomes unsuitable for training. Therefore, it is essential to have the capability to completely delete such data from the learned models. We need such an editable model that can interact effectively with doctors.

The most direct way to address the above three problems is retraining from scratch on the data after correction. However, retraining models in such cases prove prohibitively expensive, especially in large models, which is resource-intensive and time-consuming. Therefore, developing an efficient method to approximate prediction changes becomes paramount. Providing users with an adaptive and editable CBM is both crucial and urgent.

We propose Editable Concept Bottleneck Models (ECBMs) to tackle these challenges. Specifically, compared to retraining, ECBMs provide a mathematically rigorous closed-form approximation for the above three settings to address editability within CBMs efficiently. Leveraging the influence function (Cook, 2000; Cook & Weisberg, 1980), we quantify the impact of individual data points, individual concept labels, and the concept for all data on model parameters. Despite the growing attention and utility of influence functions in machine learning (Koh & Liang, 2017), their application in CBMs remains largely unexplored due to their composite structure, i.e., the intermediate representation layer.

To the best of our knowledge, we are the first to work to fill this gap by demonstrating the effectiveness of influence functions in elucidating the behavior of CBMs, especially in identifying mislabeled data and discerning the data influence. Comprehensive experiments on benchmark datasets show that our ECBMs are efficient and effective. Our contributions are summarized as follows.

- We delineate three different settings that need various levels of data or concept removal in CBMs: concept-label-level, concept-level, and data-level. To the best of our knowledge, our research marks the first exploration of data removal issues within CBMs.

- To make CBMs able to remove data or concept influence without retraining, we propose the Editable Concept Bottleneck Models (ECBMs). Our approach in ECBMs offers a mathematically rigorous closed-form approximation. Furthermore, to improve computational

efficiency, we present streamlined versions integrating Eigenvalue-corrected Kronecker-Factored Approximate Curvature (EK-FAC).

- To showcase the effectiveness and efficiency of our ECBMs, we conduct comprehensive experiments across various benchmark datasets to demonstrate our superior performance.

## 2 RELATED WORK

**Concept Bottleneck Models.** CBM (Koh et al., 2020) stands out as an innovative deep-learning approach for image classification and visual reasoning. It introduces a concept bottleneck layer into deep neural networks, enhancing model generalization and interpretability by learning specific concepts. However, CBM faces two primary challenges: its performance often lags behind that of original models lacking the concept bottleneck layer, attributed to incomplete information extraction from the original data to bottleneck features. Additionally, CBM relies on laborious dataset annotation. Researchers have explored solutions to these challenges. Chauhan et al. (2023) extend CBM into interactive prediction settings, introducing an interaction policy to determine which concepts to label, thereby improving final predictions. Oikarinen et al. (2023) address CBM limitations and propose a novel framework called Label-free CBM. This innovative approach enables the transformation of any neural network into an interpretable CBM without requiring labeled concept data, all while maintaining high accuracy. Post-hoc Concept Bottleneck models (Yuksekgonul et al., 2023) can be applied to various neural networks without compromising model performance, preserving interpretability advantages. CBMs work on the image field also includes the works of Havasi et al. (2022),Kim et al. (2023),Keser et al. (2023),Sawada & Nakamura (2022) and Sheth & Kahou (2023). Despite many works on CBMs, we are the first to investigate the interactive influence between concepts through influence functions. Our research endeavors to bridge this gap by utilizing influence functions in CBMs, thereby deciphering the interaction of concept models and providing an adaptive solution to concept editing. For more related work, please refer to Appendix I.

## 3 PRELIMINARIES

**Concept Bottleneck Models.** In this paper, we consider the original CBM, and we adopt the notations used by Koh et al. (2020). We consider a classification task with a concept set denoted as $c = \{p_1, \cdots, p_k\}$ with each $p_i$ is a concept given by experts or LLMs, and a training dataset represented as $\mathcal{D} = \{z_i\}_{i=1}^n$, where $z_i = (x_i, y_i, c_i)$. Here, for $i \in [n]$, $x_i \in \mathbb{R}^m$ represents the feature vector, $y_i \in \mathbb{R}^{d_z}$ denotes the label (with $d_z$ corresponding to the number of classes), and $c_i = (c_i^1, \cdots, c_i^k) \in \mathbb{R}^k$ represents the concept vector. In this context, $c_i^j$ represents the weight of the concept $p_j$. In CBMs, our goal is to learn two representations: one called concept predictor that transforms the input space to the concept space, denoted as $g : \mathbb{R}^m \to \mathbb{R}^k$, and another called label predictor that maps the concept space to the prediction space, denoted as $f : \mathbb{R}^k \to \mathbb{R}^{d_z}$. Usually, here the map $f$ is linear. For each training sample $z_i = (x_i, y_i, c_i)$, we consider two empirical loss functions: one is from the input space to concept space, and the other is from concept space to output space:

$$\hat{g} = \arg\min_g \sum_{j=1}^k \sum_{i=1}^n L_C(g^j(x_i), c_i^j) = \arg\min_g \sum_{j=1}^k \sum_{i=1}^n L_{C_j}(g(x_i), c_i), \quad (1)$$

$$\hat{f} = \arg\min_f \sum_{i=1}^n L_Y(f(\hat{g}(x_i)), y_i) = \arg\min_f \sum_{i=1}^n L_{Y_i}(f, \hat{g}), \quad (2)$$

where $g^j(x_i)$ is the $j$-th concept predictor with $x_i$, $L_C$ and $L_Y$ are loss functionsSee the notation table in Appendix 2.. We treat $g$ as a collection of $k$ concept predictors and separate different columns as a vector $g^j(x_i)$ for simplicity. Furthermore, in this paper, we primarily focus on the scenarios where the label predictor $f$ is a linear transformation.

For any input $x$, we aim to ensure that its predicted concept vector $\hat{c} = g(x)$ and prediction $\hat{y} = f(\hat{g}(x))$ are close to their underlying counterparts, thus capturing the essence of the original CBMs.

**Influence function.** The influence function is a measure of the dependence of the estimator on the value of any one of the points in the sample. Consider a neural network $\hat{\theta} = \arg\min_\theta \sum_{i=1}^n \ell(z_i; \theta)$ with loss function $\ell$ and dataset $D = \{z_i\}_{i=1}^n$. If we remove a point $z_m$ from the training dataset, the parameters become $\hat{\theta}_{-z_m} = \arg\min_\theta \sum_{i \neq m} \ell(z_i; \theta)$. Influence function gives us an efficient approximation for $\hat{\theta}_{-z_m}$ by defining a response function as $\hat{\theta}_{\epsilon, -z_m} = \underset{\theta \in \Theta}{\arg\min} \frac{1}{n} \sum_{i=1}^n L(z_i; \theta) + \epsilon L(z_m; \theta)$.

Perform first-order Taylor expansion on the gradient of the objective function corresponding to the $\arg\min$ process of the response function, we can obtain the influence function defined by

$$\mathcal{I}_{\hat{\theta}}(z_m) \triangleq \frac{\mathrm{d}\hat{\theta}_{\epsilon, -z_m}}{\mathrm{d}\epsilon}\bigg|_{\epsilon=0} = -H_{\hat{\theta}}^{-1} \cdot \nabla_\theta \ell(z_m; \hat{\theta}),$$

which can evaluate the influence of $z_m$ on the parameters. When the loss function $\ell$ is twice-differentiable and strongly convex in $\theta$, the Hessian $H_{\hat{\theta}}$ is positive definite and thus the influence function is well-defined. For non-convex loss, Bartlett (1953) proposed that the Hessian $H_{\hat{\theta}}$ can be replaced by $\hat{H} = G_{\hat{\theta}} + \lambda I$ where $G_{\hat{\theta}}$ is the Fisher information matrix defined by $n^{-1} \sum_{i=1}^n \nabla_\theta \ell(z_i; \theta) \nabla_\theta \ell(z_i; \theta)^{\mathrm{T}}$, $\lambda$ is a small damping term used to ensure the positive definiteness of $\hat{H}$. We can employ the Eigenvalue-corrected Kronecker-Factored Approximate Curvature (EK-FAC) method to further accelerate the computation. See Appendix C for details.

## 4 EDITABLE CONCEPT BOTTLENECK MODELS

In this section, we introduce our EBCMs for the three settings mentioned in the introduction by leveraging the influence function. Specifically, for the concept-label level, we will calculate the influence of a set of data sample's different concept labels; for the concept level, we will calculate the influence of several concepts; for the data level, we will calculate the influence of several samples.

### 4.1 CONCEPT LABEL-LEVEL EDITABLE CBM

In many cases, several data samples possess erroneous annotations for certain concepts, yet we may opt to preserve their other information, particularly considering the high cost associated with acquiring data in specific domains like medical imaging. In such scenarios, it is common practice to correct such erroneous concepts instead of removing the whole data point from the dataset. Estimating the changes in the parameters of the retraining model holds significance in this context. We name this case as concept label-level editable CBM.

Mathematically, we have a set of erroneous data $D_e$ and its associated index set $S_e \subseteq [n] \times [k]$ such that for each $(w, r) \in S_e$, we have $(x_w, y_w, c_w) \in D_e$ with $c_w^r$ is mislabeled and $\tilde{c}_w^r$ is its corrected concept label. Thus, our goal is to approximate the new CBM without retraining. The retrained concept predictor and label predictor will be represented in the following manner.

$$\hat{g}_e = \underset{g}{\arg\min} \left[ \sum_{(i,j) \notin S_e} L_C\left(g^j(x_i), c_i^j\right) + \sum_{(i,j) \in S_e} L_C\left(g^j(x_i), \tilde{c}_i^j\right) \right], \quad (3)$$

$$\hat{f}_e = \underset{f}{\arg\min} \sum_{i=1}^n L_Y\left(f\left(\hat{g}_e(x_i)\right), y_i\right). \quad (4)$$

For simple neural networks, we can use the influence function approach directly to estimate the retrained model. However, for CBM architecture, if we intervene with the true concepts, the concept predictor $\hat{g}$ fluctuates to $\hat{g}_e$ accordingly. Observing that the input data of the label predictor is the output of the concept predictor, which is also changing. Therefore, we need to adopt a two-stage editing approach. Here we consider the influence function for equation 3 and equation 4 separately. We first edit the concept predictor from $\hat{g}$ to $\bar{g}_e$, and then edit from $\hat{f}$ to $\bar{f}_e$ based on our approximated concept predictor, by the following two theorems.

**Theorem 4.1.** *The retrained concept predictor $\hat{g}_e$ defined by (3) can be approximated by:*

$$\hat{g}_e \approx \bar{g}_e \triangleq \hat{g} - H_{\hat{g}}^{-1} \cdot \sum_{(w,r) \in S_e} \left(\nabla_{\hat{g}} L_C\left(\hat{g}^r(x_w), \tilde{c}_w^r\right) - \nabla_{\hat{g}} L_C\left(\hat{g}^r(x_w), c_w^r\right)\right), \quad (5)$$

*where $H_{\hat{g}} = \nabla_{\hat{g}}^2 \sum_{i,j} L_C(\hat{g}^j(x_i), c_i^j)$ is the Hessian matrix of the loss function with respect to $\hat{g}$.*

**Theorem 4.2.** *The retrained label predictor $\hat{f}_e$ defined by equation 4 can be approximated by:*

$$\hat{f}_e \approx \bar{f}_e = \hat{f} + H_{\hat{f}}^{-1} \cdot \nabla_f \sum_{i=1}^{n} L_{Y_i} \left( \hat{f}, \hat{g} \right) - H_{\hat{f}}^{-1} \cdot \nabla_f \sum_{i=1}^{n} L_{Y_i} \left( \hat{f}, \bar{g}_e \right),$$

*where $H_{\hat{f}} = \nabla_{\hat{f}}^2 \sum_{i=1}^{n} L_{Y_i}(\hat{f}, \hat{g})$ is the Hessian matrix of the loss function with respect to $\hat{f}$, $L_{Y_i}(\hat{f}, \hat{g}) \triangleq L_Y(\hat{f}(\hat{g}(x_i)), y_i)$, and $\bar{g}_e$ is given in Theorem 4.1.*

**Difference with test-time intervention.** The ability to intervene in CBMs enables human users to interact with the model in the prediction process, for example, a medical expert can substitute the erroneous predicted concept value $\hat{c}$ directly, and then observe its effect on the final prediction $\hat{y}$. However, the fundamental flaws in the concept predictor have not been thoroughly rectified, and similar errors may persist when applied to new test data. While under the editable CBM framework, not only can test-time intervention be performed, but the concept predictor of the CBM can also undergo secondary editing based on the test data that repeatedly yields errors. This process extends the rectification from the data level to the model level.

## 4.2 Concept-level Editable CBM

In this case, a set of concepts is removed due to incorrect attribution or spurious concepts, termed concept-level edit. [1]Specifically, for the concept set, denote the erroneous concept index set as $M \subset [k]$, we aim to delete these concept labels in all training samples. We aim to investigate the impact of updating the concept set within the training data on the model's predictions. It is notable that compared to the above concept label case, the dimension of output (input) of the retrained concept predictor (label predictor) will change. If we delete $t$ concepts from the dataset, then $g$ becomes $g' : \mathbb{R}^m \to \mathbb{R}^{k-t}$ and $f$ becomes $f' : \mathbb{R}^{k-t} \to \mathbb{R}^{d_z}$. More specifically, if we retrain the CBM with the revised dataset, the corresponding concept predictor becomes:

$$\hat{g}_{-p_M} = \arg\min_{g'} \sum_{j \notin M} L_{C_j} = \arg\min_{g'} \sum_{j \notin M} \sum_{i=1}^{n} L_C(g'^j(x_i), c_i^j). \tag{6}$$

The variation of the parameters in dimension renders the application of influence function-based editing challenging for the concept predictor. This is because the influence function implements the editorial predictor by approximate parameter change from the original base after $\epsilon$-weighting the corresponding loss for a given sample, and thus, it is unable to deal with changes in parameter dimensions.

To overcome the challenge, our strategy is to develop some transformations that need to be performed on $\hat{g}_{-p_M}$ to align its dimension with $\hat{g}$ so that we can apply the influence function to edit the CBM. We achieve this by mapping $\hat{g}_{-p_M}$ to $\hat{g}_{-p_M}^* \triangleq \mathrm{P}(\hat{g}_{-p_M})$, which has the same amount of parameters as $\hat{g}$ and has the same predicted concepts $\hat{g}_{-p_M}^*(j)$ as $\hat{g}_{-p_M}(j)$ for all $j \in [k] - M$. We achieve this effect by inserting a zero row vector into the $r$-th row of the matrix in the final layer of $\hat{g}_{-p_M}$ for $r \in M$. Thus, we can see that the mapping $P$ is one-to-one. Moreover, assume the parameter space of $\hat{g}$ is $T$ and that of $\hat{g}_{-p_M}^*$, $T_0$ is the subset of $T$. Noting that $\hat{g}_{-p_M}^*$ is the optimal model of the following objective function:

$$\hat{g}_{-p_M}^* = \arg\min_{g' \in T_0} \sum_{j \notin M}^{k} \sum_{i=1}^{n} L_{C_j}(g'^j(x_i), c_i^j), \tag{7}$$

i.e., it is the optimal model of the concept predictor loss on the remaining concepts under the constraint $T_0$. Now we can apply the influence function to edit $\hat{g}$ to approximate $\hat{g}_{-p_M}^*$ with the restriction on the value of 0 for rows indexed by MM with the last layer of the neural network, denoted as $\bar{g}_{-p_M}^*$. After that, we remove from $\bar{g}_{-p_M}^*$ the parameters initially inserted to fill in the dimensional difference, which always equals 0 because of the restriction we applied in the editing stage, thus approximating the true edited concept predictor $\hat{g}_{-p_M}$. We now detail the editing process from $\hat{g}$ to $\hat{g}_{-p_M}^*$ using the following theorem.

---

[1]For convenience, in this paper, we only consider concept removal; our method can directly extend to concept insertion.

**Theorem 4.3.** *For the retrained concept predictor $\hat{g}_{-p_M}$ defined in equation 6, we map it to $\hat{g}^*_{-p_M}$ as equation 7. And we can edit the initial $\hat{g}$ to $\hat{g}^*_{-p_M}$ as:*

$$\hat{g}^*_{-p_M} \approx \bar{g}^*_{-p_M} \triangleq \hat{g} - H_{\hat{g}}^{-1} \cdot \nabla_{\hat{g}} \sum_{j \notin M} \sum_{i=1}^n L_{C_j}(\hat{g}(x_i), c_i),$$

*where $H_{\hat{g}} = \nabla_g^2 \sum_{j \notin M} \sum_{i=1}^n L_{C_j}(\hat{g}(x_i), c_i)$. Then, by removing all zero rows inserted during the mapping phase, we can naturally approximate $\hat{g}_{-p_M} \approx \mathrm{P}^{-1}(\hat{g}^*_{-p_M})$.*

For the second stage of training, assume we aim to remove concept $p_r$ for $r \in M$ and the new optimal model is $\hat{f}_{-p_M}$. We will encounter the same difficulty as in the first stage, i.e., the number of parameters of the label predictor will change. To address the issue, our key observation is that in the existing literature on CBMs, we always use linear transformation for the label predictor, meaning that the dimensions of the input with values of $0$ will have no contribution to the final prediction. To leverage this property, we fill the missing values in the input of the updated predictor with $0$, that is, replacing $\hat{g}_{-p_M}$ with $\hat{g}^*_{-p_M}$ and consider $\hat{f}_{p_M=0}$ defined by

$$\hat{f}_{p_M=0} = \arg \min_f \sum_{i=1}^n L_{Y_i}\left(f, \hat{g}^*_{-p_M}\right). \tag{8}$$

In total, we have the following lemma:

**Lemma 4.4.** *In the CBM, if the label predictor utilizes linear transformations of the form $\hat{f} \cdot c$ with input $c$, then, for each $r \in M$, we remove the $r$-th concept from $c$ and denote the new input as $c'$; set the $r$-th concept to $0$ and denote the new input as $c^0$. Then we have $\hat{f}_{-p_M} \cdot c' = \hat{f}_{p_M=0} \cdot c^0$ for any input $c$.*

Lemma 4.4 demonstrates that the retrained $\hat{f}_{-p_M}$ and $\hat{f}_{p_M=0}$, when given inputs $\hat{g}_{-p_M}(x)$ and $\hat{g}^*_{-p_M}(x)$ respectively, yield identical outputs. Consequently, we can utilize $\hat{f}_{p_M=0}$ as the editing target in place of $\hat{f}_{-p_M}$.

**Theorem 4.5.** *For the revised retrained label predictor $\hat{f}_{p_M=0}$ defined by equation 8, we can edit the initial label predictor $\hat{f}$ to $\bar{f}_{p_M=0}$ by the following equation as a substitute for $\hat{f}_{p_M=0}$:*

$$\hat{f}_{p_M=0} \approx \bar{f}_{p_M=0} \triangleq \hat{f} - H_{\hat{f}}^{-1} \cdot \nabla_{\hat{f}} \sum_{l=1}^n L_{Y_l}\left(\hat{f}, \bar{g}^*_{-p_M}\right),$$

*where $H_{\hat{f}} = \nabla_{\hat{f}}^2 \sum_{i=1}^n L_{Y_i}(\hat{f}, \hat{g})$. Deleting the $r$-th dimension of $\bar{f}_{p_M=0}$ for $r \in M$, then we can map it to $\bar{f}_{-p_M}$, which is the approximation of the final edited label predictor $\hat{f}_{-p_M}$ under concept level.*

### 4.3 DATA-LEVEL EDITABLE CBM

In this scenario, we are more concerned about fully removing the influence of data samples on CBMs due to different reasons, such as the training data involving poisoned or erroneous issues. Specifically, we have a set of samples to be removed $\{(x_i, y_i, c_i)\}_{i \in G}$ with $G \subset [n]$. Then, we define the retrained concept predictor as

$$\hat{g}_{-z_G} = \arg \min_g \sum_{j=1}^k \sum_{i \in [n]-G} L_{C_j}(g(x_i), c_i) \tag{9}$$

which can be evaluated by the following theorem:

**Theorem 4.6.** *For dataset $\mathcal{D} = \{(x_i, y_i, c_i)\}_{i=1}^n$, given a set of data $z_r = (x_r, y_r, c_r)$, $r \in G$ to be removed. Suppose the updated concept predictor $\hat{g}_{-z_G}$ is defined by equation 9, then we have the following approximation for $\hat{g}_{-z_G}$*

$$\hat{g}_{-z_G} \approx \bar{g}_{-z_G} \triangleq \hat{g} + H_{\hat{g}}^{-1} \cdot \sum_{r \in G} \nabla_g L_{C_r}(\hat{g}(x_r), c_r), \tag{10}$$

*where $H_{\hat{g}} = \nabla_{\hat{g}}^2 \sum_{i,j} L_{C_j}(\hat{g}^j(x_i), c_i^j)$ is the Hessian matrix of the loss function with respect to $\hat{g}$.*

Based on $\hat{g}_{-z_G}$, the label predictor becomes $\hat{f}_{-z_G}$ which is defined by

$$\hat{f}_{-z_G} = \arg\min_f \sum_{i \in [n]-G} L_{Y_i}(f, \hat{g}_{-z_G}). \tag{11}$$

Compared with the original loss before unlearning in equation 2, we can observe two changes in equation 11. First, we remove $|G|$ data points in the loss function $L_Y$. Secondly, the input for the loss is also changed from $\hat{g}(x_i)$ to $\hat{g}_{-z_G}$. Therefore, it is difficult to estimate directly with an influence function. Here we introduce an intermediate label predictor as

$$\tilde{f}_{-z_G} = \arg\min \sum_{i \in [n]-G} L_{Y_i}(f, \hat{g}), \tag{12}$$

and split the estimate of $\hat{f}_{-z_G} - \hat{f}$ into $\hat{f}_{-z_G} - \tilde{f}_{-z_G}$ and $\tilde{f}_{-z_G} - \hat{f}$.

**Theorem 4.7.** *For dataset $\mathcal{D} = \{(x_i, y_i, c_i)\}_{i=1}^n$, given a set of data $z_r = (x_r, y_r, c_r)$, $r \in G$ to be removed. The intermediate label predictor $\tilde{f}_{-z_G}$ is defined in equation 12. Then we have*

$$\tilde{f}_{-z_G} - \hat{f} \approx H_{\hat{f}}^{-1} \cdot \sum_{i \in [n]-G} \nabla_{\hat{f}} L_{Y_i}(\hat{f}, \hat{g}) \triangleq A_G.$$

*We denote the edited version of $\tilde{f}_{-z_G}$ as $\bar{f}_{-z_G}^* \triangleq \hat{f} + A_G$. Define $B_G$ as*

$$\hat{f}_{-z_G} - \tilde{f}_{-z_G} \approx -H_{\bar{f}_{-z_G}^*}^{-1} \cdot \nabla_{\hat{f}} \sum_{i \in [n]-G} \left( L_{Y_i}\left(\bar{f}_{-z_G}^*, \bar{g}_{-z_G}\right) - L_{Y_i}\left(\bar{f}_{-z_G}^*, \hat{g}\right) \right) \triangleq B_G,$$

*where $H_{\bar{f}_{-z_G}^*} = \nabla_{\bar{f}} \sum_{i \in [n]-G} L_{Y_i}\left(\bar{f}_{-z_G}^*, \hat{g}\right)$ is the Hessian matrix concerning $\bar{f}_{-z_G}^*$. Combining the above two-stage approximation, then, the final edited label predictor $\bar{f}_{-z_G}$ can be obtained by*

$$\bar{f}_{-z_G} = \bar{f}_{-z_G}^* + B_G = \hat{f} + A_G + B_G. \tag{13}$$

**Acceleration via EK-FAC.** As we mentioned in Section 3, as the loss function in CBMs is non-convex, the Hessian matrices in all our theorems may not be well-defined. We can use the EK-FAC approach, i.e., using $\hat{H}_\theta = G_\theta + \lambda I$ to approximate the Hessian, where $G_\theta$ is the Fisher information matrix of model $\theta$, and $\lambda$ is a small damping term used to ensure the positive definiteness. See Appendix C.1 for using EK-FAC to CBMs. Also, see Algorithm 6-8 in the Appendix for the detailed EK-FAC-based algorithms for our three levels, whose original (Hessian) versions are in Algorithm 1-3, respectively.

## 5 EXPERIMENTS

In this section, we demonstrate our main experimental results on utility evaluation, edition efficiency, and interpretability evaluation. Details and additional results are in Appendix H due to space limit.

### 5.1 EXPERIMENTAL SETTINGS

**Dataset.** We utilize three datasets: *X-ray grading (OAI)* (Nevitt et al., 2006), *Bird identification (CUB)* (Wah et al., 2011) and *Large-scale CelebFaces Attributes dataset (CelebA)* (Liu et al., 2015). OAI is a multi-center observational study of knee osteoarthritis, which comprises 36,369 data points. Specifically, we configure n=10 concepts that characterize crucial osteoarthritis indicators such as joint space narrowing, osteophytes, and calcification. Bird identification (CUB)[2] consists of 11,788 data points, which belong to 200 classes and include 112 binary attributes to describe detailed visual features of birds. CelebA comprises 202,599 celebrity images, each annotated with 40 binary attributes that detail facial features, such as hair color, eyeglasses, and smiling. As the dataset lacks predefined classification tasks, following Espinosa Zarlenga et al. (2022), we designate 8 attributes as labels and the remaining 32 attributes as concepts. For all the above datasets, we follow the same network architecture and settings outlined in Koh et al. (2020).

---

[2]The original dataset is processed. Detailed explanation can be found in H.

**Ground Truth and Baselines.** We use retrain as the ground truth method. *Retrain*: We retrain the CBM from scratch by removing the samples, concept labels, or concepts from the training set. We employ two baseline methods: CBM-IF, and ECBM. *CBM-IF*: This method is a direct implementation of our previous theorems of model updates in the three settings. See Algorithms 1-3 in Appendix for details. *ECBM*: As we discussed above, all of our model updates can be further accelerated via EK-FAC, ECBM corresponds to the EK-FAC accelerated version of Algorithms 1-3 (refer to Algorithms 6-8 in Appendix).

**Evaluation Metric.** We utilize two primary evaluation metrics to assess our models: the F1 score and runtime (RT). The *F1 score* measures the model's performance by balancing precision and recall. *Runtime*, measured in seconds, evaluates the running time of each method to update the model.

**Implementation Details.** Our experiments utilized an Intel Xeon CPU and an RTX 3090 GPU. For utility evaluation, at the concept level, one concept was randomly removed for the OAI dataset and repeated while ten concepts were randomly removed for the CUB dataset, with five different seeds. At the data level, 3% of the data points were randomly deleted and repeated 10 times with different seeds. At the concept-label level, we randomly selected 3% of the data points and modified one concept of each data randomly, repeating this 10 times for consistency across iterations.

## 5.2 Evaluation of Utility and Editing Efficiency

Our experimental results, as illustrated in Table 1, demonstrate the effectiveness of ECBMs compared to traditional retraining and CBM-IF, particularly emphasizing computational efficiency without compromising accuracy. Specifically, ECBMs achieved F1 scores close to those of retraining (0.8808 vs. 0.8825) while significantly reducing the runtime from 31.44 seconds to 8.29 seconds. This pattern is consistent in the CUB dataset, where the runtime was decreased from 27.88 seconds for retraining to 7.03 seconds for ECBMs, with a negligible difference in the F1 score (0.7971 to 0.7963). These results highlight the potential of ECBMs to provide substantial time savings—approximately 22-30% of the computational time required for retraining—while maintaining comparable accuracy. Compared to CBM-IF, ECBM also showed a slight reduction in runtime and a significant improvement in F1 score. The former verifies the effective acceleration of our algorithm by EK-FAC. This efficiency is particularly crucial in scenarios where frequent updates to model annotations are needed, confirming the utility of ECBMs in dynamic environments where running time and accuracy are critical.

We can also see that the original version of ECBM, i.e., CBM-IF, also has a lower runtime than retraining but a lower F1 score than ECBM. Such results may be due to different reasons. For example, our original theorems depend on the inverse of the Hessian matrices, which may not be well-defined for non-convex loss. Moreover, these Hessian matrices may be ill-conditioned or singular, which makes calculating their inverse imprecise and unstable.

Table 1: Performance comparison of different methods on the three datasets.

| Edit Level | Method | OAI | | CUB | | CelebA | |
|---|---|---|---|---|---|---|---|
| | | F1 score | RT (second) | F1 score | RT (second) | F1 score | RT (second) |
| Concept Label | Retrain | 0.8825±0.0054 | 31.44 | 0.7971±0.0066 | 27.88 | 0.3827±0.0272 | 57.60 |
| | CBM-IF(Ours) | 0.8639±0.0033 | 16.31 | 0.7699±0.0035 | 14.39 | 0.3561±0.0134 | 34.93 |
| | ECBM(Ours) | **0.8808±0.0039** | **8.29** | **0.7963±0.0050** | **7.03** | **0.3845±0.0327** | **15.67** |
| Concept | Retrain | 0.8448±0.0191 | 27.33 | 0.7811±0.0047 | 28.41 | 0.3776±0.0350 | 68.94 |
| | CBM-IF(Ours) | 0.8214±0.0071 | 17.38 | 0.7579±0.0065 | 15.70 | 0.3609±0.0202 | 35.56 |
| | ECBM(Ours) | **0.8403±0.0090** | **8.30** | **0.7787±0.0058** | **6.43** | **0.3761±0.0280** | **15.99** |
| Data | Retrain | 0.8811±0.0065 | 33.72 | 0.7838±0.0051 | 28.08 | 0.3797±0.0375 | 65.60 |
| | CBM-IF(Ours) | 0.8472±0.0046 | 17.84 | 0.7623±0.0031 | 15.86 | 0.3536±0.0166 | 40.08 |
| | ECBM(Ours) | **0.8797±0.0038** | **8.81** | **0.7827±0.0088** | **7.11** | **0.3748±0.0347** | **16.75** |

**Editing Multiple Samples.** To comprehensively evaluate the editing capabilities of ECBM in various scenarios, we conducted experiments on the performance with multiple samples that need to be removed. Specifically, for the concept label/data levels, we consider the different ratios of samples (1-10%) for edit, while for the concept level, we consider removing different numbers of concepts $\in \{2, 4, 6, \cdots, 20\}$. We compared the performance of retraining, CBM-IF, and ECBM methods. As shown in Figure 2, except for certain cases at the concept level, the F1 score of the ECBM method is generally around 0.0025 lower than that of the retrain method, which is significantly better than the corresponding results of the CBM-IF method. Recalling Table 1, the speed of ECBM is more than

three times faster than that of retraining. Consequently, ECBM is an editing method that achieves a trade-off between speed and effectiveness.

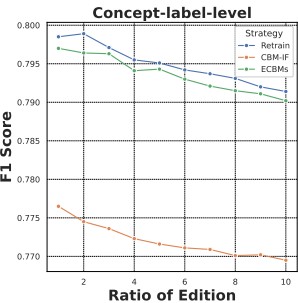 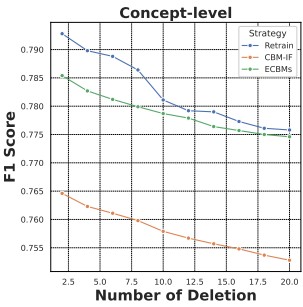 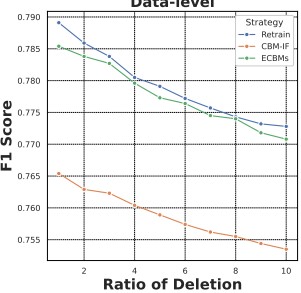

Figure 2: Impact of edition ratio on three settings on CUB dataset.

## 5.3 RESULTS ON INTERPRETABILITY

**Influence function in ECBM can measure the importance of concepts.** The original motivation of the influence function is to calculate the importance score of each sample. Here, we will show that the influence function for the concept level in Theorem 4.3 can be used to calculate the importance of each concept in CBMs, which provides an explainable tool for CBMs. In detail, we conduct our experiments on the CUB dataset. We first select 1-10 most influential and 1-10 least influential concepts by our influence function. Then, we will remove these concepts and update the model via retraining or our ECBM and analyze the change (F1 Score Difference) w.r.t. the original CBM before removal.

The results in Figure 3a demonstrate that when we remove the 1-10 most influential concepts identified by the ECBM method, the F1 score decreases by more than 0.025 compared to the CBM before removal. In contrast, Figure 3b shows that the change in the F1 score remains consistently below 0.005 when removing the least influential concepts. These findings strongly indicate that the influence function in ECBM can successfully determine the importance of concepts. Furthermore, we observe that the gap between the F1 score of retraining and ECBM is consistently smaller than 0.005, and even smaller in the case of least important concepts. This further suggests that when ECBM edits various concepts, its performance is very close to the ground truth.

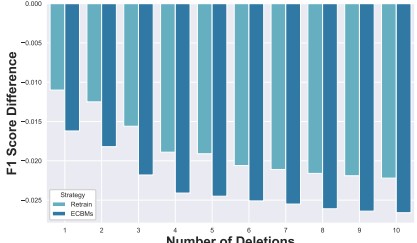 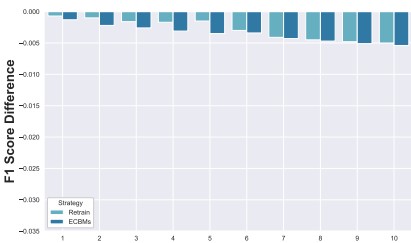

(a) Results on the 1-10 most influential concepts      (b) Results on the 1-10 least influential concepts

Figure 3: F1 score difference after removing most and least influential concepts given by our concept level influence function.

**ECBMs can erase data influence.** For the data level, ECBMs aim to facilitate an efficient removal of samples. We perform membership inference attacks (MIAs) to provide direct evidence that ECBMs can indeed erase data influence. MIA is a privacy attack that aims to infer whether a specific data sample was part of the training dataset used to train a model. The attacker exploits the model's behavior, such as overconfidence or overfitting, to distinguish between *training (member)* and *non-training (non-member)* data points. In MIAs, the attacker typically queries the model with a data sample and observes its prediction confidence or loss values, which tend to be higher for members of the training set than non-members (Shokri et al., 2017).

To quantify the success of these edits, we calculate the RMIA (Removed Membership Inference Attack) score for each category. The RMIA score is defined as the model's confidence in classifying

whether a given sample belongs to the training set. Lower RMIA values indicate that the sample behaves more like a test set (non-member) sample Zarifzadeh et al. (2024). This metric is especially crucial for edited samples, as a successful ECBM should make the removed members behave similarly to non-members, reducing their membership vulnerability. See Appendix H for its definition.

We conducted experiments by randomly selecting 200 samples from the training set (members) and 200 samples from the test set (non-members) of the CUB dataset. We calculated the RMIA scores for these samples and plotted their frequency distributions, as shown in Figure 4a. The mean RMIA score for non-members was 0.049465, while members had a mean score of 0.063505. Subsequently, we applied ECBMs to remove the 200 training samples from the model, updated the model parameters, and then recalculated the RMIA scores. After editing, the mean RMIA score for the removed-members decreased to 0.052105, significantly closer to the non-members' mean score. This shift in RMIA values demonstrates the effectiveness of ECBMs in editing the model, as the removed members now exhibit behavior closer to that of non-members. The post-editing RMIA score distributions are shown in Figure 4b. These results provide evidence of the effectiveness of ECBMs in editing the model's knowledge about specific samples.

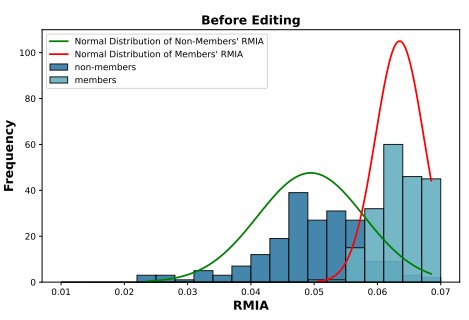
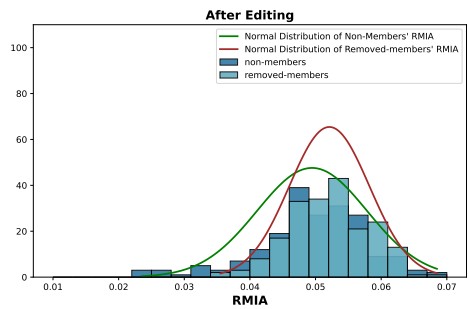

(a) RMIA Score Before Editing

(b) RMIA Score After Editing

Figure 4: RMIA scores of data before and after removal.

**Visualization.** Since CBM is an explainable model, we aim to evaluate the interpretability of our ECBM (compared to the retraining). We will present some visualization results for the concept-level edit. Figure 5 presents the top 10 most influential concepts and their corresponding predicted concept labels obtained by our ECBM and the retrain method after randomly deleting concepts for the CUB dataset. Detailed explanation can be found in Appendix H.4.1. Our ECBM can provide explanations for which concepts are crucial and how they assist the prediction. Specifically, among the top 10 most important concepts in the ground truth (retraining), ECBM can accurately recognize 9 within them. For instance, we correctly identify "has_upperparts_color::orange", "has_upper_tail_color::red", and "has_breast_color::black" as some of the most important concepts when predicting categories. Additional visualization results under data level and concept-label level on OAI and CUB datasets are included in Appendix H.4.2.

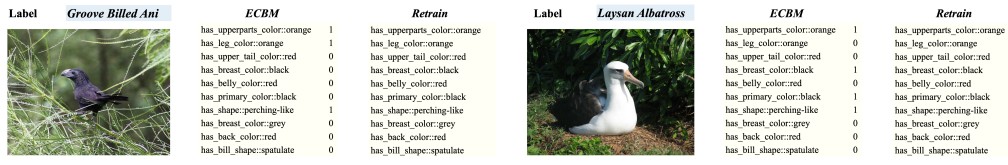

Figure 5: Visualization of the Top 10 Most Influential Concepts for CBM(Identified by ECBM or Retrain) Highlighted on an Extracted Image.

## 6 CONCLUSION

In this paper, we propose Editable Concept Bottleneck Models (ECBMs). ECBMs can address issues of removing/inserting some training data or new concepts from trained CBMs for different reasons, such as privacy concerns, data mislabelling, spurious concepts, and concept annotation errors retraining from scratch. Furthermore, to improve computational efficiency, we present streamlined versions integrating EK-FAC. Experimental results show our ECBMs are efficient and effective.

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

# A  NOTATION TABLE

| Symbol | Description |
|---|---|
| $c = \{p_1, \ldots, p_k\}$ | Set of concepts provided by experts or LLMs. |
| $\mathcal{D} = \{z_i\}_{i=1}^{n}$ | Training dataset, where $z_i = (x_i, y_i, c_i)$. |
| $x_i \in \mathbb{R}^m$ | Feature vector for the $i$-th sample. |
| $y_i \in \mathbb{R}^{d_z}$ | Label for the $i$-th sample, with $d_z$ being the number of classes. |
| $c_i = (c_i^1, \ldots, c_i^k) \in \mathbb{R}^k$ | Concept vector for the $i$-th sample. |
| $\tilde{c}_w^r$ | Corrected concept label for the $w$-th sample and $r$-th concept. |
| $c_i^j$ | Weight of the concept $p_j$ in the concept vector $c_i$. |
| $g : \mathbb{R}^m \to \mathbb{R}^k$ | Concept predictor mapping input space to concept space. |
| $f : \mathbb{R}^k \to \mathbb{R}^{d_z}$ | Label predictor mapping concept space to prediction space. |
| $L_C(g^j(x), c^j)$ | Loss function for the $j$-th concept predictor. |
| $L_{C_j}(g(x), c)$ | Loss function for the $j$-th concept predictor(for simplicity). |
| $L_Y(f(\hat{g}(x)), y)$ | Loss function from concept space to output space. |
| $L_{Y_i}(f, \hat{g})$ | Loss function for the $i$-th input based on $f, \hat{g}$(for simplicity). |
| $H_{\hat{\theta}}$ | Hessian matrix of the loss function with respect to $\hat{\theta}$. |
| $G_{\hat{\theta}}$ | Fisher information matrix of model $\hat{\theta}$. |
| $\lambda$ | Damping term for ensuring positive definiteness of the Hessian. |
| $\hat{g}$ | Estimated concept predictor. |
| $\hat{f}$ | Estimated label predictor. |
| $\hat{g}_e$ | Retrained concept predictor after correcting erroneous data. |
| $\hat{f}_e$ | Retrained label predictor after correcting erroneous data. |
| $\hat{g}_{-p_M}$ | Retrained concept predictor after removing concepts indexed by $M$. |
| $\hat{g}^*_{-p_M}$ | Mapped concept predictor with the same dimensionality as $\hat{g}$. |
| $\bar{g}_{-p_M}$ | Approximation of the retrained concept predictor $\hat{g}_{-p_M}$. |
| $\hat{f}_{p_M=0}$ | Label predictor after setting the $r$-th concept to zero for $r \in M$. |
| $\bar{f}_{p_M=0}$ | Approximation of the label predictor $\hat{f}_{p_M=0}$. |
| $H_{\hat{g}}$ | Hessian matrix of the loss function with respect to $\hat{g}$. |
| $H_{\hat{f}}$ | Hessian matrix of the loss function with respect to $\hat{f}$. |
| $M \subset [k]$ | Set of erroneous concept indices to be removed. |
| $G \subset [n]$ | Set of indices of samples to be removed from the dataset. |
| $z_r = (x_r, y_r, c_r)$ | Data sample to be removed, where $r \in G$. |
| $\hat{g}_{-z_G}$ | Retrained concept predictor after removing samples indexed by $G$. |
| $\bar{g}_{-z_G}$ | Approximation of the retrained concept predictor $\hat{g}_{-z_G}$. |
| $\hat{f}_{-z_G}$ | Intermediate label predictor. |
| $\bar{f}_{-z_G}$ | Final edited label predictor after removing samples indexed by $G$. |

Table 2: Notation Table

# B  INFLUENCE FUNCTION

Consider a neural network $\hat{\theta} = \arg\min_\theta \sum_{i=1}^{n} \ell(z_i, \theta)$ with loss function $L$ and dataset $D = \{z_i\}_{i=1}^{n}$. That is $\hat{\theta}$ minimize the empirical risk

$$R(\theta) = \sum_{i=1}^{n} L(z_i, \theta)$$

Assume $R$ is strongly convex in $\theta$. Then $\theta$ is uniquely defined. If we remove a point $z_m$ from the training dataset, the parameters become $\hat{\theta}_{-z_m} = \arg\min_\theta \sum_{i \neq m} L(z_i, \theta)$. Up-weighting $z_m$ by $\epsilon$ small enough, then the revised risk $R(\theta)' = \frac{1}{n} \sum_{i=1}^{n} L(z_i; \theta) + \epsilon L(z_m; \theta)$ is still strongly convex. Then the response function $\hat{\theta}_{\epsilon, -z_m} = R(\theta)'$ is also uniquely defined. The parameter change

is denoted as $\Delta_\epsilon = \hat\theta_{\epsilon,-z_m} - \hat\theta$. Since $\hat\theta_{\epsilon,-z_m}$ is the minimizer of $R(\theta)'$, we have the first-order optimization condition as

$$\nabla_{\hat\theta_{\epsilon,-z_m}} R(\theta) + \epsilon \cdot \nabla_{\hat\theta_{\epsilon,-z_m}} L(z_m, \hat\theta_{\epsilon,-z_m}) = 0$$

Since $\hat\theta_{\epsilon,-z_m} \to \hat\theta \ as \ \epsilon \to 0$, we perform a Taylor expansion of the right-hand side:

$$\left[\nabla R(\hat\theta) + \epsilon\nabla L(z_m, \hat\theta)\right] + \left[\nabla^2 R(\hat\theta) + \epsilon\nabla^2 L(z_m, \hat\theta)\right]\Delta_\epsilon \approx 0$$

Noting $\epsilon\nabla^2 L(z_m, \hat\theta)\Delta_\epsilon$ is $o(\|\Delta_\epsilon\|)$ term, which is smaller than other parts, we drop it in the following analysis. Then the Taylor expansion equation becomes

$$\left[\nabla R(\hat\theta) + \epsilon\nabla L(z_m, \hat\theta)\right] + \nabla^2 R(\hat\theta) \cdot \Delta_\epsilon \approx 0$$

Solving for $\Delta_\epsilon$, we obtain:

$$\Delta_\epsilon = -\left[\nabla^2 R(\hat\theta) + \epsilon\nabla^2 L(z, \hat\theta)\right]^{-1}\left[\nabla R(\hat\theta) + \epsilon\nabla L(z, \hat\theta)\right].$$

Remember $\theta$ minimizes $R$, then $\nabla R(\hat\theta) = 0$. Dropping $o(\epsilon)$ term, we have

$$\Delta_\epsilon = -\epsilon\nabla^2 R(\hat\theta)^{-1}\nabla L(z, \hat\theta).$$

$$\left.\frac{d\hat\theta_{\epsilon,-z_m}}{d\epsilon}\right|_{\epsilon=0} = \left.\frac{d\Delta_\epsilon}{d\epsilon}\right|_{\epsilon=0} = -H_{\hat\theta}^{-1}\nabla L(z, \hat\theta) \equiv \mathcal{I}_{up,params}(z).$$

Besides, we can obtain the approximation of $\hat\theta_{-z_m}$ directly by $\hat\theta_{-z_m} \approx \hat\theta + \mathcal{I}_{up,params}(z)$.

## C  ACCELERATION FOR INFLUENCE FUNCTION

**EK-FAC.**  EK-FAC method relies on two approximations to the Fisher information matrix, equivalent to $G_{\hat\theta}$ in our setting, which makes it feasible to compute the inverse of the matrix.

Firstly, assume that the derivatives of the weights in different layers are uncorrelated, which implies that $G_{\hat\theta}$ has a block-diagonal structure. Suppose $\hat{g}_\theta$ can be denoted by $\hat{g}_\theta(x) = g_{\theta_L} \circ \cdots \circ g_{\theta_l} \circ \cdots \circ g_{\theta_1}(x)$ where $l \in [L]$. We fold the bias into the weights and vectorize the parameters in the $l$-th layer into a vector $\theta_l \in \mathbb{R}^{d_l}$, $d_l \in \mathbb{N}$ is the number of $l$-th layer parameters. Then $G_{\hat\theta}$ can be reaplcaed by $\left(G_1(\hat\theta), \cdots, G_L(\hat\theta)\right)$, where $G_l(\hat\theta) \triangleq n^{-1}\sum_{i=1}^{n}\nabla_{\hat\theta_l}\ell_i\nabla_{\theta_l}\ell_i^{\mathrm{T}}$. Denote $h_l$, $o_l$ as the output and pre-activated output of $l$-th layer. Then $G_l(\theta)$ can be approximated by

$$G_l(\theta) \approx \hat{G}_l(\theta) \triangleq \frac{1}{n}\sum_{i=1}^{n} h_{l-1}(x_i) h_{l-1}(x_i)^T \otimes \frac{1}{n}\sum_{i=1}^{n}\nabla_{o_l}\ell_i\nabla_{o_l}\ell_i^T \triangleq \Omega_{l-1} \otimes \Gamma_l.$$

Furthermore, in order to accelerate transpose operation and introduce the damping term, perform eigenvalue decomposition of matrix $\Omega_{l-1}$ and $\Gamma_l$ and obtain the corresponding decomposition results as $Q_\Omega\Lambda_\Omega Q_\Omega^\top$ and $Q_\Gamma\Lambda_\Gamma Q_\Gamma^\top$. Then the inverse of $\hat{H}_l(\theta)$ can be obtained by

$$\hat{H}_l(\theta)^{-1} \approx \left(\hat{G}_l(\hat{g}) + \lambda_l I_{d_l}\right)^{-1} = \left(Q_{\Omega_{l-1}} \otimes Q_{\Gamma_l}\right)\left(\Lambda_{\Omega_{l-1}} \otimes \Lambda_{\Gamma_l} + \lambda_l I_{d_l}\right)^{-1}\left(Q_{\Omega_{l-1}} \otimes Q_{\Gamma_l}\right)^{\mathrm{T}}.$$

Besides, George et al. (2018) proposed a new method that corrects the error in equation 14 which sets the $i$-th diagonal element of $\Lambda_{\Omega_{l-1}} \otimes \Lambda_{\Gamma_l}$ as $\Lambda_{ii}^* = n^{-1}\sum_{j=1}^{n}\left(\left(Q_{\Omega_{l-1}} \otimes Q_{\Gamma_l}\right)\nabla_{\theta_l}\ell_j\right)_i^2$.

### C.1  EK-FAC FOR CBMS

In our CBM model, the label predictor is a single linear layer, and Hessian computing costs are affordable. However, the concept predictor is based on Resnet-18, which has many parameters. Therefore, we perform EK-FAC for $\hat{g}$.

$$\hat{g} = \arg\min_g \sum_{j=1}^{k} L_{C_j} = \arg\min_g \sum_{j=1}^{k}\sum_{i=1}^{n} L_C(g^j(x_i), c_i^j),$$

we define $H_{\hat{g}} = \nabla_{\hat{g}}^2 \sum_{i,j} L_{C_j}(g(x_i), c_i)$ as the Hessian matrix of the loss function with respect to the parameters.

To this end, consider the $l$-th layer of $\hat{g}$ which takes as input a layer of activations $\{a_{j,t}\}$ where $j \in \{1, 2, \ldots, J\}$ indexes the input map and $t \in \mathcal{T}$ indexes the spatial location which is typically a 2-D grid. This layer is parameterized by a set of weights $W = (w_{i,j,\delta})$ and biases $b = (b_i)$, where $i \in \{1, \ldots, I\}$ indexes the output map, and $\delta \in \Delta$ indexes the spatial offset (from the center of the filter).

The convolution layer computes a set of pre-activations as

$$[S_l]_{i,t} = s_{i,t} = \sum_{\delta \in \Delta} w_{i,j,\delta} a_{j,t+\delta} + b_i.$$

Denote the loss derivative with respect to $s_{i,t}$ as

$$\mathcal{D}s_{i,t} = \frac{\partial \sum L_{C_j}}{\partial s_{i,t}},$$

which can be computed during backpropagation.

The activations are actually stored as $A_{l-1}$ of dimension $|\mathcal{T}| \times J$. Similarly, the weights are stored as an $I \times |\Delta|J$ array $W_l$. The straightforward implementation of convolution, though highly parallel in theory, suffers from poor memory access patterns. Instead, efficient implementations typically leverage what is known as the expansion operator $[\![\cdot]\!]$. For instance, $[\![A_{l-1}]\!]$ is a $|\mathcal{T}| \times J|\Delta|$ matrix, defined as

$$[\![A_{l-1}]\!]_{t,j|\Delta|+\delta} = [A_{l-1}]_{(t+\delta),j} = a_{j,t+\delta},$$

In order to fold the bias into the weights, we need to add a homogeneous coordinate (i.e. a column of all 1's) to the expanded activations $[\![A_{l-1}]\!]$ and denote this as $[\![A_{l-1}]\!]_{\mathrm{H}}$. Concatenating the bias vector to the weights matrix, then we have $\theta_l = (b_l, W_l)$.

Then, the approximation for $H_{\hat{g}}$ is given as:

$$G^{(l)}(\hat{g}) = \mathbb{E}\left[\mathcal{D}w_{i,j,\delta} \mathcal{D}w_{i',j',\delta'}\right] = \mathbb{E}\left[\left(\sum_{t \in \mathcal{T}} a_{j,t+\delta} \mathcal{D}s_{i,t}\right)\left(\sum_{t' \in \mathcal{T}} a_{j',t'+\delta'} \mathcal{D}s_{i',t'}\right)\right]$$

$$\approx \mathbb{E}\left[[\![A_{l-1}]\!]_{\mathrm{H}}^{\top} [\![A_{l-1}]\!]_{\mathrm{H}}\right] \otimes \frac{1}{|\mathcal{T}|} \mathbb{E}\left[\mathcal{D}S_l^{\top} \mathcal{D}S_l\right] \triangleq \Omega_{l-1} \otimes \Gamma_l.$$

Estimate the expectation using the mean of the training set,

$$G^{(l)}(\hat{g}) \approx \frac{1}{n} \sum_{i=1}^{n} \left([\![A_{l-1}^i]\!]_{\mathrm{H}}^{\top} [\![A_{l-1}^i]\!]_{\mathrm{H}}\right) \otimes \frac{1}{n} \sum_{i=1}^{n} \left(\frac{1}{|\mathcal{T}|} \mathcal{D}S_l^{i\top} \mathcal{D}S_l^i\right) \triangleq \hat{\Omega}_{l-1} \otimes \hat{\Gamma}_l.$$

Furthermore, if the factors $\hat{\Omega}_{l-1}$ and $\hat{\Gamma}_l$ have eigen decomposition $Q_\Omega \Lambda_\Omega Q_\Omega^\top$ and $Q_\Gamma \Lambda_\Gamma Q_\Gamma^\top$, respectively, then the eigen decomposition of $\hat{\Omega}_{l-1} \otimes \hat{\Gamma}_l$ can be written as:

$$\hat{\Omega}_{l-1} \otimes \hat{\Gamma}_l = Q_\Omega \Lambda_\Omega Q_\Omega^\top \otimes Q_\Gamma \Lambda_\Gamma Q_\Gamma^\top$$

$$= (Q_\Omega \otimes Q_\Gamma)(\Lambda_\Omega \otimes \Lambda_\Gamma)(Q_\Omega \otimes Q_\Gamma)^\top.$$

Since subsequent inverse operations are required and the current approximation for $G^{(l)}(\hat{g})$ is PSD, we actually use a damped version as

$$\hat{G}^l(\hat{g})^{-1} = (G_l(\hat{g}) + \lambda_l I_{d_l})^{-1} = \left(Q_{\Omega_{l-1}} \otimes Q_{\Gamma_l}\right)\left(\Lambda_{\Omega_{l-1}} \otimes \Lambda_{\Gamma_l} + \lambda_l I_{d_l}\right)^{-1}\left(Q_{\Omega_{l-1}} \otimes Q_{\Gamma_l}\right)^{\mathrm{T}}. \tag{14}$$

Besides, George et al. (2018) proposed a new method that corrects the error in equation 14 which sets the $i$-th diagonal element of $\Lambda_{\Omega_{l-1}} \otimes \Lambda_{\Gamma_l}$ as

$$\Lambda_{ii}^* = n^{-1} \sum_{j=1}^{n} \left(\left(Q_{\Omega_{l-1}} \otimes Q_{\Gamma_l}\right) \nabla_{\theta_l} \ell_j\right)_i^2.$$

## D   PROOF OF CONCEPT-LABEL-LEVEL INFLUENCE

We have a set of erroneous data $D_e$ and its associated index set $S_e \subseteq [n] \times [k]$ such that for each $(w, r) \in S_e$, we have $(x_w, y_w, c_w) \in D_e$ with $c_w^r$ is mislabeled and $\tilde{c}_w^r$ is its corrected concept label. Thus, our goal is to approximate the new CBM without retraining.

**Proof Sketch.**   Our goal is to edit $\hat{g}$ and $\hat{f}$ to $\hat{g}_e$ and $\hat{f}_e$. (i) First, we introduce new parameters $\hat{g}_{\epsilon,e}$ that minimize a modified loss function with a small perturbation $\epsilon$. (ii) Then, we perform a Newton step around $\hat{g}$ and obtain an estimate for $\hat{g}_e$. (iii) Then, we consider changing the concept predictor at one data point $(x_{i_c}, y_{i_c}, c_{i_c})$ and retraining the model to obtain a new label predictor $\hat{f}_{i_c}$, obtain an approximation for $\hat{f}_{i_c}$. (iv) Next, we iterate $i_c$ over $1, 2, \cdots, n$, sum all the equations together, and perform a Newton step around $\hat{f}$ to obtain an approximation for $\hat{f}_e$. (v) Finally, we bring the estimate of $\hat{g}$ into the equation for $\hat{f}_e$ to obtain the final approximation.

**Theorem D.1.** *The retrained concept predictor $\hat{g}_e$ defined by*

$$\hat{g}_e = \arg\min \left[ \sum_{(i,j) \notin S_e} L_C\left(g^j(x_i), c_i^j\right) + \sum_{(i,j) \in S_e} L_C\left(g^j(x_i), \tilde{c}_i^j\right) \right], \tag{15}$$

*can be approximated by:*

$$\hat{g}_e \approx \bar{g}_e \triangleq \hat{g} - H_{\hat{g}}^{-1} \cdot \sum_{(w,r) \in S_e} \left(\nabla_{\hat{g}} L_C\left(\hat{g}^r(x_w), \tilde{c}_w^r\right) - \nabla_{\hat{g}} L_C\left(\hat{g}^r(x_w), c_w^r\right)\right), \tag{16}$$

*where $H_{\hat{g}} = \nabla_{\hat{g}}^2 \sum_{i,j} L_C(\hat{g}^j(x_i), c_i^j)$ is the Hessian matrix of the loss function respect to $\hat{g}$.*

*Proof.* For the index $(w, r) \in S_e$, indicating the $r$-th concept of the $w$-th data is wrong, we correct this concept $c_w^r$ to $\tilde{c}_w^r$. Rewrite $\hat{g}_e$ as

$$\hat{g}_e = \arg\min \left[ \sum_{i,j} L_C\left(g^j(x_i), c_i^j\right) + \sum_{(w,r) \in S_e} L_C\left(g^r(x_w), \tilde{c}_w^r\right) - \sum_{(w,r) \in S_e} L_C\left(g^r(x_w), c_w^r\right) \right]. \tag{17}$$

To approximate this effect, define new parameters $\hat{g}_{\epsilon,e}$ as

$$\hat{g}_{\epsilon,e} \triangleq \arg\min \left[ \sum_{i,j} L_C\left(g^j(x_i), c_i^j\right) + \sum_{(w,r) \in S_e} \epsilon \cdot L_C\left(g^r(x_w), \tilde{c}_w^r\right) - \sum_{(w,r) \in S_e} \epsilon \cdot L_C\left(g^r(x_w), c_w^r\right) \right]. \tag{18}$$

Then, because $\hat{g}_{\epsilon,e}$ minimizes equation 18, we have

$$\nabla_{\hat{g}} \sum_{i,j} L_C\left(\hat{g}_{\epsilon,e}^j(x_i), c_i^j\right) + \sum_{(w,r) \in S_e} \epsilon \cdot \nabla_{\hat{g}} L_C\left(\hat{g}_{\epsilon,e}^r(x_w), \tilde{c}_w^r\right) - \sum_{(w,r) \in S_e} \epsilon \cdot \nabla_{\hat{g}} L_C\left(\hat{g}_{\epsilon,e}^r(x_w), c_w^r\right) = 0.$$

Perform a Taylor expansion of the above equation at $\hat{g}$,

$$\nabla_{\hat{g}} \sum_{i,j} L_C\left(\hat{g}^j(x_i), c_i^j\right) + \sum_{(w,r) \in S_e} \epsilon \cdot \nabla_{\hat{g}} L_C\left(\hat{g}^r(x_w), \tilde{c}_w^r\right) - \sum_{(w,r) \in S_e} \epsilon \cdot \nabla_{\hat{g}} L_C\left(\hat{g}^r(x_w), c_w^r\right)$$
$$+ \nabla_{\hat{g}}^2 \sum_{i,j} L_C\left(\hat{g}^j(x_i), c_i^j\right) \cdot (\hat{g}_{\epsilon,e} - \hat{g}) \approx 0. \tag{19}$$

Because of equation 15, the first term of equation 19 equals 0. Then we have

$$\hat{g}_{\epsilon,e} - \hat{g} = - \sum_{(w,r) \in S_e} \epsilon \cdot H_{\hat{g}}^{-1} \cdot \left(\nabla_{\hat{g}} L_C\left(\hat{g}^r(x_w), \tilde{c}_w^r\right) - \nabla_{\hat{g}} L_C\left(\hat{g}^r(x_w), c_w^r\right)\right),$$

where

$$H_{\hat{g}} = \nabla_{\hat{g}}^2 \sum_{i,j} L_C \left( \hat{g}^j(x_i), c_i^j \right).$$

Then, we do a Newton step around $\hat{g}$ and obtain

$$\hat{g}_e \approx \bar{g}_e \triangleq \hat{g} - H_{\hat{g}}^{-1} \cdot \sum_{(w,r) \in S_e} \left( \nabla_{\hat{g}} L_C \left( \hat{g}^r(x_w), \tilde{c}_w^r \right) - \nabla_{\hat{g}} L_C \left( \hat{g}^r(x_w), c_w^r \right) \right). \tag{20}$$

$\square$

**Theorem D.2.** *The retrained label predictor $\hat{f}_e$ defined by*

$$\hat{f}_e = \arg\min \left[ \sum_{i=1}^{n} L_Y \left( f \left( \hat{g}_e(x_i) \right), y_i \right) \right]$$

*can be approximated by:*

$$\hat{f}_e \approx \bar{f}_e = \hat{f} + H_{\hat{f}}^{-1} \cdot \nabla_f \sum_{i=1}^{n} L_{Y_i} \left( \hat{f}, \hat{g} \right) - H_{\hat{f}}^{-1} \cdot \nabla_f \sum_{i=1}^{n} L_{Y_i} \left( \hat{f}, \bar{g}_e \right),$$

*where $H_{\hat{f}} = \nabla_{\hat{f}}^2 \sum_{i=1}^{n} L_{Y_i}(\hat{f}, \hat{g})$ is the Hessian matrix of the loss function respect to $\hat{f}$, $L_{Y_i}(\hat{f}, \hat{g}) \triangleq L_Y(\hat{f}(\hat{g}(x_i)), y_i)$, and $\bar{g}_e$ is given in Theorem D.1.*

*Proof.* Now we come to deduce the edited label predictor towards $\hat{f}_e$.

First, we consider only changing the concept predictor at one data point $(x_{i_c}, y_{i_c}, c_{i_c})$ and retrain the model to obtain a new label predictor $\hat{f}_{i_c}$.

$$\hat{f}_{i_c} = \arg\min \left[ \sum_{i=1, i \neq i_c}^{n} L_Y \left( f \left( \hat{g}(x_i) \right), y_i \right) + L_Y \left( f \left( \hat{g}_e(x_{i_c}) \right), y_{i_c} \right) \right].$$

We rewrite the above equation as follows:

$$\hat{f}_{i_c} = \arg\min \left[ \sum_{i=1}^{n} L_Y \left( f \left( \hat{g}(x_i) \right), y_i \right) + L_Y \left( f \left( \hat{g}_e(x_{i_c}) \right), y_{i_c} \right) - L_Y \left( f \left( \hat{g}(x_{i_c}) \right), y_{i_c} \right) \right].$$

We define $\hat{f}_{\epsilon, i_c}$ as:

$$\hat{f}_{\epsilon, i_c} = \arg\min \left[ \sum_{i=1}^{n} L_Y \left( f \left( \hat{g}(x_i) \right), y_i \right) + \epsilon \cdot L_Y \left( f \left( \hat{g}_e(x_{i_c}) \right), y_{i_c} \right) - \epsilon \cdot L_Y \left( f \left( \hat{g}(x_{i_c}) \right), y_{i_c} \right) \right].$$

Derive with respect to $f$ at both sides of the above equation. we have

$$\nabla_{\hat{f}} \sum_{i=1}^{n} L_Y \left( \hat{f}_{\epsilon, i_c} \left( \hat{g}(x_i) \right), y_i \right) + \epsilon \cdot \nabla_{\hat{f}} L_Y \left( \hat{f}_{\epsilon, i_c} \left( \hat{g}_e(x_{i_c}) \right), y_{i_c} \right) - \epsilon \cdot \nabla_{\hat{f}} L_Y \left( \hat{f}_{\epsilon, i_c} \left( \hat{g}(x_{i_c}) \right), y_{i_c} \right) = 0$$

Perform a Taylor expansion of the above equation at $\hat{f}$,

$$\nabla_{\hat{f}} \sum_{i=1}^{n} L_Y \left( \hat{f} \left( \hat{g}(x_i) \right), y_i \right) + \epsilon \cdot \nabla_{\hat{f}} L_Y \left( \hat{f} \left( \hat{g}_e(x_{i_c}) \right), y_{i_c} \right)$$

$$- \epsilon \cdot \nabla_{\hat{f}} L_Y \left( \hat{f} \left( \hat{g}(x_{i_c}) \right), y_{i_c} \right) + \nabla_{\hat{f}}^2 \sum_{i=1}^{n} L_Y \left( \hat{f} \left( \hat{g}(x_i) \right), y_i \right) \cdot \left( \hat{f}_{\epsilon, i_c} - \hat{f} \right) = 0$$

Then we have

$$\hat{f}_{\epsilon,i_c} - \hat{f} \approx -\epsilon \cdot H_{\hat{f}}^{-1} \cdot \nabla_f \left( L_Y \left( \hat{f} \left( \hat{g}_e \left( x_{i_c} \right) \right), y_{i_c} \right) - L_Y \left( \hat{f} \left( \hat{g} \left( x_{i_c} \right) \right), y_{i_c} \right) \right),$$

where $H_{\hat{f}}^{-1} = \nabla_{\hat{f}}^2 \sum_{i=1}^{n} L_Y \left( \hat{f} \left( \hat{g} \left( x_i \right) \right), y_i \right)$.

Iterate $i_c$ over $1, 2, \cdots, n$, and sum all the equations together, we can obtain:

$$\hat{f}_{\epsilon,e} - \hat{f} \approx -\epsilon \cdot H_{\hat{f}}^{-1} \cdot \sum_{i=1}^{n} \nabla_f \left( L_Y \left( \hat{f} \left( \hat{g}_e \left( x_i \right) \right), y_i \right) - L_Y \left( \hat{f} \left( \hat{g} \left( x_i \right) \right), y_i \right) \right).$$

Perform a Newton step around $\hat{f}$ and we have

$$\hat{f}_e \approx \hat{f} - H_{\hat{f}}^{-1} \cdot \sum_{i=1}^{n} \nabla_f \left( L_Y \left( \hat{f} \left( \hat{g}_e \left( x_i \right) \right), y_i \right) - L_Y \left( \hat{f} \left( \hat{g} \left( x_i \right) \right), y_i \right) \right). \tag{21}$$

Bringing the edited 20 of $g$ into equation 21, we have

$$\hat{f}_e \approx \hat{f} - H_{\hat{f}}^{-1} \cdot \sum_{i=1}^{n} \nabla_f \left( L_Y \left( \hat{f} \left( \bar{g}_e \left( x_i \right) \right), y_i \right) - L_Y \left( \hat{f} \left( \hat{g} \left( x_i \right) \right), y_i \right) \right)$$

$$= \hat{f} - H_{\hat{f}}^{-1} \cdot \sum_{i=1}^{n} \nabla_f \left( L_{Y_i} \left( \hat{f}, \bar{g}_e \right) - L_{Y_i} \left( \hat{f}, \hat{g} \right) \right)$$

$$= \hat{f} + H_{\hat{f}}^{-1} \cdot \nabla_f \sum_{i=1}^{n} L_{Y_i} \left( \hat{f}, \hat{g} \right) - H_{\hat{f}}^{-1} \cdot \nabla_f \sum_{i=1}^{n} L_{Y_i} \left( \hat{f}, \bar{g}_e \right) \triangleq \bar{f}_e.$$

$\square$

## E    PROOF OF CONCEPT-LEVEL INFLUENCE

We address situations that delete $p_r$ for $r \in M$ concept removed dataset. Our goal is to estimate $\hat{g}_{-p_M}, \hat{f}_{-p_M}$, which is the concept and label predictor trained on the $p_r$ for $r \in M$ concept removed dataset.

**Proof Sketch.**    The main ideas are as follows: (i) First, we define a new predictor $\hat{g}_{p_M}^*$, which has the same dimension as $\hat{g}$ and the same output as $\hat{g}_{-p_M}$. Then deduce an approximation for $\hat{g}_{p_M}^*$. (ii) Then, we consider setting $p_r = 0$ instead of removing it, we get $\hat{f}_{p_M=0}$, which is equivalent to $\hat{f}_{-p_M}$ according to lemma E.1. We estimate this new predictor as a substitute. (iii) Next, we assume we only use the updated concept predictor $\hat{g}_{p_M}^*$ for one data $(x_{i_r}, y_{i_r}, c_{i_r})$ and obtain a new label predictor $\hat{f}_{ir}$, and obtain a one-step Newtonian iterative approximation of $\hat{f}_{ir}$ with respect to $\hat{f}$. (iv) Finally, we repeat the above process for all data points and combine the estimate of $\hat{g}$ in Theorem $E.2$, we obtain a closed-form solution of the influence function for $\hat{f}$.

First, we introduce our following lemma:

**Lemma E.1.** *For the concept bottleneck model, if the label predictor utilizes linear transformations of the form $\hat{f} \cdot c$ with input c,then, for each $r \in M$, we remove the $r$-th concept from c and denote the new input as $c'$. Set the $r$-th concept to 0 and denote the new input as $c^0$. Then we have $\hat{f}_{-p_M} \cdot c' = \hat{f}_{p_M=0} \cdot c^0$ for any c.*

*Proof.* Assume the parameter space of $\hat{f}_{-p_M}$ and $\hat{f}_{p_M=0}$ are $\Gamma$ and $\Gamma_0$, respectively, then there exists a surjection $P : \Gamma \to \Gamma_0$. For any $\theta \in \Gamma$, $P(\theta)$ is the operation that removes the $r$-th row of $\theta$ for $r \in M$. Then we have:

$$P(\theta) \cdot c' = \sum_{t \notin M} \theta[j] \cdot c'[j] = \sum_{t} \theta[t] \mathbb{I}\{t \notin M\} c[t] = \theta \cdot c^0.$$

Thus, the loss function $L_Y(\theta, c^0) = L_Y(P(\theta), c')$ of both models is the same for every sample in the second stage. Besides, by formula derivation, we have, for $\theta' \in \Gamma_0$, for any $\theta$ in $P^{-1}(\theta')$,

$$\frac{\partial L_Y(\theta, c^0)}{\partial \theta} = \frac{\partial L_Y(P(\theta), c')}{\partial \theta'}$$

Thus, if the same initialization is performed, $\hat{f}_{-p_M} \cdot c' = \hat{f}_{p_M=0} \cdot c^0$ for any $c$ in the dataset. $\square$

**Theorem E.2.** *For the retrained concept predictor $\hat{g}_{-p_M}$ defined by*

$$\hat{g}_{-p_M} = \arg\min_{g'} \sum_{j \notin M} L_{C_j} = \arg\min_{g'} \sum_{j \notin M} \sum_{i=1}^{n} L_C(g'^j(x_i), c_i^j),$$

*we map it to $\hat{g}^*_{-p_M}$ as*

$$\hat{g}^*_{-p_M} = \arg\min_{g' \in \Gamma_0} \sum_{j \notin M}^{k} \sum_{i=1}^{n} L_C(g'^j(x_i), c_i^j).$$

*And we can edit the initial $\hat{g}$ to $\hat{g}^*_{-p_M}$ as:*

$$\hat{g}^*_{-p_M} \approx \bar{g}^*_{-p_M} \triangleq \hat{g} - H_{\hat{g}}^{-1} \cdot \nabla_{\hat{g}} \sum_{j \notin M} \sum_{i=1}^{n} L_C(\hat{g}^j(x_i), c_i^j), \tag{22}$$

*where $H_{\hat{g}} = \nabla_g^2 \sum_{j \notin M} \sum_{i=1}^{n} L_C(\hat{g}^j(x_i), c_i^j)$.*

*Then, by removing all zero rows inserted during the mapping phase, we can naturally approximate $\hat{g}_{-p_M} \approx P^{-1}(\hat{g}^*_{-p_M})$.*

*Proof.* At this level, we consider the scenario that removes a set of mislabeled concepts or introduces new ones. Because after removing concepts from all the data, the new concept predictor has a different dimension from the original. We denote $g^j(x_i)$ as the $j$-th concept predictor with $x_i$, and $c_i^j$ as the $j$-th concept in data $z_i$. For simplicity, we treat $g$ as a collection of $k$ concept predictors and separate different columns as a vector $g^j(x_i)$. Actually, the neural network gets $g$ as a whole.

For the comparative purpose, we introduce a new notation $\hat{g}^*_{-p_M}$. Specifically, we define weights of $\hat{g}$ and $\hat{g}^*_{-p_M}$ for the last layer of the neural network as follows.

$$\hat{g}_{-p_M}(x) = \underbrace{\begin{pmatrix} w_{11} & w_{12} & \cdots & w_{1m} \\ w_{21} & w_{22} & \cdots & w_{2m} \\ \vdots & \vdots & & \vdots \\ w_{(k-1)1} & w_{(k-1)2} & \cdots & w_{(k-1)m} \end{pmatrix}}_{(k-1)\times m} \cdot \underbrace{\begin{pmatrix} x^1 \\ x^2 \\ \vdots \\ x^m \end{pmatrix}}_{m \times 1} = \underbrace{\begin{pmatrix} c_1 \\ \vdots \\ c_{r-1} \\ c_{r+1} \\ \vdots \\ c_k \end{pmatrix}}_{(k-1)\times 1}$$

$$\hat{g}^*_{-p_M}(x) = \underbrace{\begin{pmatrix} w_{11} & w_{12} & \cdots & w_{1m} \\ \vdots & \vdots & & \vdots \\ w_{(r-1)1} & w_{(r-1)2} & \cdots & w_{(r-1)m} \\ 0 & 0 & \cdots & 0 \\ w_{(r+1)1} & w_{(r+1)2} & \cdots & w_{(r+1)m} \\ \vdots & \vdots & & \vdots \\ w_{k1} & w_{k2} & \cdots & w_{km} \end{pmatrix}}_{k \times m} \cdot \underbrace{\begin{pmatrix} x^1 \\ \vdots \\ x^{r-1} \\ x^r \\ x^{r+1} \\ \vdots \\ x^m \end{pmatrix}}_{m \times 1} = \underbrace{\begin{pmatrix} c_1 \\ \vdots \\ c_{r-1} \\ 0 \\ c_{r+1} \\ \vdots \\ c_k \end{pmatrix}}_{k \times 1},$$

where $r$ is an index from the index set $M$.

Firstly, we want to edit to $\hat{g}^*_{-p_M} \in T_0 = \{w_{\text{final}} = 0\} \subseteq T$ based on $\hat{g}$, where $w_{\text{final}}$ is the parameter of the final layer of neural network. Let us take a look at the definition of $\hat{g}^*_{-p_M}$:

$$\hat{g}^*_{-p_M} = \arg\min_{g' \in T_0} \sum_{j \notin M} \sum_{i=1}^{n} L_C(g'^j(x_i), c_i^j).$$

Then, we separate the $r$-th concept-related item from the rest and rewrite $\hat{g}$ as the following form:

$$\hat{g} = \arg\min_{g \in T} \left[ \sum_{j=1, j \neq r}^{k} \sum_{i=1}^{n} L_C(g^j(x_i), c_i^j) + \sum_{r \in M} \sum_{i=1}^{n} L_C(g^r(x_i), c_i^r) \right].$$

Then, if the $r$-th concept part is up-weighted by some small $\epsilon$, this gives us the new parameters $\hat{g}_{\epsilon, p_M}$, which we will abbreviate as $\hat{g}_\epsilon$ below.

$$\hat{g}_{\epsilon, p_M} \triangleq \arg\min_{g \in T} \left[ \sum_{j \notin M} \sum_{i=1}^{n} L_C(g^j(x_i), c_i^j) + \epsilon \cdot \sum_{r \in M} \sum_{i=1}^{n} L_C(g^r(x_i), c_i^r) \right].$$

Obviously, when $\epsilon \to 0$, $\hat{g}_\epsilon \to \hat{g}^*_{-p_M}$. We can obtain the minimization conditions from the definitions above.

$$\nabla_{\hat{g}^*_{-p_M}} \sum_{j \notin M} \sum_{i=1}^{n} L_{C_j}(\hat{g}^*_{-p_M}(x_i), c_i) = 0. \tag{23}$$

$$\nabla_{\hat{g}_\epsilon} \sum_{j \notin M} \sum_{i=1}^{n} L_{C_j}(\hat{g}_\epsilon(x_i), c_i) + \epsilon \cdot \nabla_{\hat{g}_\epsilon} \sum_{r \in M} \sum_{i=1}^{n} L_{C_r}(\hat{g}_\epsilon(x_i), c_i) = 0.$$

Perform a first-order Taylor expansion of equation 23 with respect to $\hat{g}_\epsilon$, then we get

$$\nabla_g \sum_{j \notin M} \sum_{i=1}^{n} L_{C_j}(\hat{g}_\epsilon(x_i), c_i) + \nabla_g^2 \sum_{j \notin M} \sum_{i=1}^{n} L_{C_j}(\hat{g}_\epsilon(x_i), c_i) \cdot (\hat{g}^*_{-p_M} - \hat{g}_\epsilon) \approx 0.$$

Then we have

$$\hat{g}^*_{-p_M} - \hat{g}_\epsilon = -H_{\hat{g}_\epsilon}^{-1} \cdot \nabla_g \sum_{j \notin M} \sum_{i=1}^{n} L_{C_j}(\hat{g}_\epsilon(x_i), c_i).$$

Where $H_{\hat{g}_\epsilon} = \nabla_g^2 \sum_{j \notin M} \sum_{i=1}^{n} L_{C_j}(\hat{g}_\epsilon(x_i), c_i)$.

We can see that:

When $\epsilon = 0$,

$$\hat{g}_\epsilon = \hat{g}^*_{-p_M},$$

When $\epsilon = 1$, $\hat{g}_\epsilon = \hat{g}$,

$$\hat{g}^*_{-p_M} - \hat{g} \approx -H_{\hat{g}}^{-1} \cdot \nabla_g \sum_{j \notin M} \sum_{i=1}^{n} L_{C_j}(\hat{g}(x_i), c_i),$$

where $H_{\hat{g}} = \nabla_g^2 \sum_{j \notin M} \sum_{i=1}^{n} L_{C_j}(\hat{g}(x_i), c_i)$.

Then, an approximation of $\hat{g}^*_{-p_M}$ is obtained.

$$\hat{g}^*_{-p_M} \approx \hat{g} - H_{\hat{g}}^{-1} \cdot \nabla_g \sum_{j \notin M} \sum_{i=1}^{n} L_{C_j}(\hat{g}(x_i), c_i). \tag{24}$$

$\square$

**Theorem E.3.** *For the retrained label predictor $\hat{f}_{-p_M}$ defined as*

$$\hat{f}_{-p_M} = \arg\min_{f'} \sum_{i=1}^{n} L_Y = \arg\min_{f'} \sum_{i=1}^{n} L_Y(f'(\hat{g}_{-p_M}(x_i)), y_i),$$

*We can consider its equivalent version $\hat{f}_{p_M=0}$ as:*

$$\hat{f}_{p_M=0} = \arg\min_{f} \sum_{i=1}^{n} L_{Y_i}\left(f, \hat{g}^*_{-p_M}\right),$$

*which can be edited by*

$$\hat{f}_{p_M=0} \approx \bar{f}_{p_M=0} \triangleq \hat{f} - H_{\hat{f}}^{-1} \cdot \nabla_{\hat{f}} \sum_{l=1}^{n} L_{Y_l}\left(\hat{f}, \bar{g}^*_{-p_M}\right),$$

*where $H_{\hat{f}} = \nabla_{\hat{f}}^2 \sum_{i=1}^{n} L_{Y_i}(\hat{f}, \hat{g})$ is the Hessian matrix of the loss function respect to is the Hessian matrix of the loss function respect to $\hat{f}$.*

*Proof.* Now, we come to the approximation of $\hat{f}_{-p_M}$. Noticing that the input dimension of $f$ decreases to $k - |M|$. We consider setting $p_r = 0$ for all data points in the training phase of the label predictor and get another optimal model $\hat{f}_{p_M=0}$. From lemma E.1, we know that for the same input $x$, $\hat{f}_{p_M=0}(x) = \hat{f}_{-p_M}$. And the values of the corresponding parameters in $\hat{f}_{p_M=0}$ and $\hat{f}_{-p_M}$ are equal.

Now, let us consider how to edit the initial $\hat{f}$ to $\hat{f}_{p_M=0}$. Firstly, assume we only use the updated concept predictor $\hat{g}^*_{-p_M}$ for one data $(x_{i_r}, y_{i_r}, c_{i_r})$ and obtain the following $\hat{f}_{ir}$, which is denoted as

$$\hat{f}_{ir} = \arg\min_{f} \left[ \sum_{i=1}^{n} L_Y(f(\hat{g}(x_i)), y_i) + L_Y(f(\hat{g}^*_{-p_M}(x_{ir})), y_{ir}) - L_Y(f(\hat{g}(x_{ir})), y_{ir}) \right].$$

Then up-weight the $i_r$-th data by some small $\epsilon$ and have the following new parameters:

$$\hat{f}_{\epsilon,ir} = \arg\min_{f} \left[ \sum_{i=1}^{n} L_Y(f(\hat{g}(x_i)), y_i) + \epsilon \cdot L_Y(f(\hat{g}^*_{-p_M}(x_{ir})), y_{ir}) - \epsilon \cdot L_Y(f(\hat{g}(x_{ir})), y_{ir}) \right].$$

Deduce the minimized condition subsequently,

$$\nabla_f \sum_{i=1}^{n} L_Y(\hat{f}_{ir}(\hat{g}(x_i)), y_i) + \epsilon \cdot \nabla_f L_Y(\hat{f}_{ir}(\hat{g}^*_{-p_M}(x_{ir})), y_{ir}) - \epsilon \cdot \nabla_f L_Y(\hat{f}_{ir}(\hat{g}(x_{ir})), y_{ir}) = 0.$$

If we expand first term of $\hat{f}$, which $\hat{f}_{ir,\epsilon} \to \hat{f}(\epsilon \to 0)$, then

$$\nabla_f \sum_{i=1}^{n} L_Y\left(\hat{f}(\hat{g}(x_i)), y_i\right) + \epsilon \cdot \nabla_f L_Y(\hat{f}(\hat{g}^*_{-p_M}(x_{ir})), y_{ir}) - \epsilon \cdot \nabla_f L_Y(\hat{f}(\hat{g}(x_{ir})), y_{ir})$$

$$+ \left( \nabla_f^2 \sum_{i=1}^{n} L_Y\left(\hat{f}(\hat{g}(x_i)), y_i\right) \right) \cdot (\hat{f}_{ir,\epsilon} - \hat{f}) = 0.$$

Note that $\nabla_f \sum_{i=1}^{n} L_Y(\hat{f}(\hat{g}(x_i)), y_i) = 0$. Thus we have

$$\hat{f}_{ir,\epsilon} - \hat{f} = H_{\hat{f}}^{-1} \cdot \epsilon \left( \nabla_f L_Y(\hat{f}(\hat{g}^*_{-p_M}(x_{ir})), y_{ir}) - \nabla_f L_Y(\hat{f}(\hat{g}(x_{ir})), y_{ir}) \right).$$

We conclude that

$$\left. \frac{\mathrm{d}\hat{f}_{\epsilon,ir}}{\mathrm{d}\epsilon} \right|_{\epsilon=0} = H_{\hat{f}}^{-1} \cdot \left( \nabla_{\hat{f}} L_Y(\hat{f}(\hat{g}^*_{-p_M}(x_{ir})), y_{ir}) - \nabla_{\hat{f}} L_Y(\hat{f}(\hat{g}(x_{ir})), y_{ir}) \right).$$

Perform a one-step Newtonian iteration at $\hat{f}$ and we get the approximation of $\hat{f}_{i_r}$.

$$\hat{f}_{ir} \approx \hat{f} + H_{\hat{f}}^{-1} \cdot \left( \nabla_{\hat{f}} L_Y(\hat{f}(\hat{g}(x_{ir})), y_{ir}) - \nabla_{\hat{f}} L_Y(\hat{f}(\hat{g}^*_{-p_M}(x_{ir})), y_{ir}) \right).$$

Reconsider the definition of $\hat{f}_{i_r}$, we use the updated concept predictor $\hat{g}^*_{-p_M}$ for one data $(x_{i_r}, y_{i_r}, c_{i_r})$. Now we carry out this operation for all the other data and estimate $\hat{f}_{p_M=0}$. Combining the minimization condition from the definition of $\hat{f}$, we have

$$\hat{f}_{p_M=0} \approx \hat{f} + H_{\hat{f}}^{-1} \cdot \left( \nabla_{\hat{f}} \sum_{i=1}^{n} L_Y(\hat{f}(\hat{g}(x_i)), y_i) - \nabla_{\hat{f}} \sum_{i=1}^{n} L_Y(\hat{f}(\hat{g}^*_{-p_M}(x_i)), y_i) \right)$$

$$= \hat{f} + H_{\hat{f}}^{-1} \cdot \left( -\nabla_{\hat{f}} \sum_{i=1}^{n} L_Y(\hat{f}(\hat{g}^*_{-p_M}(x_i)), y_i) \right)$$

$$= \hat{f} - H_{\hat{f}}^{-1} \sum_{l=1}^{n} \nabla_{\hat{f}} L_Y(\hat{f}(\hat{g}^*_{-p_M}(x_l)), y_l). \tag{25}$$

Theorem E.2 gives us the edited version of $\hat{g}^*_{-p_M}$. Substitute it into equation 25, and we get the final closed-form edited label predictor under concept level:

$$\hat{f}_{p_M=0} \approx \bar{f}_{p_M=0} \triangleq \hat{f} - H_{\hat{f}}^{-1} \cdot \nabla_{\hat{f}} \sum_{l=1}^{n} L_{Y_l} \left( \hat{f}, \bar{g}^*_{-p_M} \right),$$

where $H_{\hat{f}} = \nabla_{\hat{f}}^2 \sum_{i=1}^{n} L_{Y_i}(\hat{f}, \hat{g})$ is the Hessian matrix of the loss function respect to is the Hessian matrix of the loss function respect to $\hat{f}$. □

# F PROOF OF DATA-LEVEL INFLUENCE

We address situations that for dataset $\mathcal{D} = \{(x_i, y_i, c_i)\}_{i=1}^n$, given a set of data $z_r = (x_r, y_r, c_r)$, $r \in G$ to be removed. Our goal is to estimate $\hat{g}_{-z_G}, \hat{f}_{-z_G}$, which is the concept and label predictor trained on the $z_r$ for $r \in G$ removed dataset.

**Proof Sketch.** (i) First, we estimate the retrained concept predictor $\hat{g}_{-z_G}$. (ii) Then, we define a new label predictor $\tilde{f}_{-z_G}$ and estimate $\tilde{f}_{-z_G} - \hat{f}$. (iii) Next, in order to reduce computational complexity, use the lemma method to obtain the approximation of the Hessian matrix of $\tilde{f}_{-z_G}$. (iv) Next, we compute the difference $\hat{f}_{-z_G} - \tilde{f}_{-z_G}$ as

$$-H_{\tilde{f}_{-z_G}}^{-1} \cdot \left( \nabla_{\hat{f}} L_Y \left( \tilde{f}_{-z_G}(\hat{g}_{-z_G}(x_{i_r})), y_{i_r} \right) - \nabla_{\hat{f}} L_Y \left( \tilde{f}_{-z_G}(\hat{g}(x_{i_r})), y_{i_r} \right) \right).$$

(v) Finally, we divide $\hat{f}_{-z_G} - \hat{f}$, which we actually concerned with, into $\left( \hat{f}_{-z_G} - \tilde{f}_{-z_G} \right) + \left( \tilde{f}_{-z_G} - \hat{f} \right)$.

**Theorem F.1.** *For dataset $\mathcal{D} = \{(x_i, y_i, c_i)\}_{i=1}^n$, given a set of data $z_r = (x_r, y_r, c_r)$, $r \in G$ to be removed. Suppose the updated concept predictor $\hat{g}_{-z_G}$ is defined by*

$$\hat{g}_{-z_G} = \arg\min_g \sum_{j \in [k]} \sum_{i \in [n]-G} L_{C_j}(\hat{g}(x_i), c_i)$$

*where $L_C(\hat{g}(x_i), c_i) \triangleq \sum_{j=1}^{k} L_{C_j}(\hat{g}(x_i), c_i)$. Then we have the following approximation for $\hat{g}_{-z_G}$*

$$\hat{g}_{-z_G} \approx \bar{g}_{-z_G} \triangleq \hat{g} + H_{\hat{g}}^{-1} \cdot \sum_{r \in G} \nabla_g L_C(\hat{g}(x_r), c_r), \tag{26}$$

*where $H_{\hat{g}} = \nabla_{\hat{g}}^2 \sum_{i,j} L_C(\hat{g}^j(x_i), c_i^j)$ is the Hessian matrix of the loss function respect to $\hat{g}$.*

*Proof.* Firstly, we rewrite $\hat{g}_{-z_G}$ as

$$\hat{g}_{-z_G} = \arg\min_g \left[ \sum_{i=1}^n L_C(\hat{g}(x_i), c_i) - \sum_{r \in G} L_C(g(x_r), c_r) \right],$$

Then we up-weighted the $r$-th data by some $\epsilon$ and have a new predictor $\hat{g}_{-z_G,\epsilon}$, which is abbreviated as $\hat{g}_\epsilon$:

$$\hat{g}_\epsilon \triangleq \arg\min_g \left[ \sum_{i=1}^n L_C(g(x_i), c_i) - \epsilon \cdot \sum_{r \in G} L_C(g(x_r), c_r) \right]. \tag{27}$$

Because $\hat{g}_\epsilon$ minimizes the right side of equation 27, we have

$$\nabla_{\hat{g}_\epsilon} \sum_{i=1}^n L_Y(\hat{g}_\epsilon(x_i), c_i) - \epsilon \cdot \nabla_{\hat{g}_\epsilon} \sum_{r \in G} L_Y(\hat{g}_\epsilon(x_r), c_r) = 0.$$

When $\epsilon \to 0$, $\hat{g}_\epsilon \to \hat{g}$. So we can perform a first-order Taylor expansion with respect to $\hat{g}$, and we have

$$\nabla_g \sum_{i=1}^n L_C(\hat{g}(x_i), c_i) - \epsilon \cdot \nabla_g \sum_{r \in G} L_C(\hat{g}(x_r), c_r) + \nabla_g^2 \sum_{i=1}^n L_C(\hat{g}(x_i), c_i) \cdot (\hat{g}_\epsilon - \hat{g}) \approx 0. \tag{28}$$

Recap the definition of $\hat{g}$:

$$\hat{g} = \arg\min_g \sum_{i=1}^n L_Y(g(x_i), c_i),$$

Then, the first term of equation 28 equals 0. Let $\epsilon \to 0$, then we have

$$\left. \frac{d\hat{g}_\epsilon}{d\epsilon} \right|_{\epsilon=0} = H_{\hat{g}}^{-1} \cdot \sum_{r \in G} \nabla_g L_C(\hat{g}(x_r), c_r),$$

where $H_{\hat{g}}^{-1} = \nabla_g^2 \sum_{i=1}^n \ell(\hat{g}(x_i), c_i)$.

Remember when $\epsilon \to 0$, $\hat{g}_\epsilon \to \hat{g}_{-z_G}$. Perform a Newton step at $\hat{g}$, then we obtain the method to edit the original concept predictor under concept level:

$$\hat{g}_{-z_G} \approx \bar{g}_{-z_G} \triangleq \hat{g} + H_{\hat{g}}^{-1} \cdot \sum_{r \in G} \nabla_g L_C(\hat{g}(x_r), c_r).$$

$\square$

**Theorem F.2.** *For dataset $\mathcal{D} = \{(x_i, y_i, c_i)\}_{i=1}^n$, given a set of data $z_r = (x_r, y_r, c_r)$, $r \in G$ to be removed. The label predictor $\hat{f}_{-z_G}$ trained on the revised dataset becomes*

$$\hat{f}_{-z_G} = \arg\min_f \sum_{i \in [n]-G} L_{Y_i}(f, \hat{g}_{-z_G}). \tag{29}$$

*The intermediate label predictor $\tilde{f}_{-z_G}$ is defined by*

$$\tilde{f}_{-z_G} = \arg\min_f \sum_{i \in [n]-G} L_{Y_i}(f, \hat{g}),$$

*Then $\tilde{f}_{-z_G} - \hat{f}$ can be approximated by*

$$\tilde{f}_{-z_G} - \hat{f} \approx H_{\hat{f}}^{-1} \cdot \sum_{i \in [n]-G} \nabla_{\hat{f}} L_{Y_i}(\hat{f}, \hat{g}) \triangleq A_G. \tag{30}$$

*We denote the edited version of $\tilde{f}_{-z_G}$ as $\bar{f}_{-z_G}^* \triangleq \hat{f} + A_G$. And $\hat{f}_{-z_G} - \tilde{f}_{-z_G}$ can be approximated by*

$$\hat{f}_{-z_G} - \tilde{f}_{-z_G} \approx - H_{\bar{f}_{-z_G}^*}^{-1} \cdot \left( \nabla_{\hat{f}} \sum_{i \in [n]-G} L_{Y_i}\left(\bar{f}_{-z_G}^*, \bar{g}_{-z_G}\right) - \nabla_{\hat{f}} \sum_{i \in [n]-G} L_{Y_i}\left(\bar{f}_{-z_G}^*, \hat{g}\right) \right) \triangleq B_G, \tag{31}$$

where $H_{\bar{f}^*_{-z_G}} = \nabla_{\bar{f}} \sum_{i \in [n]-G} L_{Y_i} \left( \bar{f}^*_{-z_G}, \hat{g} \right)$ *is the Hessian matrix of the loss function on the intermediate dataset concerning* $\bar{f}^*_{-z_G}$*. Then, the final edited label predictor* $\bar{f}_{-z_G}$ *can be obtained by*

$$\bar{f}_{-z_G} = \bar{f}^*_{-z_G} + B_G = \hat{f} + A_G + B_G. \tag{32}$$

*Proof.* We can see that there is a huge gap between $\hat{f}_{-z_G}$ and $\hat{f}$. Thus, firstly, we define $\tilde{f}_{-z_G}$ as

$$\tilde{f}_{-z_G} = \arg\min_f \sum_{i=1}^n L_Y \left( f(\hat{g}(x_i)), y_i \right) - \sum_{r \in G} L_Y \left( f(\hat{g}(x_r)), y_r \right).$$

Then, we define $\tilde{f}_{\epsilon, -z_G}$ as follows to estimate $\tilde{f}_{-z_G}$.

$$\tilde{f}_{\epsilon, -z_G} = \arg\min_f \sum_{i=1}^n L_Y \left( f(\hat{g}(x_i)), y_i \right) - \epsilon \cdot \sum_{r \in G} L_Y \left( f(\hat{g}(x_r)), y_r \right).$$

From the minimization condition, we have

$$\nabla_{\tilde{f}} \sum_{i=1}^n L_Y \left( \tilde{f}_{\epsilon, -z_G}(\hat{g}(x_i)), y_i \right) - \epsilon \cdot \sum_{r \in G} \nabla_{\tilde{f}} L_Y \left( \tilde{f}_{\epsilon, -z_G}(\hat{g}(x_r)), y_r \right) = 0.$$

Perform a first-order Taylor expansion at $\hat{f}$,

$$\nabla_{\hat{f}} \sum_{i=1}^n L_Y \left( \hat{f}(\hat{g}(x_i)), y_i \right) - \epsilon \cdot \nabla_{\hat{f}} \sum_{r \in G} L_Y \left( \hat{f}(\hat{g}(x_r)), y_r \right)$$
$$+ \nabla_{\hat{f}}^2 \sum_{i=1}^n L_Y \left( \hat{f}(\hat{g}(x_i)), y_i \right) \cdot \left( \tilde{f}_{\epsilon, -z_G} - \hat{f} \right) = 0.$$

Then $\tilde{f}_{-z_G}$ can be approximated by

$$\tilde{f}_{-z_G} \approx \hat{f} + H_{\hat{f}}^{-1} \cdot \sum_{r \in G} \nabla_{\hat{f}} L_Y \left( \hat{f}(\hat{g}(x_r)), y_r \right) \triangleq A_G. \tag{33}$$

Then the edit version of $\tilde{f}_{-z_G}$ is defined as

$$\bar{f}^*_{-z_G} = \hat{f} + A_G \tag{34}$$

Then we estimate the difference between $\hat{f}_{-z_G}$ and $\tilde{f}_{-z_G}$. Rewrite $\tilde{f}_{-z_G}$ as

$$\tilde{f}_{-z_G} = \arg\min_f \sum_{i \in S}^n L_Y \left( f(\hat{g}(x_i)), y_i \right), \tag{35}$$

where $S \triangleq [n] - G$.

Compare equation 29 with 35, we still need to define an intermediary predictor $\hat{f}_{-z_G, ir}$ as

$$\hat{f}_{-z_G, ir} = \arg\min_f \left[ \sum_{\substack{i \in S \\ i \neq i_r}} L_{Y_i} \left( f, \hat{g}(x_i) \right) + L_{Y_{ir}} \left( f, \hat{g}_{-z_G} \right) \right]$$
$$= \arg\min_f \left[ \sum_{i \in S} L_{Y_i} \left( f, \hat{g} \right) + L_{Y_{ir}} \left( f, \hat{g}_{-z_G} \right) - L_{Y_{ir}} \left( f, \hat{g} \right) \right].$$

Up-weight the $i_r$ data by some $\epsilon$, we define $\hat{f}_{\epsilon,-z_G,i_r}$ as

$$\hat{f}_{\epsilon,-z_G,ir} = \arg\min_f \left[ \sum_{i \in S} L_{Y_i}(f, \hat{g}) + \epsilon \cdot L_{Y_{ir}}(f, \hat{g}_{-z_G}) - \epsilon \cdot L_{Y_{ir}}(f, \hat{g}) \right].$$

We denote $\hat{f}_{\epsilon,-z_G,ir}$ as $\hat{f}_\epsilon^*$ in the following proof. Then, from the minimization condition, we have

$$\nabla_{\hat{f}} \sum_{i \in S} L_{Y_i}\left(\hat{f}_\epsilon^*, \hat{g}\right) + \epsilon \cdot \nabla_{\hat{f}} L_{Y_{ir}}\left(\hat{f}_\epsilon^*, \hat{g}_{-z_G}\right) - \epsilon \cdot \nabla_{\hat{f}} L_{Y_{ir}}\left(\hat{f}_\epsilon^*, \hat{g}(x_{i_r})\right). \tag{36}$$

When $\epsilon \to 0$, $\hat{f}_\epsilon^* \to \tilde{f}_{-z_G}$. Then we perform a Taylor expansion at $\tilde{f}_{-z_G}$ of equation 36 and have

$$\nabla_{\hat{f}} \sum_{i \in S} L_{Y_i}\left(\tilde{f}_{-z_G}, \hat{g}\right) + \epsilon \cdot \nabla_{\hat{f}} L_{Y_{ir}}\left(\tilde{f}_{-z_G}, \hat{g}_{-z_G}\right)$$
$$- \epsilon \cdot \nabla_{\hat{f}} L_{Y_{ir}}\left(\tilde{f}_{-z_G}, \hat{g}\right) + \nabla_{\hat{f}}^2 \sum_{i \in S} L_{Y_i}\left(\tilde{f}_{-z_G}, \hat{g}\right) \cdot (\hat{f}_\epsilon^* - \tilde{f}_{-z_G}) \approx 0.$$

Organizing the above equation gives

$$\hat{f}_\epsilon^* - \tilde{f}_{-z_G} \approx -\epsilon \cdot H_{\tilde{f}_{-z_G}}^{-1} \cdot \left( \nabla_{\hat{f}} L_{Y_{ir}}\left(\tilde{f}_{-z_G}, \hat{g}_{-z_G}\right) - \nabla_{\hat{f}} L_{Y_{ir}}\left(\tilde{f}_{-z_G}, \hat{g}\right) \right),$$

where $H_{\tilde{f}_{-z_G}} = \nabla_{\hat{f}}^2 \sum_{i \in S} L_{Y_i}\left(\tilde{f}_{-z_G}, \hat{g}\right)$.

When $\epsilon = 1$, $\hat{f}_\epsilon^* = \hat{f}_{-z_G,ir}$. Then we perform a Newton iteration with step size 1 at $\tilde{f}_{-z_G}$,

$$\hat{f}_{-z_G,ir} - \tilde{f}_{-z_G} \approx -H_{\tilde{f}_{-z_G}}^{-1} \cdot \left( \nabla_{\hat{f}} L_{Y_{ir}}\left(\tilde{f}_{-z_G}, \hat{g}_{-z_G}\right) - \nabla_{\hat{f}} L_{Y_{ir}}\left(\tilde{f}_{-z_G}, \hat{g}\right) \right)$$

Iterate $i_r$ through set $S$, and we have

$$\hat{f}_{-z_G} - \tilde{f}_{-z_G} \approx -H_{\tilde{f}_{-z_G}}^{-1} \cdot \left( \nabla_{\hat{f}} \sum_{i \in S} L_{Y_i}\left(\tilde{f}_{-z_G}, \hat{g}_{-z_G}\right) - \nabla_{\hat{f}} \sum_{i \in S} L_{Y_i}\left(\tilde{f}_{-z_G}, \hat{g}\right) \right) \tag{37}$$

The edited version of $\hat{g}_{-z_G}$ has been deduced as $\bar{g}_{-z_G}$ in theorem F.1, substituting this approximation into equation 37, then we have

$$\hat{f}_{-z_G} - \tilde{f}_{-z_G} \approx -H_{\tilde{f}_{-z_G}}^{-1} \cdot \left( \nabla_{\hat{f}} \sum_{i \in S} L_{Y_i}\left(\tilde{f}_{-z_G}, \bar{g}_{-z_G}\right) - \nabla_{\hat{f}} \sum_{i \in S} L_{Y_i}\left(\tilde{f}_{-z_G}, \hat{g}\right) \right). \tag{38}$$

Noting that we cannot obtain $\hat{f}_{-z_G}$ and $H_{\tilde{f}_{-z_G}}$ directly because we do not retrain the label predictor but edit it to $\bar{f}_{-z_G}^*$ as a substitute. Therefore, we approximate $\hat{f}_{-z_G}$ with $\bar{f}_{-z_G}^*$ and $H_{\tilde{f}_{-z_G}}$ with $H_{\bar{f}_{-z_G}^*}$ which is defined by:

$$H_{\bar{f}_{-z_G}^*} = \nabla_{\hat{f}}^2 \sum_{i \in S} L_{Y_i}\left(\bar{f}_{-z_G}^*, \hat{g}\right)$$

Then we define $B_G$ as

$$B_G \triangleq -H_{\bar{f}_{-z_G}^*}^{-1} \cdot \left( \nabla_{\hat{f}} \sum_{i \in S} L_{Y_i}\left(\bar{f}_{-z_G}^*, \bar{g}_{-z_G}\right) - \nabla_{\hat{f}} \sum_{i \in S} L_{Y_i}\left(\bar{f}_{-z_G}^*, \hat{g}\right) \right) \tag{39}$$

Combining equation 34 and equation 39, then we deduce the final closed-form edited label predictor as

$$\bar{f}_{-z_G} = \bar{f}_{-z_G}^* + B_G = \hat{f} + A_G + B_G.$$

$\square$

## G  ALGORITHM

---

**Algorithm 1** Concept-label-level ECBM

---

1: **Input:** Dataset $\mathcal{D} = \{(x_i, y_i, c_i)\}_{i=1}^n$, original concept predictor $\hat{f}$, and label predictor $\hat{g}$, a set of erroneous data $D_e$ and its associated index set $S_e$.
2: For the index $(w, r)$ in $S_e$, correct $c_w^r$ to the right label $c_w^r{}'$ for the $w$-th data $(x_w, y_w, c_w)$.
3: Compute the Hessian matrix of the loss function respect to $\hat{g}$:

$$H_{\hat{g}} = \nabla_{\hat{g}}^2 \sum_{i,j} L_{C_j}(\hat{g}^j(x_i), c_i^j).$$

4: Update concept predictor $\tilde{g}$:

$$\tilde{g} = \hat{g} - H_{\hat{g}}^{-1} \cdot \sum_{(w,r) \in S_e} \left( \nabla_{\hat{g}} L_{C_r} \left( \hat{g}^r(x_w), c_w^r{}' \right) - \nabla_{\hat{g}} L_{C_r} \left( \hat{g}^r(x_w), c_w^r \right) \right).$$

5: Compute the Hessian matrix of the loss function respect to $\hat{f}$:

$$H_{\hat{f}} = \nabla_{\hat{f}}^2 \sum_{i=1}^n L_{Y_i}(\hat{f}, \hat{g}).$$

6: Update label predictor $\tilde{f}$:

$$\tilde{f} = \hat{f} + H_{\hat{f}}^{-1} \cdot \nabla_f \sum_{i=1}^n L_Y \left( \hat{f}\left( \hat{g}(x_i) \right), y_i \right) - H_{\hat{f}}^{-1} \cdot \nabla_f \sum_{l=1}^n \left( L_Y \left( \hat{f}\left( \tilde{g}(x_l) \right), y_l \right) \right).$$

7: **Return:** $\tilde{f}, \tilde{g}$.

---

**Algorithm 2** Concept-level ECBM

---

1: **Input:** Dataset $\mathcal{D} = \{(x_i, y_i, c_i)\}_{i=1}^n$, original concept predictor $\hat{f}$, label predictor $\hat{g}$ and the to be removed concept index set $M$.
2: For $r \in M$, set $p_r = 0$ for all the data $z \in \mathcal{D}$.
3: Compute the Hessian matrix of the loss function respect to $\hat{g}$:

$$H_{\hat{g}} = \nabla_{\hat{g}}^2 \sum_{j \notin M} \sum_{i=1}^n L_{C_j}(\hat{g}^j(x_i), c_i^j).$$

4: Update concept predictor $\tilde{g}^*$:

$$\tilde{g}^* = \hat{g} - H_{\hat{g}}^{-1} \cdot \nabla_{\hat{g}} \sum_{j \notin M} \sum_{i=1}^n L_{C_j}(\hat{g}^j(x_i), c_i^j).$$

5: Compute the Hessian matrix of the loss function respect to $\hat{f}$:

$$H_{\hat{f}} = \nabla_{\hat{f}}^2 \sum_{i=1}^n L_Y(\hat{f}(\hat{g}(x_i)), y_i).$$

6: Update label predictor $\tilde{f}$:

$$\tilde{f} = \hat{f} - H_{\hat{f}}^{-1} \cdot \nabla_{\hat{f}} \sum_{l=1}^n L_Y \left( \hat{f}\left( \tilde{g}^*(x_l) \right), y_l \right).$$

7: Map $\tilde{g}^*$ to $\tilde{g}$ by removing the $r$-th row of the matrix in the final layer of $\tilde{g}^*$ for $r \in M$.
8: **Return:** $\tilde{f}, \tilde{g}$.

---

**Algorithm 3** Data-level ECBM

1: **Input:** Dataset $\mathcal{D} = \{(x_i, y_i, c_i)\}_{i=1}^N$, original concept predictor $\hat{f}$, label predictor $\hat{g}$, and the to be removed data index set $G$.

2: For $r \in G$, remove the $r$-th data $(x_r, y_r, c_r)$ from $\mathcal{D}$ and define the new dataset as $\mathcal{S}$.

3: Compute the Hessian matrix of the loss function with respect to $\hat{g}$:

$$H_{\hat{g}} = \nabla_{\hat{g}}^2 \sum_{i,j} L_{C_j}(\hat{g}^j(x_i), c_i^j).$$

4: Update concept predictor $\tilde{g}$:

$$\tilde{g} = \hat{g} + H_{\hat{g}}^{-1} \cdot \sum_{r \in G} \nabla_g L_C(\hat{g}(x_r), c_r)$$

5: Update label predictor $\tilde{f}$. Compute the Hessian matrix of the loss function with respect to $\hat{f}$:

$$H_{\hat{f}} = \nabla_{\hat{f}}^2 \sum_{i=1}^n L_Y(\hat{f}(\hat{g}(x_i)), y_i).$$

6: Compute $A$ as:

$$A = H_{\hat{f}}^{-1} \cdot \sum_{i \in [n] - G} \nabla_{\hat{f}} L_Y\left(\hat{f}(\hat{g}(x_i)), y_i\right)$$

7: Obtain $\bar{f}$ as

$$\bar{f} = \hat{f} + A$$

8: Compute the Hessian matrix of the loss function concerning $\bar{f}$:

$$H_{\bar{f}} = \nabla_{\bar{f}}^2 \sum_{i \in [n] - G} L_Y(\bar{f}(\hat{g}(x_i)), y_i).$$

9: Compute $B$ as

$$B = -H_{\bar{f}}^{-1} \cdot \sum_{i \in [n] - G} \nabla_{\hat{f}} \left(L_Y(\bar{f}(\tilde{g}(x_i)), y_i) - L_Y(\bar{f}(\hat{g}(x_i)), y_i)\right)$$

10: Update the label predictor $\tilde{f}$ as: $\tilde{f} = \hat{f} + A + B$.

11: **Return:** $\tilde{f}, \tilde{g}$.

---

**Algorithm 4** EK-FAC for Concept Predictor $g$

1: **Input:** Dataset $\mathcal{D} = \{(x_i, y_i, c_i)\}_{i=1}^N$, original concept predictor $\hat{g}$.

2: **for** the $l$-th convolution layer of $\hat{g}$: **do**

3:     Define the input activations $\{a_{j,t}\}$, weights $W = (w_{i,j,\delta})$, and biases $b = (b_i)$ of this layer;

4:     Obtain the expanded activations $[\![A_{l-1}]\!]$ as:

$$[\![A_{l-1}]\!]_{t,j|\Delta|+\delta} = [A_{l-1}]_{(t+\delta),j} = a_{j,t+\delta},$$

5:     Compute the pre-activations:

$$[S_l]_{i,t} = s_{i,t} = \sum_{\delta \in \Delta} w_{i,j,\delta} a_{j,t+\delta} + b_i.$$

6:     During the backpropagation process, obtain the $\mathcal{D}s_{i,t}$ as:

$$\mathcal{D}s_{i,t} = \frac{\partial \sum_{j=1}^k \sum_{i=1}^n L_{C_j}}{\partial s_{i,t}}$$

7:     Compute $\hat{\Omega}_{l-1}$ and $\hat{\Gamma}_l$:

$$\hat{\Omega}_{l-1} = \frac{1}{n} \sum_{i=1}^{n} \left( [\![A_{l-1}^i]\!]_{\mathrm{H}}^{\top} [\![A_{l-1}^i]\!]_{\mathrm{H}} \right)$$

$$\hat{\Gamma}_l = \frac{1}{n} \sum_{i=1}^{n} \left( \frac{1}{|\mathcal{T}|} \mathcal{D}S_l^{i\top} \mathcal{D}S_l^i \right)$$

8:     Perform eigenvalue decomposition of $\hat{\Omega}_{l-1}$ and $\hat{\Gamma}_l$, obtain $Q_\Omega, \Lambda_\Omega, Q_\Gamma, \Lambda_\Gamma$, which satisfies

$$\hat{\Omega}_{l-1} = Q_\Omega \Lambda_\Omega Q_\Omega^{\top}$$

$$\hat{\Gamma}_l = Q_\Gamma \Lambda_\Gamma Q_\Gamma^{\top}$$

9:     Define a diagonal matrix $\Lambda$ and compute the diagonal element as

$$\Lambda_{ii}^* = n^{-1} \sum_{j=1}^{n} \left( \left( Q_{\Omega_{l-1}} \otimes Q_{\Gamma_l} \right) \nabla_{\theta_l} L_{C_j} \right)_i^2.$$

10:    Compute $\hat{H}_l^{-1}$ as

$$\hat{H}_l^{-1} = \left( Q_{\Omega_{l-1}} \otimes Q_{\Gamma_l} \right) \left( \Lambda + \lambda_l I_{d_l} \right)^{-1} \left( Q_{\Omega_{l-1}} \otimes Q_{\Gamma_l} \right)^{\mathrm{T}}$$

11: **end for**

12: Splice $H_l$ sequentially into large diagonal matrices

$$\hat{H}_{\hat{g}}^{-1} = \begin{pmatrix} \hat{H}_1^{-1} & & \mathbf{0} \\ & \ddots & \\ \mathbf{0} & & \hat{H}_d^{-1} \end{pmatrix}$$

where $d$ is the number of the convolution layer of the concept predictor.

13: **Return: the inverse Hessian matrix $\hat{H}_{\hat{g}}^{-1}$.**

---

**Algorithm 5** EK-FAC for Label Predictor $f$

---

1: **Input:** Dataset $\mathcal{D} = \{(x_i, y_i, c_i)\}_{i=1}^N$, original label predictor $\hat{f}$.

2: Denote the pre-activated output of $\hat{f}$ as $f'$, Compute $A$ as

$$A = \frac{1}{n} \cdot \sum_{i=1}^{n} \hat{g}(x_i) \cdot \hat{g}(x_i)^{\mathrm{T}}$$

3: Comput $B$ as:

$$B = \frac{1}{n} \cdot \sum_{i=1}^{n} \nabla_{f'} L_Y(\hat{f}(\hat{g}(x_i)), y_i) \cdot \nabla_{f'} L_Y(\hat{f}(\hat{g}(x_i)), y_i)^{\mathrm{T}}$$

4: Perform eigenvalue decomposition of AA and BB, obtain $Q_A, \Lambda_A, Q_B, \Lambda_B$, which satisfies

$$A = Q_A \Lambda_A Q_A^{\top}$$

$$B = Q_B \Lambda_B Q_B^{\top}$$

5: Define a diagonal matrix $\Lambda$ and compute the diagonal element as

$$\Lambda_{ii}^* = n^{-1} \sum_{j=1}^{n} \left( \left( Q_A \otimes Q_B \right) \nabla_{\hat{f}} L_{Y_j} \right)_i^2.$$

6: Compute $\hat{H}_{\hat{f}}^{-1}$ as

$$\hat{H}_{\hat{f}}^{-1} = (Q_A \otimes Q_B)(\Lambda + \lambda I_d)^{-1}(Q_A \otimes Q_B)^{\mathrm{T}}$$

7: **Return: the inverse Hessian matrix $\hat{H}_{\hat{f}}^{-1}$.**

---

**Algorithm 6** EK-FAC Concept-label-level ECBM

1: **Input:** Dataset $\mathcal{D} = \{(x_i, y_i, c_i)\}_{i=1}^N$, original concept predictor $\hat{f}$, label predictor $\hat{g}$, and the to be removed data index set $G$, and damping parameter $\lambda$.
2: For $r \in G$, remove the $r$-th data $(x_r, y_r, c_r)$ from $\mathcal{D}$ and define the new dataset as $\mathcal{S}$.
3: **Use EK-FAC method in algorithm 4 to accelerate iHVP problem for $\hat{g}$ and obtain the inverse Hessian matrix $\hat{H}_{\hat{g}}^{-1}$**
4: Update concept predictor $\tilde{g}$:

$$\tilde{g} = \hat{g} - H_{\hat{g}}^{-1} \cdot \sum_{(w,r) \in S_e} \left( \nabla_{\hat{g}} L_{C_r}\left(\hat{g}^r(x_w), c_w^r{}'\right) - \nabla_{\hat{g}} L_{C_r}\left(\hat{g}^r(x_w), c_w^r\right) \right).$$

5: **Use EK-FAC method in algorithm 5 to accelerate iHVP problem for $\hat{f}$ and obtain $\hat{H}_{\hat{f}}^{-1}$**
6: Update label predictor $\tilde{f}$:

$$\tilde{f} = \hat{f} + H_{\hat{f}}^{-1} \cdot \nabla_f \sum_{i=1}^n L_Y\left(\hat{f}\left(\hat{g}(x_i)\right), y_i\right) - H_{\hat{f}}^{-1} \cdot \nabla_f \sum_{l=1}^n \left( L_Y\left(\hat{f}\left(\tilde{g}(x_l)\right), y_l\right) \right).$$

7: **Return:** $\tilde{f}, \tilde{g}$.

---

**Algorithm 7** EK-FAC Concept-level ECBM

1: **Input:** Dataset $\mathcal{D} = \{(x_i, y_i, c_i)\}_{i=1}^n$, original concept predictor $\hat{f}$, label predictor $\hat{g}$ and the to be removed concept index set $M$, and damping parameter $\lambda$.
2: For $r \in M$, set $p_r = 0$ for all the data $z \in \mathcal{D}$.
3: **Use EK-FAC method in algorithm 4 to accelerate iHVP problem for $\hat{g}$ and obtain the inverse Hessian matrix $\hat{H}_{\hat{g}}^{-1}$**
4: Update concept predictor $\tilde{g}$:

$$\tilde{g}^* = \hat{g} - H_{\hat{g}}^{-1} \cdot \nabla_{\hat{g}} \sum_{j \notin M} \sum_{i=1}^n L_{C_j}(\hat{g}^j(x_i), c_i^j).$$

5: **Use EK-FAC method in algorithm 5 to accelerate iHVP problem for $\hat{f}$ and obtain $\hat{H}_{\hat{f}}^{-1}$**
6: Update label predictor $\tilde{f}$:

$$\tilde{f} = \hat{f} - H_{\hat{f}}^{-1} \cdot \nabla_{\hat{f}} \sum_{l=1}^n L_Y\left(\hat{f}\left(\tilde{g}^*(x_l)\right), y_l\right).$$

7: Map $\tilde{g}^*$ to $\tilde{g}$ by removing the $r$-th row of the matrix in the final layer of $\tilde{g}^*$ for $r \in M$.
8: **Return:** $\tilde{f}, \tilde{g}$.

---

**Algorithm 8** EK-FAC Data-level ECBM

1: **Input:** Dataset $\mathcal{D} = \{(x_i, y_i, c_i)\}_{i=1}^n$, original concept predictor $\hat{f}$, and label predictor $\hat{g}$, a set of erroneous data $D_e$ and its associated index set $S_e$, and damping parameter $\lambda$.
2: For the index $(w, r)$ in $S_e$, correct $c_w^r$ to the right label $c_w^r{}'$ for the $w$-th data $(x_w, y_w, c_w)$.
3: **Use EK-FAC method in algorithm 4 to accelerate iHVP problem for $\hat{g}$ and obtain the inverse Hessian matrix $\hat{H}_{\hat{g}}^{-1}$**
4: Update concept predictor $\tilde{g}$:

$$\tilde{g} = \hat{g} - H_{\hat{g}}^{-1} \cdot \sum_{(w,r) \in S_e} \left( \nabla_{\hat{g}} L_{C_r}\left(\hat{g}^r(x_w), c_w^r{}'\right) - \nabla_{\hat{g}} L_{C_r}\left(\hat{g}^r(x_w), c_w^r\right) \right).$$

5: **Use EK-FAC method in algorithm 5 to accelerate iHVP problem for $\hat{f}$ and obtain $H_{\hat{f}}^{-1}$**
   Compute $A$ as:

$$A = H_{\hat{f}}^{-1} \cdot \sum_{i \in [n]-G} \nabla_{\hat{f}} L_Y \left( \hat{f}(\hat{g}(x_i)), y_i \right)$$

   Obtain $\bar{f}$ as

$$\bar{f} = \hat{f} + A$$

6: **Use EK-FAC method in algorithm 5 to accelerate iHVP problem for $\bar{f}$ and obtain $H_{\bar{f}}^{-1}$**
   Compute $B'$ as

$$B' = -H_{\bar{f}}^{-1} \cdot \sum_{i \in [n]-G} \nabla_{\hat{f}} \left( L_Y(\bar{f}(\tilde{g}(x_i)), y_i) - L_Y(\bar{f}(\hat{g}(x_i)), y_i) \right)$$

   Update the label predictor $\tilde{f}$ as: $\tilde{f} = \hat{f} + A + B'$.
7: **Return:** $\tilde{f}, \tilde{g}$.

## H    ADDITIONAL EXPERIMENTS

### H.1    EXPERIMENTAL SETTING

**Methodology for Processing CUB Dataset** For CUB dataset, we follow the setting in  Koh et al. (2020). We aggregate instance-level concept annotations into class-level concepts via majority voting: e.g., if more than 50% of crows have black wings in the data, then we set all crows to have black wings.

#### H.1.1    REVISED TIMING METHOD IN TABLE 1

We present another version of the runtime calculation method for Table 1. Note that the timing method in Table 1 is based on the average time taken for each data point used during the update process. In this version, we present another timing method, where RT represents the total runtime for the updating.  According to Table 3, when the timing method is changed to total runtime, the speed

Table 3: Performance comparison of different methods on the three datasets.

| Edit Level | Method | OAI | | CUB | | CelebA | |
|---|---|---|---|---|---|---|---|
| | | **F1 score** | **RT (minute)** | **F1 score** | **RT (minute)** | **F1 score** | **RT (minute)** |
| Concept Label | Retrain | 0.8825±0.0054 | 297.77 | 0.7971±0.0066 | 85.56 | 0.3827±0.0272 | 304.71 |
| | CBM-IF(Ours) | 0.8639±0.0033 | 4.63 | 0.7699±0.0035 | 1.33 | 0.3561±0.0134 | 5.54 |
| | ECBM(Ours) | **0.8808±0.0039** | **2.36** | **0.7963±0.0050** | **0.65** | **0.3845±0.0327** | **2.49** |
| Concept | Retrain | 0.8448±0.0191 | 258.84 | 0.7811±0.0047 | 87.21 | 0.3776±0.0350 | 355.85 |
| | CBM-IF(Ours) | 0.8214±0.0071 | 4.94 | 0.7579±0.0065 | 1.45 | 0.3609±0.0202 | 5.51 |
| | ECBM(Ours) | **0.8403±0.0090** | **2.36** | **0.7787±0.0058** | **0.59** | **0.3761±0.0280** | **2.48** |
| Data | Retrain | 0.8811±0.0065 | 319.37 | 0.7838±0.0051 | 86.20 | 0.3797±0.0375 | 325.62 |
| | CBM-IF(Ours) | 0.8472±0.0046 | 5.07 | 0.7623±0.0031 | 1.46 | 0.3536±0.0166 | 5.97 |
| | ECBM(Ours) | **0.8797±0.0038** | **2.50** | **0.7827±0.0088** | **0.65** | **0.3748±0.0347** | **2.49** |

of ECBM far exceeds that of retrain, being approximately 200 times faster. Even on the CelebA dataset with 202,599 data points, ECBM can update the model in less than 3 minutes, with an F1 score deviation of only 0.0018 compared to retraining.

**RMIA score.** The RMIA score is computed as:

$$LR_\theta(x, z) \approx \frac{\Pr(f_\theta(x)|\mathcal{N}(\mu_{x,\bar{z}}(x), \sigma_{x,\bar{z}}^2(x)))}{\Pr(f_\theta(x)|\mathcal{N}(\mu_{\bar{x},z}(x), \sigma_{\bar{x},z}^2(x)))} \times \frac{\Pr(f_\theta(z)|\mathcal{N}(\mu_{x,\bar{z}}(z), \sigma_{x,\bar{z}}^2(z)))}{\Pr(f_\theta(z)|\mathcal{N}(\mu_{\bar{x},z}(z), \sigma_{\bar{x},z}^2(z)))}$$

where $f_\theta(x)$ represents the model's output (logits) for the data point $x$, $\mathcal{N}(\mu, \sigma^2)$ denotes a Gaussian distribution with mean $\mu$ and variance $\sigma^2$, $\mu_{x,\bar{z}}(x)$ is the mean of the model's outputs for $x$ under

the assumption that $x$ belongs to the training set, and $\sigma_{x,\bar{z}}^2(x)$ is the variance of the model's outputs for $x$. The likelihoods $\Pr(f_\theta(x)|\mathcal{N})$ represent the probability that the model's output $f_\theta(x)$ follows the Gaussian distribution parameterized by $\mu$ and $\sigma^2$, under the two different hypotheses: $x$ being a member of the training set versus not being a member.

## H.2 IMPROVEMENT VIA HARMFUL DATA REMOVAL

We conducted addition experiments on CUB datasets with synthetically introduced noisy concepts or labels. Firstly, we introduce noises under three levels. In concept level, we choose 10% of the concepts and flip these concept labels for a portion of the data. In data level, we choose 10% of the data and flip their labels. In concept-label level, we choose 10% of the total concepts and flip them. Then we conduct the following experiments. We introduce noises into the three levels and train the model. After that, we remove the noise and obtain the retrained mdoel, which is the ground truth(gt) of this harmful data removal task. In contrast, we use ECBM to remove the harmful data.

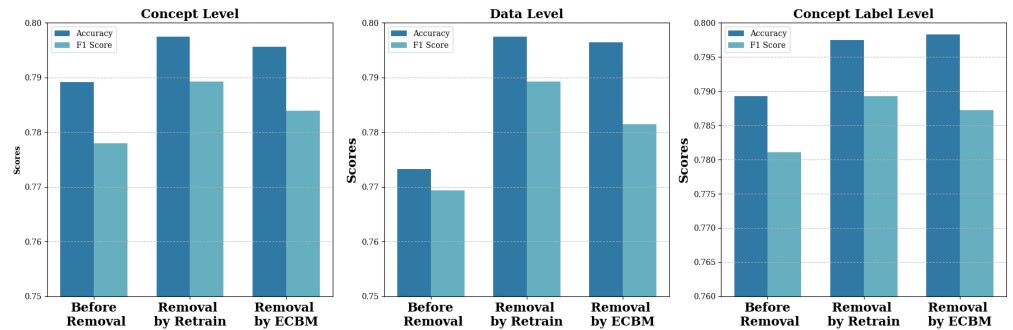

Figure 6: Model performance after the removal of harmful data.

From Figure 6, it can be observed that the model performance improves across all three settings after noise removal and subsequent retraining or ECBM editing. This confirms that the performance of ECBM is nearly equivalent to retraining in various experimental scenarios, further providing evidence of the robustness of our method.

## H.3 PERIODIC EDITING PERFORMANCE

ECBM can perform periodic editing. To evalutae the multiple editing performance of ECBM, we conduct the following experiments. Firstly, we introduce noises under three levels. In concept level, we choose 10% of the concepts and flip these concept labels for a portion of the data. In data level, we choose 10% of the data and flip their labels. In concept-label level, we choose 10% of the total concepts and flip them. Then we conduct the following experiments.

In the concept level, we firstly remove 1% of the concepts, then retrain or use ECBM to edit and repeat. In the data level, we firstly remove 1% of the data, then retrain or use ECBM to edit. In the concept label level, we firstly remove one concept label from 1% of the data, then retrain or use ECBM to edit. Note that when remove the next 1% of the concepts, ECBM edit the model based on the last editing result. The results at each level are shown in Figure 7, 8 and 9.

From the above three levels, we can find that with the mislabeled information removed, the retrained model achieves better performance in both accuracy and F1 score than the initial model. Furthermore, the performance of the ECBM-edited model is similar to that of the retrained model, even after 10 rounds editing, which demonstrates the ability of our ECBM method to handle multiple edits.

## H.4 MORE VISUALIZATION RESULTS AND EXPLANATION

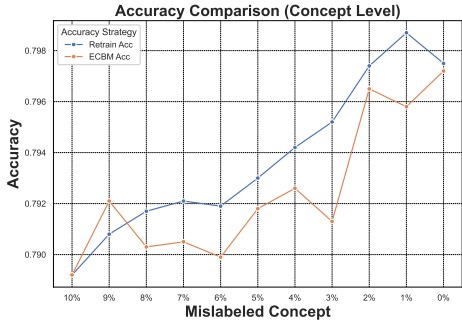 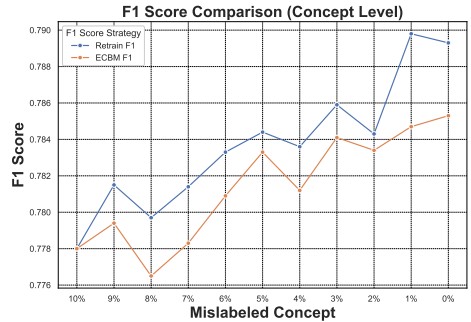

(a) The accuracy of the edited model compared with retrained.

(b) The F1 score of the edited model compared with retrained.

Figure 7: Accuracy and F1 score difference of the edited model compared with retrained at concept level.

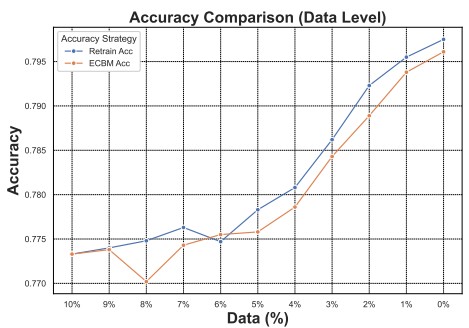 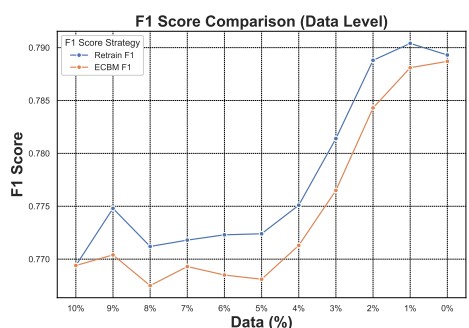

(a) The accuracy of the edited model compared with retrained.

(b) The F1 score of the edited model compared with retrained.

Figure 8: Accuracy and F1 score difference of the edited model compared with retrained at data level.

### H.4.1 EXPLANATION FOR VISUALIZATION RESULTS

At the concept level, we remove each concept one at a time and retrain the CBM, and subsequently evaluate the model performance. We rank the concepts in descending order based on the model performance loss. Concepts that, when removed, cause significant changes in model performance are considered influential concepts. The top 10 concepts are shown in the retrain column as illustrated in Figure 5. In contrast, we use our ECBM method instead of the retrain method, as outlined in Algorithm 7, and the top 10 concepts are shown in the ECBM column of Figure 5.

To help readers connect the top 10 influential concepts with the input image, we provide visualizations of the data and list the concept labels corresponding to the top 10 influential concepts, which are shown in Figure 5,10, 11.

For the other two levels and for additional datasets, we also conduct a similar procedure, and the corresponding visualization results are presented in Figure 12, 13, 14, 15, and 16.

### H.4.2 VISUALIZATION RESULTS

We provide our additional visualization results in Figure 10, 11, 12, 13, 14, 15, and 16.

## I MORE RELATED WORK

**Influence Function.** The influence function, initially a staple in robust statistics Cook (2000); Cook & Weisberg (1980), has seen extensive adoption within machine learning since Koh & Liang (2017) introduced it to the field. Its versatility spans various applications, including detecting mislabeled data, interpreting models, addressing model bias, and facilitating machine unlearning tasks. Notable

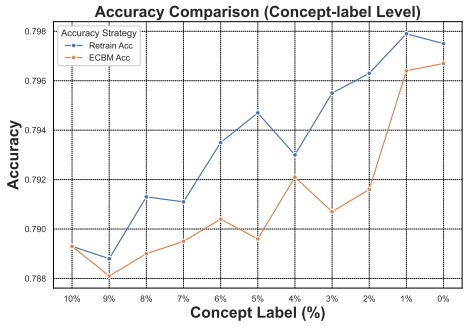 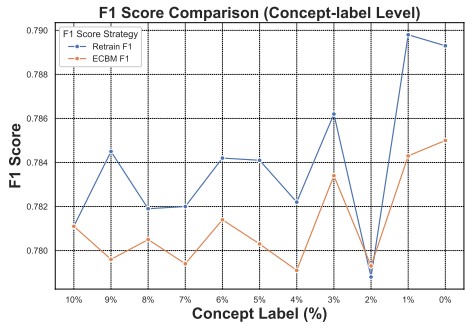

(a) The accuracy of the edited model compared with retrained.

(b) The F1 score of the edited model compared with retrained.

Figure 9: Accuracy and F1 score difference of the edited model compared with retrained at concept-label level.

works in machine unlearning encompass unlearning features and labels Warnecke et al. (2023), minimax unlearning Liu et al. (2024), forgetting a subset of image data for training deep neural networks Golatkar et al. (2020a; 2021), graph unlearning involving nodes, edges, and features. Recent advancements, such as the LiSSA method Agarwal et al. (2017); Kwon et al. (2023) and kNN-based techniques Guo et al. (2021), have been proposed to enhance computational efficiency. Besides, various studies have applied influence functions to interpret models across different domains, including natural language processing Han et al. (2020) and image classification Basu et al. (2021), while also addressing biases in classification models Wang et al. (2019), word embeddings Brunet et al. (2019), and finetuned models Chen et al. (2020). Despite numerous studies on influence functions, we are the first to utilize them to construct the editable CBM. Moreover, compared to traditional neural networks, CBMs are more complicated in their influence function. Because we only need to change the predicted output in the traditional influence function. While in CBMs, we should first remove the true concept, then we need to approximate the predicted concept in order to approximate the output. Bridging the gap between the true and predicted concepts poses a significant theoretical challenge in our proof.

**Model Unlearning.** Model unlearning has gained significant attention in recent years, with various methods (Bourtoule et al., 2021; Brophy & Lowd, 2021; Cao & Yang, 2015; Chen et al., 2022a;b) proposed to efficiently remove the influence of certain data from trained machine learning models. Existing approaches can be broadly categorized into exact and approximate unlearning methods. Exact unlearning methods aim to replicate the results of retraining by selectively updating only a portion of the dataset, thereby avoiding the computational expense of retraining on the entire dataset (Sekhari et al., 2021; Chowdhury et al., 2024). Approximate unlearning methods, on the other hand, seek to adjust model parameters to approximately satisfy the optimality condition of the objective function on the remaining data (Golatkar et al., 2020a; Guo et al., 2019; Izzo et al., 2021). These methods are further divided into three subcategories: (1) Newton step-based updates that leverage Hessian-related terms [22, 26, 31, 34, 40, 43, 49], often incorporating Gaussian noise to mitigate residual data influence. To reduce computational costs, some works approximate the Hessian using the Fisher information matrix (Golatkar et al., 2020a) or small Hessian blocks (Mehta et al., 2022). (2) Neural tangent kernel (NTK)-based unlearning approximates training as a linear process, either by treating it as a single linear change (Golatkar et al., 2020b). (3) SGD path tracking methods, such as DeltaGrad (Wu et al., 2020) and unrollSGD (Thudi et al., 2022), reverse the optimization trajectory of stochastic gradient descent during training. Despite their advancements, these methods fail to handle the special architecture of CBMs. Moreover, given the high cost of obtaining data, we sometimes prefer to correct the data rather than remove it, which model unlearning is unable to achieve.

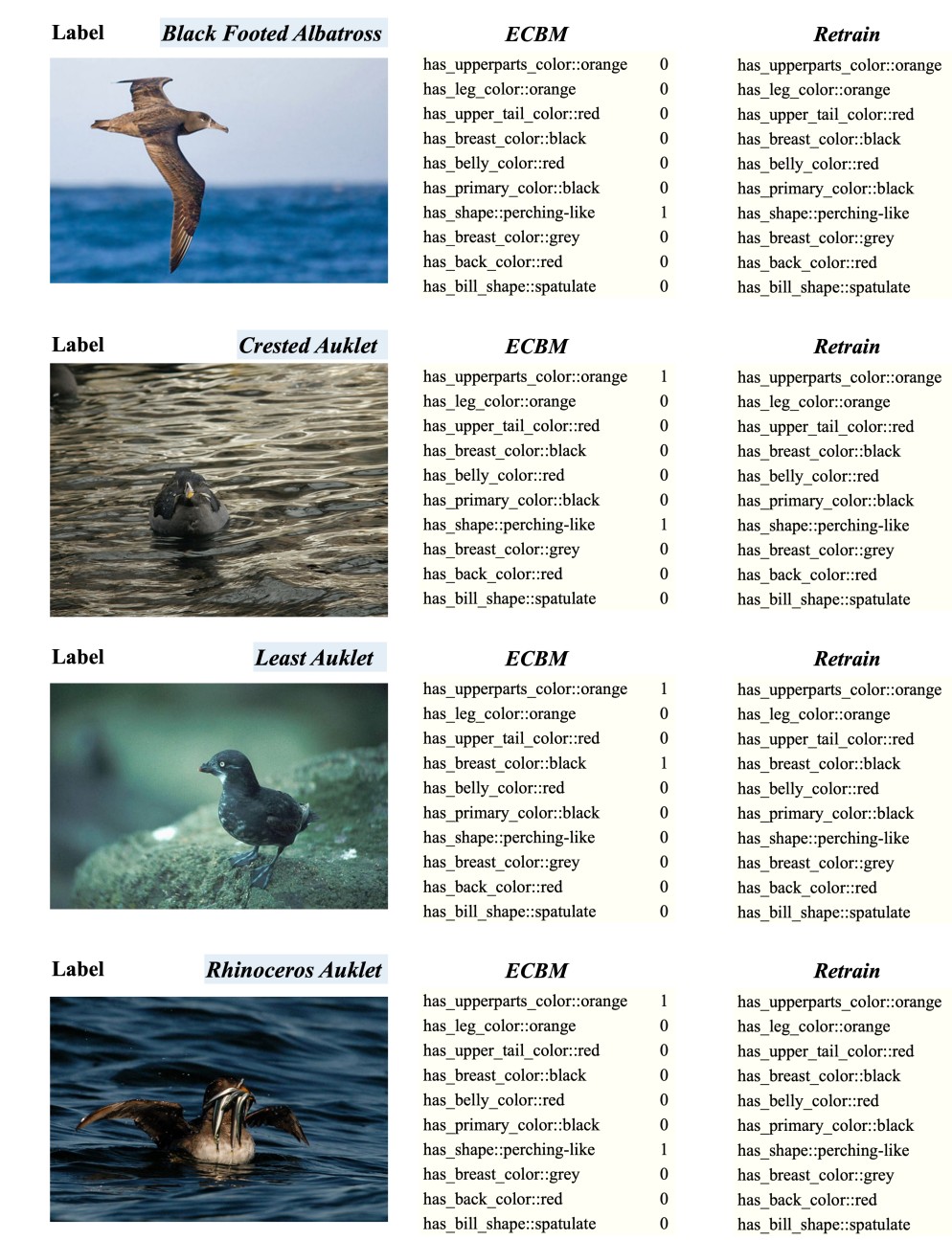

**Label** — *Black Footed Albatross*

| ECBM | | Retrain |
|---|---|---|
| has_upperparts_color::orange | 0 | has_upperparts_color::orange |
| has_leg_color::orange | 0 | has_leg_color::orange |
| has_upper_tail_color::red | 0 | has_upper_tail_color::red |
| has_breast_color::black | 0 | has_breast_color::black |
| has_belly_color::red | 0 | has_belly_color::red |
| has_primary_color::black | 0 | has_primary_color::black |
| has_shape::perching-like | 1 | has_shape::perching-like |
| has_breast_color::grey | 0 | has_breast_color::grey |
| has_back_color::red | 0 | has_back_color::red |
| has_bill_shape::spatulate | 0 | has_bill_shape::spatulate |

**Label** — *Crested Auklet*

| ECBM | | Retrain |
|---|---|---|
| has_upperparts_color::orange | 1 | has_upperparts_color::orange |
| has_leg_color::orange | 0 | has_leg_color::orange |
| has_upper_tail_color::red | 0 | has_upper_tail_color::red |
| has_breast_color::black | 0 | has_breast_color::black |
| has_belly_color::red | 0 | has_belly_color::red |
| has_primary_color::black | 0 | has_primary_color::black |
| has_shape::perching-like | 1 | has_shape::perching-like |
| has_breast_color::grey | 0 | has_breast_color::grey |
| has_back_color::red | 0 | has_back_color::red |
| has_bill_shape::spatulate | 0 | has_bill_shape::spatulate |

**Label** — *Least Auklet*

| ECBM | | Retrain |
|---|---|---|
| has_upperparts_color::orange | 1 | has_upperparts_color::orange |
| has_leg_color::orange | 0 | has_leg_color::orange |
| has_upper_tail_color::red | 0 | has_upper_tail_color::red |
| has_breast_color::black | 1 | has_breast_color::black |
| has_belly_color::red | 0 | has_belly_color::red |
| has_primary_color::black | 0 | has_primary_color::black |
| has_shape::perching-like | 0 | has_shape::perching-like |
| has_breast_color::grey | 0 | has_breast_color::grey |
| has_back_color::red | 0 | has_back_color::red |
| has_bill_shape::spatulate | 0 | has_bill_shape::spatulate |

**Label** — *Rhinoceros Auklet*

| ECBM | | Retrain |
|---|---|---|
| has_upperparts_color::orange | 1 | has_upperparts_color::orange |
| has_leg_color::orange | 0 | has_leg_color::orange |
| has_upper_tail_color::red | 0 | has_upper_tail_color::red |
| has_breast_color::black | 0 | has_breast_color::black |
| has_belly_color::red | 0 | has_belly_color::red |
| has_primary_color::black | 0 | has_primary_color::black |
| has_shape::perching-like | 1 | has_shape::perching-like |
| has_breast_color::grey | 0 | has_breast_color::grey |
| has_back_color::red | 0 | has_back_color::red |
| has_bill_shape::spatulate | 0 | has_bill_shape::spatulate |

Figure 10: Visualization of the top-10 most influential concepts for different classes in CUB.

## J    LIMITATIONS AND BROADER IMPACTS

It is important to acknowledge that the ECBM approach is essentially an approximation of the model that would be obtained by retraining with the edited data. However, results indicate that this approximation is effective in real-world applications. Concept Bottleneck Models (CBMs) have garnered much attention for their ability to elucidate the prediction process through a human-understandable concept layer. However, most previous studies focused on cases where the data, including concepts, are clean. In many scenarios, we always need to remove/insert some training data or new concepts from trained CBMs due to different reasons, such as data mislabeling, spurious concepts, and concept annotation errors. Thus, the challenge of deriving efficient editable CBMs

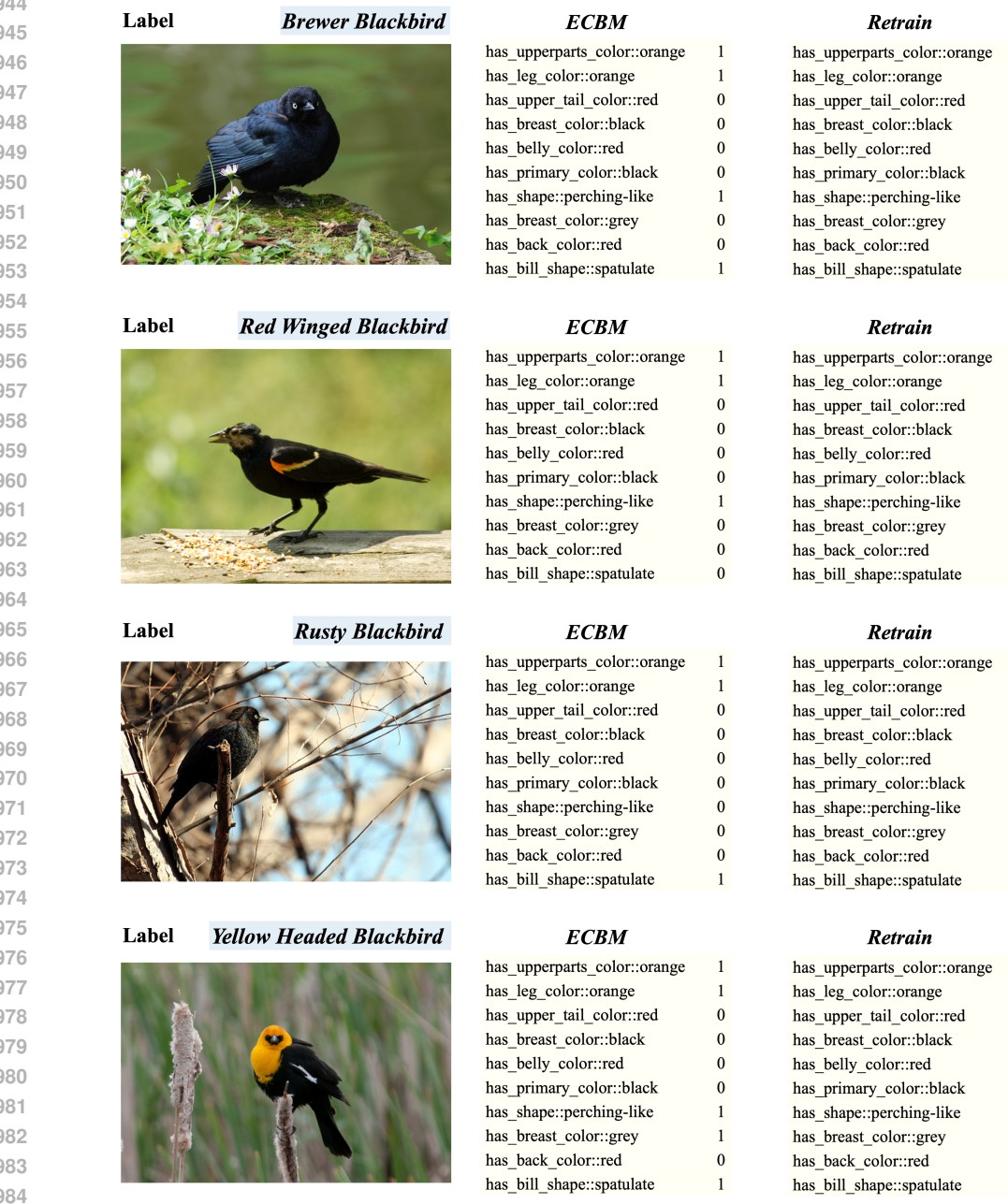

**Label** — *Brewer Blackbird*

| ECBM | | Retrain |
|---|---|---|
| has_upperparts_color::orange | 1 | has_upperparts_color::orange |
| has_leg_color::orange | 1 | has_leg_color::orange |
| has_upper_tail_color::red | 0 | has_upper_tail_color::red |
| has_breast_color::black | 0 | has_breast_color::black |
| has_belly_color::red | 0 | has_belly_color::red |
| has_primary_color::black | 0 | has_primary_color::black |
| has_shape::perching-like | 1 | has_shape::perching-like |
| has_breast_color::grey | 0 | has_breast_color::grey |
| has_back_color::red | 0 | has_back_color::red |
| has_bill_shape::spatulate | 1 | has_bill_shape::spatulate |

**Label** — *Red Winged Blackbird*

| ECBM | | Retrain |
|---|---|---|
| has_upperparts_color::orange | 1 | has_upperparts_color::orange |
| has_leg_color::orange | 1 | has_leg_color::orange |
| has_upper_tail_color::red | 0 | has_upper_tail_color::red |
| has_breast_color::black | 0 | has_breast_color::black |
| has_belly_color::red | 0 | has_belly_color::red |
| has_primary_color::black | 0 | has_primary_color::black |
| has_shape::perching-like | 1 | has_shape::perching-like |
| has_breast_color::grey | 0 | has_breast_color::grey |
| has_back_color::red | 0 | has_back_color::red |
| has_bill_shape::spatulate | 0 | has_bill_shape::spatulate |

**Label** — *Rusty Blackbird*

| ECBM | | Retrain |
|---|---|---|
| has_upperparts_color::orange | 1 | has_upperparts_color::orange |
| has_leg_color::orange | 1 | has_leg_color::orange |
| has_upper_tail_color::red | 0 | has_upper_tail_color::red |
| has_breast_color::black | 0 | has_breast_color::black |
| has_belly_color::red | 0 | has_belly_color::red |
| has_primary_color::black | 0 | has_primary_color::black |
| has_shape::perching-like | 0 | has_shape::perching-like |
| has_breast_color::grey | 0 | has_breast_color::grey |
| has_back_color::red | 0 | has_back_color::red |
| has_bill_shape::spatulate | 1 | has_bill_shape::spatulate |

**Label** — *Yellow Headed Blackbird*

| ECBM | | Retrain |
|---|---|---|
| has_upperparts_color::orange | 1 | has_upperparts_color::orange |
| has_leg_color::orange | 1 | has_leg_color::orange |
| has_upper_tail_color::red | 0 | has_upper_tail_color::red |
| has_breast_color::black | 0 | has_breast_color::black |
| has_belly_color::red | 0 | has_belly_color::red |
| has_primary_color::black | 0 | has_primary_color::black |
| has_shape::perching-like | 1 | has_shape::perching-like |
| has_breast_color::grey | 1 | has_breast_color::grey |
| has_back_color::red | 0 | has_back_color::red |
| has_bill_shape::spatulate | 1 | has_bill_shape::spatulate |

Figure 11: Visualization of the top-10 most influential concepts for different classes in CUB.

without retraining from scratch persists, particularly in large-scale applications. To address these challenges, we propose Editable Concept Bottleneck Models (ECBMs). Specifically, ECBMs support three different levels of data removal: concept-label-level, concept-level, and data-level. ECBMs enjoy mathematically rigorous closed-form approximations derived from influence functions that obviate the need for re-training. Experimental results demonstrate the efficiency and effectiveness of our ECBMs, affirming their adaptability within the realm of CBMs. Our ECBM can be an interactive model with doctors in the real world, which is an editable explanation tool.

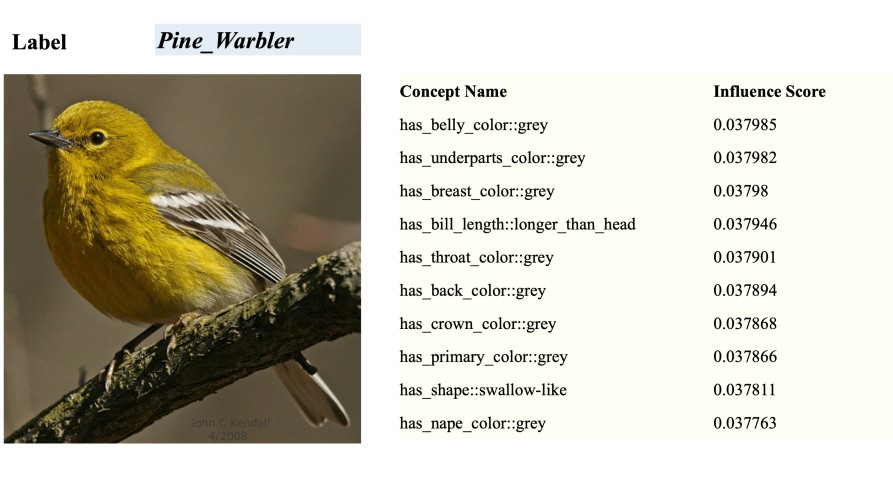

**Label** *Pine_Warbler*

| Concept Name | Influence Score |
| --- | --- |
| has_belly_color::grey | 0.037985 |
| has_underparts_color::grey | 0.037982 |
| has_breast_color::grey | 0.03798 |
| has_bill_length::longer_than_head | 0.037946 |
| has_throat_color::grey | 0.037901 |
| has_back_color::grey | 0.037894 |
| has_crown_color::grey | 0.037868 |
| has_primary_color::grey | 0.037866 |
| has_shape::swallow-like | 0.037811 |
| has_nape_color::grey | 0.037763 |

**Label** *Bewick_Wren*

| Concept Name | Influence Score |
| --- | --- |
| has_wing_color::blue | 0.04231 |
| has_crown_color::blue | 0.042196 |
| has_forehead_color::blue | 0.042055 |
| has_bill_shape::spatulate | 0.041994 |
| has_under_tail_color::blue | 0.041622 |
| has_head_pattern::unique_pattern | 0.041412 |
| has_upper_tail_color::blue | 0.041179 |
| has_nape_color::blue | 0.040844 |
| has_shape::swallow-like | 0.040686 |
| has_tail_pattern::spotted | 0.040507 |

**Label** *Song_Sparrow*

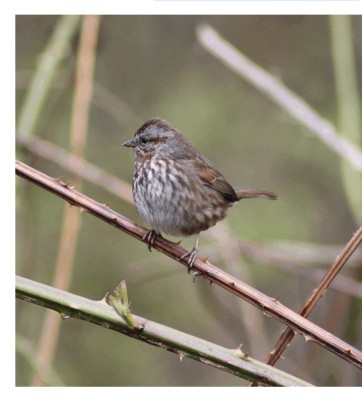

| Concept Name | Influence Score |
| --- | --- |
| has_upperparts_color::blue | 0.036309 |
| has_wing_color::blue | 0.036304 |
| has_primary_color::blue | 0.036271 |
| has_back_color::blue | 0.036261 |
| has_crown_color::blue | 0.036219 |
| has_breast_color::blue | 0.036178 |
| has_underparts_color::blue | 0.03616 |
| has_nape_color::blue | 0.036104 |
| has_upper_tail_color::blue | 0.036083 |
| has_forehead_color::blue | 0.035959 |

Figure 12: Visualization of the most influential concept label related to different data in CUB.

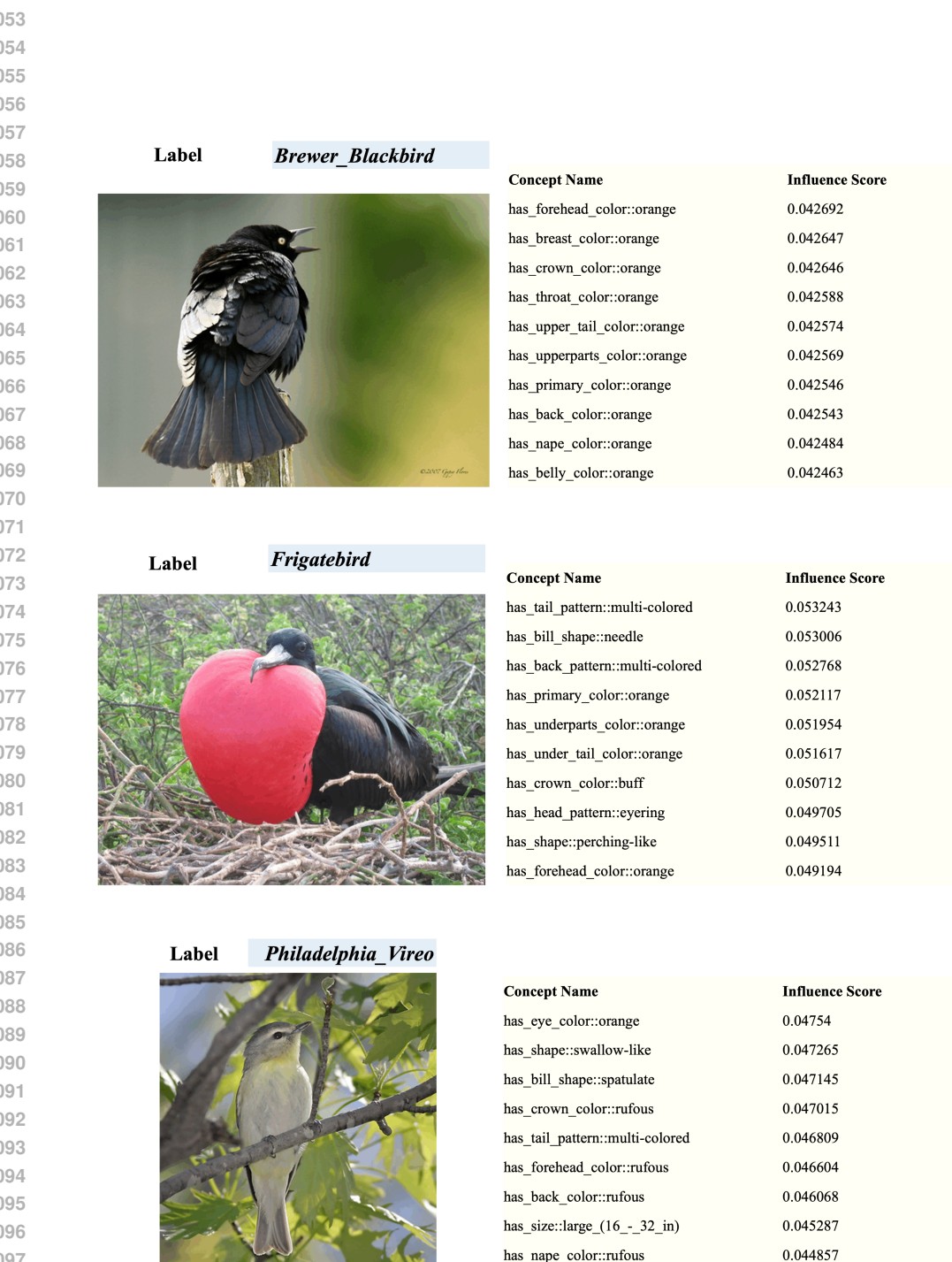

Figure 13: Visualization of the most influential concept label related to different data in CUB.

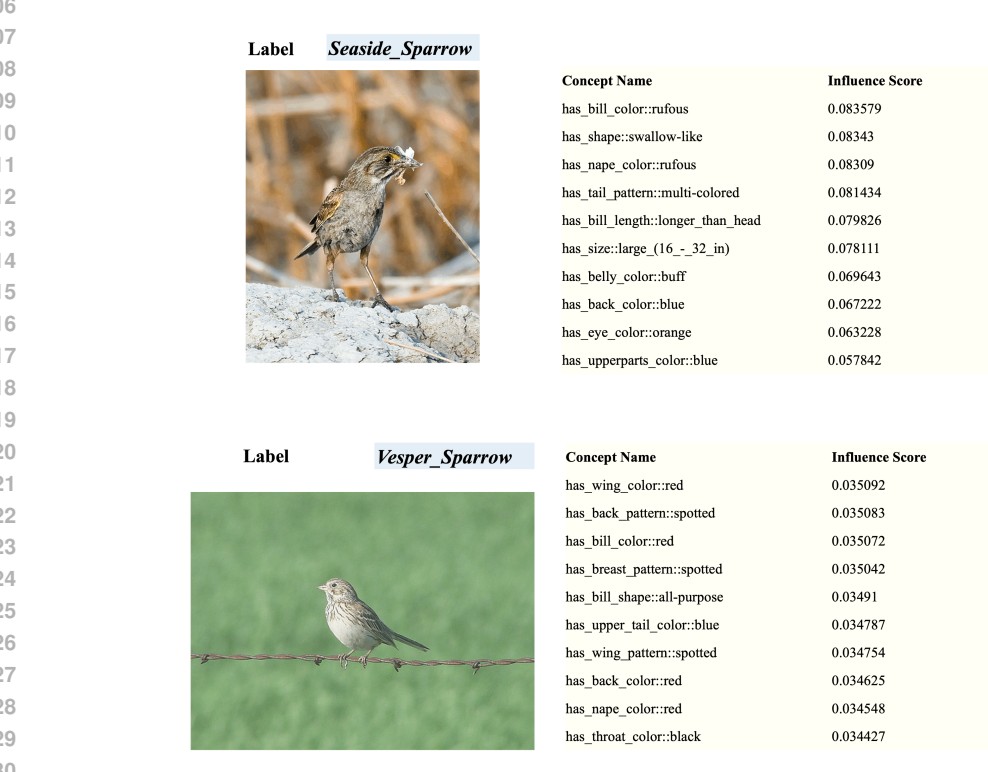

| Concept Name | Influence Score |
|---|---|
| has_bill_color::rufous | 0.083579 |
| has_shape::swallow-like | 0.08343 |
| has_nape_color::rufous | 0.08309 |
| has_tail_pattern::multi-colored | 0.081434 |
| has_bill_length::longer_than_head | 0.079826 |
| has_size::large_(16_-_32_in) | 0.078111 |
| has_belly_color::buff | 0.069643 |
| has_back_color::blue | 0.067222 |
| has_eye_color::orange | 0.063228 |
| has_upperparts_color::blue | 0.057842 |

| Concept Name | Influence Score |
|---|---|
| has_wing_color::red | 0.035092 |
| has_back_pattern::spotted | 0.035083 |
| has_bill_color::red | 0.035072 |
| has_breast_pattern::spotted | 0.035042 |
| has_bill_shape::all-purpose | 0.03491 |
| has_upper_tail_color::blue | 0.034787 |
| has_wing_pattern::spotted | 0.034754 |
| has_back_color::red | 0.034625 |
| has_nape_color::red | 0.034548 |
| has_throat_color::black | 0.034427 |

Figure 14: Visualization of the most influential concept label related to different data in CUB.

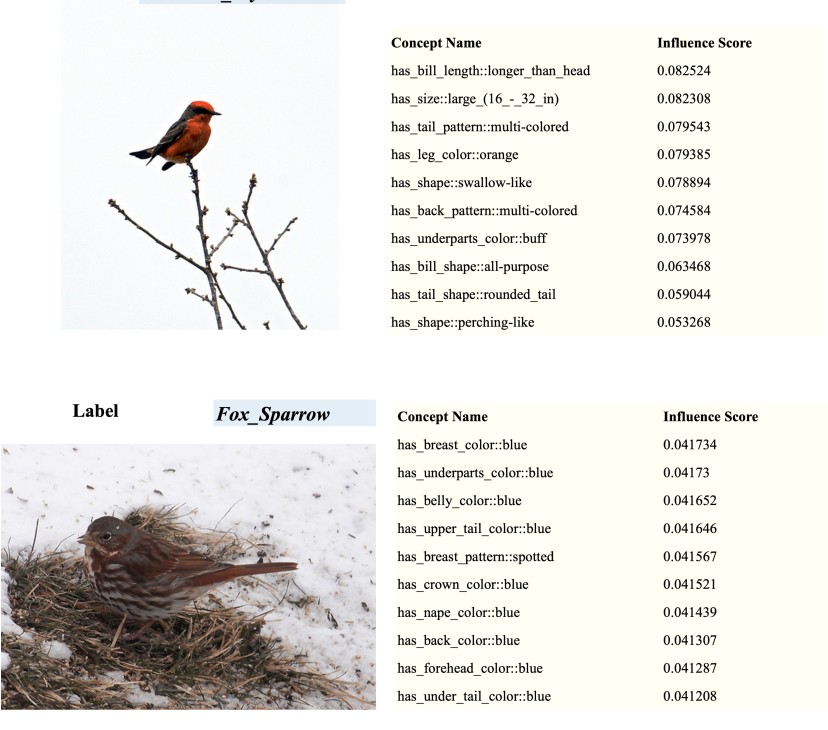

| Concept Name | Influence Score |
|---|---|
| has_bill_length::longer_than_head | 0.082524 |
| has_size::large_(16_-_32_in) | 0.082308 |
| has_tail_pattern::multi-colored | 0.079543 |
| has_leg_color::orange | 0.079385 |
| has_shape::swallow-like | 0.078894 |
| has_back_pattern::multi-colored | 0.074584 |
| has_underparts_color::buff | 0.073978 |
| has_bill_shape::all-purpose | 0.063468 |
| has_tail_shape::rounded_tail | 0.059044 |
| has_shape::perching-like | 0.053268 |

| Concept Name | Influence Score |
|---|---|
| has_breast_color::blue | 0.041734 |
| has_underparts_color::blue | 0.04173 |
| has_belly_color::blue | 0.041652 |
| has_upper_tail_color::blue | 0.041646 |
| has_breast_pattern::spotted | 0.041567 |
| has_crown_color::blue | 0.041521 |
| has_nape_color::blue | 0.041439 |
| has_back_color::blue | 0.041307 |
| has_forehead_color::blue | 0.041287 |
| has_under_tail_color::blue | 0.041208 |

Figure 15: Visualization of the most influential concept label related to different data in CUB.

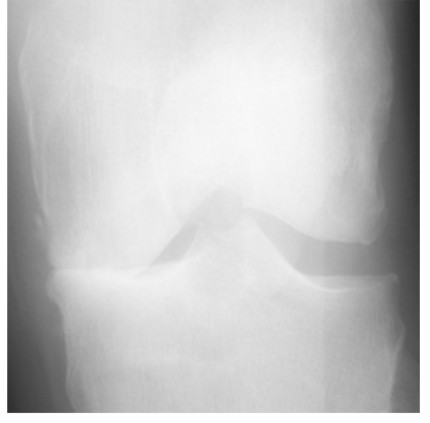

| Concept Name | Influence Score |
|---|---|
| Joint space narrowing | 0.3358 |
| Joint space narrowing lateral | 0.1622 |
| Sclerosis femur medial | 0.1161 |
| Sclerosis femur lateral | 0.0993 |
| Sclerosis tibia lateral | 0.0878 |
| Osteophytes tibia medial | 0.0724 |
| Osteophytes femur lateral | 0.047 |
| Osteophytes tibia lateral | 0.031 |
| Osteophytes femur medial | 0.0271 |
| Sclerosis tibia medial | 0.0213 |

| Concept Name | Influence Score |
|---|---|
| Joint space narrowing | 0.3506 |
| Osteophytes femur medial | 0.1698 |
| Osteophytes tibia medial | 0.0991 |
| Osteophytes tibia lateral | 0.0824 |
| Joint space narrowing lateral | 0.0728 |
| Sclerosis tibia lateral | 0.0674 |
| Osteophytes femur lateral | 0.0595 |
| Sclerosis femur lateral | 0.0467 |
| Sclerosis femur medial | 0.0272 |
| Sclerosis tibia medial | 0.0245 |

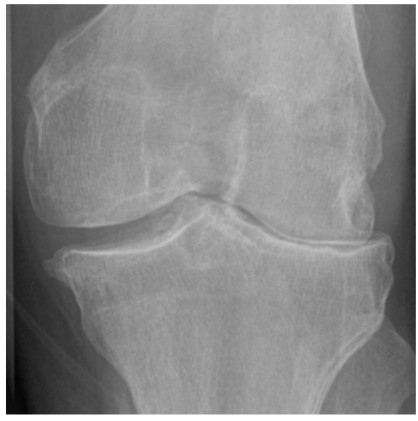

| Concept Name | Influence Score |
|---|---|
| Joint space narrowing | 0.2978 |
| Joint space narrowing lateral | 0.2018 |
| Osteophytes femur lateral | 0.1247 |
| Sclerosis tibia lateral | 0.0949 |
| Sclerosis tibia medial | 0.0892 |
| Osteophytes femur medial | 0.055 |
| Sclerosis femur medial | 0.0463 |
| Osteophytes tibia medial | 0.0387 |
| Sclerosis femur lateral | 0.0321 |
| Osteophytes tibia lateral | 0.0195 |

Figure 16: Visualization of the most influential concept label related to different data in OAI.

