# OpenReview forum: "Editable Concept Bottleneck Models"
_ICLR.cc/2025/Conference — Submitted to ICLR 2025_

### Official Review · Reviewer_AK63 · 2024-11-03

**Soundness:** 3
**Presentation:** 3
**Contribution:** 3
**Rating:** 6
**Confidence:** 2

**Summary:**

This paper delineates 3 levels of editable settings which are concept-label-level, concept-level and data-level. Based on the three settings, the author proposes a concept bottleneck model (CBM) to remove the data or concept influence without retraining based on influence function. Furthermore, the author proposes Editable Concept Bottleneck Model (ECBM) and the streamlined version EK-FAC. Moreover, the author provides the mathematical proof for the three settings and the methods. The extensive experiments shows the effectiveness of the method.

**Strengths:**

- The notation and problem definition is clear.
- The paper is easy to follow.
- The effect of influence function in Section 5.3 is insightful.
- The result visualization is clear.

**Weaknesses:**

- The baseline methods are not complete. For example, there are still some methods for EBM like, probability EBM [1], etc.
- The influence function is only suitable for convex function. I wonder how the author applies it on the non-convex function, like deep neural network.
- The approximate equal sign is mixed with the equal sign, which is not rigorous, like in Theorem 4.7. Please correct me if I have some mis-understanding.


[1] Kim, Eunji, et al. "Probabilistic concept bottleneck models." arXiv preprint arXiv:2306.01574 (2023).

**Questions:**

- I am doubtful with the running time in Table 1. How the running time can be less 1 minute even for retraining method.

---

> ### Author Response · Authors · 2024-11-24
> **Response to Reviewer AK63**
>
> ### Weaknesses:
> >The baseline methods are not complete. For example, there are still some methods for EBM like, probability EBM [1], etc.
>
> Our paper proposes a CBM editing method. Probability EBM is not used as a baseline for our method, as it does not align with the primary objectives of our approach.
>
> >The influence function is only suitable for convex function. I wonder how the author applies it on the non-convex function, like deep neural network.
>
> We disagree. While the classical influence function is most straightforwardly applied to convex functions, its principles can be adapted to non-convex functions through regularization techniques. IF has been widely studied for deep neural networks, starting from [5].  Moreover, the influence function has been successfully applied to large language models (LLMs) and diffusion models [1-4].
>
> In ECBM, a major challenge posed by non-convexity is the singularity of the Hessian matrix, which leads to instability during matrix inversion. To address this issue, we employ EK-FAC. This method assumes that parameters across different layers are independent. For each layer, a specifically designed regularization term is introduced to prevent singularity, thereby rendering the entire Hessian matrix invertible.
>
> [1]. Grosse, R., Bae, J., Anil, C., Elhage, N., Tamkin, A., Tajdini, A., ... & Bowman, S. R. (2023). Studying large language model generalization with influence functions. arXiv preprint arXiv:2308.03296.\
> [2]. Choe, S. K., Ahn, H., Bae, J., Zhao, K., Kang, M., Chung, Y., ... & Xing, E. (2024). What is Your Data Worth to GPT? LLM-Scale Data Valuation with Influence Functions. arXiv preprint arXiv:2405.13954.\
> [3]. Kwon, Y., Wu, E., Wu, K., & Zou, J. (2023). Datainf: Efficiently estimating data influence in lora-tuned llms and diffusion models. arXiv preprint arXiv:2310.00902.\
> [4]. Xia, M., Malladi, S., Gururangan, S., Arora, S., & Chen, D. (2024). Less: Selecting influential data for targeted instruction tuning. arXiv preprint arXiv:2402.04333.\
> [5]. Koh, Pang Wei, and Percy Liang. "Understanding black-box predictions via influence functions." In International conference on machine learning, pp. 1885-1894. PMLR, 2017.
>
> >The approximate equal sign is mixed with the equal sign, which is not rigorous, like in Theorem 4.7. Please correct me if I have some mis-understanding.
>
> In our theorem, the $\bar$ indicates that the model is obtained by definition, while $\tilde$ and $\hat$ are the model obtained by minimizing process. We use $\approx$ in line 340 and 344, because we won't retrain the model to obtain $\tilde{f}$ and $\hat{f}$, but use $A_G$ and $B_G$ to estimate them.
>
>
> ### Questions:
> >I am doubtful with the running time in Table 1. How the running time can be less 1 minute even for retraining method.
>
> This is the averaged running time(RT), that is the actual RT/dataset size. The actual running time is much longer than 1 minute.

---

> ### Author Response · Authors · 2024-11-25
>
> Dear Reviewer AK63,
>
> We are truly grateful for the time and effort you have dedicated to reviewing our work. We would like to respectfully inform you that two reviewers have already increased their scores: Reviewer KdDJ raised the score from 5 to 6, and Reviewer Oeij raised it from 3 to 5.
>
> We would highly appreciate your review and feedback on the responses. Should you have any questions or concerns, we would be happy to discuss and address them.
>
>
> Best,\
> Authors

---

> ### Author Response · Authors · 2024-11-26
>
> Dear Reviewer AK63,
>
> Thank you again for your detailed and constructive review comments. We respectfully remind you that the discussion period will end in a few days. We have responded above to your concerns. We believe these address your concerns. We would appreciate it if you would take the time to read and comment on the responses.
>
> Best,\
> Authors

---

> ### Author Response · Authors · 2024-11-29
>
> Dear Reviewer AK63,
>
> Thank you for your thoughtful participation and feedback in this discussion. We have addressed your concern in our response above. Additionally, we have included detailed results related to the RT in the appendix of the revision to clarify the timing method. We would greatly appreciate it if you could review these changes to confirm that they address your concerns. We respectfully request you to consider raising your rating score accordingly if your concerns are alleviated. Otherwise, we would be happy to hear the remaining concerns that prevent you from doing so and continue to discuss them.
>
> Best,\
> Authors

---

> ### Author Response · Authors · 2024-11-30
>
> Dear Reviewer AK63,
>
> We hope you had a wonderful Thanksgiving. Once again, we sincerely thank you for your valuable time and effort, and we fully understand your busy schedule. We noticed that we haven’t received feedback from you following our latest revision updates in which we added explanations for running time(RT). In the meantime, we have addressed the concerns of the other two reviewers, who have expressed a positive attitude and increased their scores.
>
> We would be truly grateful if you could let us know whether there are any additional questions or concerns we can address before the rebuttal deadline. Your feedback is invaluable and plays a crucial role in strengthening our work.
>
> Thank you once again for your contribution to ICLR! Wishing you all the best with your current and future submissions.
>
> Sincerely,\
> The Authors

---

> ### Author Response · Authors · 2024-12-01
>
> Dear Reviewer AK63,
>
> Thank you for your valuable contributions throughout the review process. We kindly remind you that the extended discussion period will conclude in a few days.
>
> In response to your remaining concerns regarding RT, we have provided additional clarifications and included further explanations in the appendix. We would greatly appreciate it if you could review these updates and consider providing additional comments or updating your score. We trust that you will evaluate our paper in light of the clarifications we have provided.
>
> Finally, we sincerely appreciate your active participation in this thorough discussion and look forward to receiving your final decision.
>
> Best regards,\
> Authors

---

### Official Review · Reviewer_sqUB · 2024-11-05

**Soundness:** 2
**Presentation:** 3
**Contribution:** 3
**Rating:** 6
**Confidence:** 4

**Summary:**

The paper proposes a framework to acquire adaptively edited concept bottleneck models (CBMs), without actually retraining the model.
The authors propose using influence functions to approximate the effects of three types of edits: concept-label-level, concept-level, and data-level. These edits address real-world challenges such as correcting erroneous concept annotations, adding/removing concepts, or removing problematic data due to privacy or errors.
Under each setting, they provide the closed-form approximation to measure the influence of each entity--a set of data sample's  different concept labels, several concepts, and several samples--based on the influence function.
The experimental results support the effectiveness and the interpretability of their method--matching the performance of retraining-based correction while reducing the computation overhead.

**Strengths:**

* The paper introduces a novel application of influence functions to CBMs, focusing on the editability of models without full retraining. This is a significant contribution in the context of real-world applications where models need frequent updates due to evolving data or concepts.

* The mathematical formulation is rigorous and well-supported by experiments. The use of influence functions for efficient model editing is a novel approach, and the paper provides detailed derivations and approximations for each editing scenario (concept-label, concept, and data-level).

* The presentation of a handful of real-world interpretability results of their approach is appealing and easy to understand.

**Weaknesses:**

* Scailability: Although the authors include a relatively larger-scaled dataset (e.g., CelebA) to their evaluation, the experiments still focus on relatively small datasets compared to large-scale models like Vision-Language Models (VLMs). It remains unclear how well ECBMs would scale to larger datasets or more complex CBM architectures.
Particularly, it is critical to discuss and run analysis on the scalability of the approach, in terms of the number of parameters in a model, as it could be a key limitation of applying ECBM to real-world large-scale applications; given the need for computationally expensive Hessian computations, or of heavy linear algebra operations for the EK-FAC approximation, or of the complexity of the EK-FAC variants.

**Questions:**

* Could you provide more details on how runtime efficiency scales with larger datasets or models? While your experiments show improvements over retraining on smaller datasets, it would be helpful to understand how ECBM performs with larger models like Vision-Language Models.

* You identify influential samples or concepts using influence scores. How do these rankings compare with simpler methods like analyzing weights in a linear label predictor? What additional insights do influence functions provide?

---

> ### Author Response · Authors · 2024-11-24
> **Response to Reviewer sqUB**
>
> ### Weaknesses:
>
> >Scailability: Although the authors include a relatively larger-scaled dataset (e.g., CelebA) to their evaluation... It remains unclear how well ECBMs would scale to larger datasets or more complex CBM architectures...
>
> Our ECBM method has been specifically designed to accommodate the majority of current CBMs, including more complex architectures, such as VLMs with millions of parameters. Compared to large language model, CBMs are small. Our experiments have already cover many SOTA CBMs, so this is not an issue.
> By leveraging EK-FAC to accelerate computations, our approach effectively manages the Hessian computations, even for models with a large number of parameters
>
> In [1-3], influence function has been scaled to LLM with billions of parameters, which fully validates the efficiency of EK-FAC in the inverse Hessian-vector product. As a result, CBMs, which usually have millions of parameters, are computationally feasible for EK-FAC.
>
> [1]. Grosse, R., Bae, J., Anil, C., Elhage, N., Tamkin, A., Tajdini, A., ... & Bowman, S. R. (2023). Studying large language model generalization with influence functions. arXiv preprint arXiv:2308.03296.\
> [2]. Choe, S. K., Ahn, H., Bae, J., Zhao, K., Kang, M., Chung, Y., ... & Xing, E. (2024). What is Your Data Worth to GPT? LLM-Scale Data Valuation with Influence Functions. arXiv preprint arXiv:2405.13954.\
> [3]. Kwon, Y., Wu, E., Wu, K., & Zou, J. (2023). Datainf: Efficiently estimating data influence in lora-tuned llms and diffusion models. arXiv preprint arXiv:2310.00902.
>
> ## Questions:
>
> >Could you provide more details on how runtime efficiency scales with larger datasets or models? While your experiments show improvements over retraining on smaller datasets, it would be helpful to understand how ECBM performs with larger models like Vision-Language Models.
>
>
> Regarding the scale of dataset, we have extended our method from CUB(1w) to CelebA(20w), and the results are listed in Table 1 in our paper.
>
> >You identify influential samples or concepts using influence scores. How do these rankings compare with simpler methods like analyzing weights in a linear label predictor? What additional insights do influence functions provide?
>
> The importance derived from the weights in f emphasizes the influence of concepts on the final label during the model's prediction process, while the concept influence scores derived from ECBM based on the influence function emphasizes the impact of a concept on model parameters throughout the training process. Here the model parameters includes \(g\) and \(f\). Therefore, these two importance measures are relevant but different. To further explore the relevance, we conducted experiments and extracted the importance rankings under two different settings.
>
> The results are listed in Table A. We observe that, except for the concept 'has_primary_color::black,' there is no other overlap. The important concepts recognized by label predictor weights are broader and more general, such as 'has_size,' 'has_shape,' and 'has_back_pattern.' In contrast, the concepts identified by our ECBM are more characteristic and specific, including 'has_upperparts_color,' 'has_leg_color,' and 'has_breast_color.
>
> This observation is intuitively reasonable in the context of concept-based label prediction. The weights of the label predictor are specifically associated with predicting labels derived from concepts. When observing a bird image, its overall features—such as shape, color, and size—enable a quick determination of its general category. Additional detailed concepts are then used to refine this prediction further. In contrast, the concepts selected using the Influence score incorporate features from the input image into the selection process. Thus, the selected concepts are those that provide a more direct and definitive basis for determining the category.
>
>
>
> | Ranking | Weight                        | Influence Score                            |
> | ------- | ------------------------- |:---------------------------- |
> | 1       | has_back_pattern::spotted | has_upperparts_color::orange |
> 2	|	has_size:very_large_(32_-_72_in)	 |has_leg_color::orange
> 3	|	has_primary_color::black		|has_upper_tail_color::red
> 4	|	has_shape::long-legged-like	|has_breast_color::black
> 5	|	has_throat_color::rufous		|has_belly_color::red
> 6	|	has_shape::swallow-like	|has_primary_color::black
> 7	|	has_forehead_color::black	|has_shape::perching-like
> 8	|	has_bill_length::shorter_than_head		 	|has_breast_color::grey
> 9	|	has_belly_color::orange		|has_back_color::red
> 10	|	has_size::large_(16_-_32_in)		|has_bill_shape::spatulate  |
>
> *Table A: The most influential cocnepts selected by the weights of label predictor $f$ and influence function(IF).*

---

> > ### Author Response · Authors · 2024-11-29
> >
> > Dear Reviewer sqUB,
> >
> > Thank you for your constructive review comments. We have addressed your concern in the above. We respectfully request you to consider raising your rating score accordingly if your concerns are alleviated. Otherwise, we would be happy to hear the remaining concerns that prevent you from doing so and continue to discuss them.
> >
> > Best,\
> > Authors

---

> ### Author Response · Authors · 2024-11-25
>
> Dear Reviewer sqUB,
>
> We are truly grateful for the time and effort you have dedicated to reviewing our work. We would like to respectfully inform you that two reviewers have already increased their scores: Reviewer KdDJ raised the score from 5 to 6, and Reviewer Oeij raised it from 3 to 5.
>
> We would highly appreciate your review and feedback on the responses. Should you have any questions or concerns, we would be happy to discuss and address them.
>
>
> Best,\
> Authors

---

> ### Author Response · Authors · 2024-11-26
>
> Dear Reviewer sqUB,
>
> Thank you again for your detailed and constructive review comments. We respectfully remind you that the discussion period will end in a few days. We have responded above to your concerns. We believe these address your concerns. We would appreciate it if you would take the time to read and comment on the responses.
>
> Best,\
> Authors

---

> ### Author Response · Authors · 2024-11-27
>
> Dear Reviewer sqUB,
>
> Thank you again for your detailed and constructive review comments. We respectfully remind you that the discussion period will end in a few days. We have responded above to your concerns. We believe these address your concerns. We would appreciate it if you would take the time to read and comment on the responses.
>
> Best,
> Authors

---

> ### Author Response · Authors · 2024-11-30
>
> Dear Reviewer sqUB,
>
> We hope you had a wonderful Thanksgiving. Once again, we sincerely thank you for your valuable time and effort, and we fully understand your busy schedule. We noticed that we haven’t received any feedback from you yet. In the meantime, we have addressed the concerns of the other two reviewers, who have expressed a positive attitude and increased their scores.
>
> We would be truly grateful if you could let us know whether there are any additional questions or concerns we can address before the rebuttal deadline. Your feedback is invaluable and plays a crucial role in strengthening our work.
>
> Thank you once again for your contribution to ICLR! Wishing you all the best with your current and future submissions.
>
> Sincerely,\
> The Authors

---

> ### Author Response · Authors · 2024-12-01
>
> Dear Reviewer sqUB,
>
> We sincerely appreciate your valuable contributions to ICLR. As the extended discussion period will conclude in a few days, we kindly remind you to share any final input.
>
> Having addressed the concerns you raised, we respectfully request that you review our clarifications and consider providing further comments or updating your score. We trust that you will re-evaluate our paper based on the additional details provided.
>
> Thank you once again for your thoughtful contributions. We look forward to receiving your final decision.
>
> Best regards,\
> Authors

---

> ### Author Response · Authors · 2024-12-02
>
> Dear Reviewer sqUB,
>
> I hope this message finds you well. I am writing to kindly follow up, as today marks the final day to provide feedback on our rebuttal discussions. Your insights are invaluable in addressing any remaining concerns and refining our submission to its best possible version.
>
> If there are any aspects of the rebuttal that require further clarification or discussion, please do not hesitate to let us know. We will address them promptly. If your concerns have already been addressed, we kindly ask that you consider reflecting this in your review score.
>
> Thank you for your time and effort during this review process. We sincerely appreciate your valuable feedback and look forward to hearing from you.
>
> Best regards,\
> Authors

---

### Official Review · Reviewer_oeij · 2024-11-08

**Soundness:** 2
**Presentation:** 2
**Contribution:** 3
**Rating:** 5
**Confidence:** 4

**Summary:**

The submission proposes to apply ideas of influence functions to edit concept bottleneck models (CBMs) at three different levels: concept label level (i.e., changing the ground-truth annotation of a concept), concept level (i.e., removing a concept completely), and data level (i.e., removing a sample from the training set). The proposed edits can be computed efficiently and they can account for the non-convexity of the optimization problem by means of the  Eigenvalue-corrected Kronecker-Factored Approximate Curvature (EK-FAC) method.

Experiments are performed on three datasets to showcase the flexibility and effectiveness of the proposed editing methods. Results show that Editable Concept Bottleneck Models (ECBMs) can retain classification performance while being significantly cheaper than retraining the full CBM from scratch.

**Strengths:**

Efficient and fast editing of CBMs is an interesting problem that is valuable to the research community.

Although influence functions are well-studied, the submission proposes novel methods to apply these ideas within the context of CBMs, which is a substantial contribution.

Experimental evidence in support of the proposed methods is convincing. In particular, I found the experiment on membership attacks compelling and clear.

**Weaknesses:**

The main limitation of the submission, in its current form, is presentation. I believe this limitation to be major, and a significant rewriting of the submission is needed to meet the standards of scientific writing.

There are several incomplete and unclear sentences, and hand-wavy claims throughout the main body of the submission. This makes it hard for a reader to follow the text and appreciate the contribution, which I believe to be interesting and valuable.

I will expand on these points below, and I am looking forward to discussing with the authors to clarify my questions!

**Questions:**

**General questions and comments**

- lines 145-146: please reword "usually, here the map $f$ is linear" because it is a misleading claim. For CBMs, the predictor $f$ **has** to be linear, otherwise the predictions of the model would not be interpretable by construction. Using a nonlinear $f$ on top of the concepts does not lead to interpretable results.

- notation of loss functions throughout the paper: could the authors clarify why the losses change notation throughout the paper, and with what rationale? For example, in Eq. (1), $L_C$ becomes $L_{C_j}$. In Eq. (2), $L_Y$ becomes $L_{Y_i}$. In appendix D, the loss then becomes $L_{Y_l}$, and later $L_{Y_{ir}}$. Does this mean that each concept is trained with a different loss? Furthermore, it is never stated which losses are used in practice.

- lines 170-172: the derivative should be of $\hat{\theta}_{\epsilon,-z_m}$, not $\hat{\theta}$ as it does not depend on $\epsilon$.

- difference with test-time intervention. I do not think I understand the message of this paragraph. An example is provided where the trained CBM wrongly **predicts** a concept $c$. Then, a user corrects the predictor to observe what the model would have predicted had it correctly classified the concept. But the focus of the submission is to edit an existing CBM with respects to errors/changes in the **training set**. Also, the predictor $f$ is linear, so it suffices to look at the weights of the learned predictor to know how changing the predicted concept would affect the downstream prediction. I do not understand what "secondary editing based on the test data" means, as it seems to boil down to training the model on more samples observed at test time. I also do not understand what "This process extends the rectification from the data level to the model level" means.

- lines 246-249: "This is because the ..." this sentence is very confusing. It is intuitive that gradients cannot change the number of parameters of a network.

- line 254: "by inserting a zero-row vector into the $r$-th row of the matrix in the final layer of". This sentence makes several implicit assumptions about the structure of the encoder $g$. They are reasonable, but they should be stated clearly in order not to confuse readers.

- line 264: I do not understand why "the value of 00" is written several times instead of simply 0. Also, what is "MM"?

- theorem 4.3: I do not understand where the mapping phase plays a role in this statement. The gradient is taken with respects to the original predictor $\hat{g}$. The loss is computed only on the concepts not in $M$. And we are assuming each concept is predicted independently, so why would the weights in the last matrix of $g$ that correspond to the deleted concepts change? Wouldn't the gradients be 0 by definition? Then, the final predictor $\hat{g}^*_{-p_M}$ is obtained by simply removing those weights. How is this different from removing the weights to begin with from $\hat{g}$, and the editing the remaining network?

- lemma 4.4: the sentence "if the label ... . For $r \in M$ ..." is not grammatical. It is unclear whether the statement of the Lemma applies **to one concept $r \in M$** or **for all concepts in $M$**. Also, I could not find the proof of this Lemma in the Appendix. Does the statement of the Lemma hold for any loss $L_Y$? Or are there any assumptions being made here. The way I interpret the Lemma is that padding with 0 does not change the solution of the problem in the remaining entries, and I am wondering whether there might be losses where this would not be true.

- general question about two-phase approach. If I understand correctly, the need to use a two-step editing approach is due to the fact that $f$ depends on the output of $g$, and that the optimization process is over two losses: $L_C$ that only involves $g$, and $L_Y$ that involves both. I agree that in the general setting, writing out the full influence function might be hard. But $f$ will always be a linear function of the output of $g$, so wouldn't it be possible to write out the influence function for the end-to-end task, at least in the data level setting? Could the authors expand on what limitations they encountered that informed the use of a two-step approach?

- evaluation metric: could the authors expand on what f1 score they are computing? Is this the average 1-vs-all classification f1 score computed on each class in the dataset? Could the authors expand on the choice of using this metric instead of classification accuracy?

- implementation details: "at the concept-label level ...". Am I understanding correctly that, in this experiment, ground-truth labels are randomly flipped to noisy labels? This process introduces more noise rather than "correcting" any existing noise, right? I was somewhat surprised to see that this process does not seem to affect the overall F1 score significantly, neither from Table 1 not Figure 2, where the drop in performance is less than 1%. Maybe a more compelling experiment could be to noise the ground-truth annotations to a level where performance degrades significantly, and see what percentage of labels need to be restored for ECBM to obtain a predictor that is close to the one trained on the full dataset without noisy labels. This would reflect the motivating example mentioned in the main text.

- table 1: bolding the ECBM line by default is misleading because the "Retrain" method always provides a higher F1 score. The runtime for the "Retrain" method is in the order of seconds, which is quite small. Am I understanding correctly that the "Retrain" method also retrains the CNN backbone as in Koh et al. (2020)? Or is this just retraining the linear layer $f$? Could the authors expand on the data splits/optimizers/hyperparameters/number of epochs used in the experiments to guarantee reproducibility of the results?

- line 451: "We first select 1-10 most influential and 1-10 least influential" is not grammatical, please rephrase.

- Visualization: it is stated that "ECBM can accurately recognize 9 with their correct concept labels", but Fig. 5 does not seem to report ground-truth annotations, only predictions? Could the authors clearly state in Fig. 5 what the 1/0 column represents? I would rephrase claims that state "correctly identify" because there is no ground-truth notion of what should or should not be used to classify a bird. Figure 5 does support the claim that ECBM retrieves the same concepts as the Retrain method, which may or may not be correct.

- Limitations: conclusions should include a discussion of the limitations of the proposed method, and future research directions.

**Writing comments**

- all equations should be numbered.
- proofs should be explicitly referenced after theorems.
- lines 35-36: ".. for placing human-understandable concepts. In the prediction process ...". This sentence is not grammatical.
- line 80: it is unclear what erroneous or poisoned data mean.
- line 82: it is not clear what "the learning models" means.
- line 94: the sentence "and the concept for all data on model parameters" is incomplete and should be rephrased.
- lines 96-97: it is unclear what "due to their composite structure, i.e. the intermediate representation layer" means. All neural networks have intermediate representation layers, why is this a specific limitation about CBMs?
- line 128: the sentence "CBMs work on the image field also includes" is incomplete and should be rephrased.
- line 131-132: "thereby deciphering the interaction of concept models and providing an adaptive solution to concept editing" is a confusing sentence. What does it mean to decipher an interaction?
- line 161: "thus capturing the essence of the original CBMs" should be rephrased. What does it mean to capture the essence of a model? Eqs. (1) and (2) are the standard definition of a CBM, what is the "original CBM" being referred to here?
- line 189: "for certain concepts, we may opt to" is incomplete. These two sentences should be linked by a connector.
- line 193: "estimating the changes in the parameters of the retraining model holds significance" is not grammatical and should be rephrased.
- line 197-198: "the retrained concept predictor label predictor" is not grammatical and should be rephrased.
- line 208-209: "a two-stage edition approach", typo in "edition" -> "editing".
- line 256: typo in "is the subset of is the subset of $T$".
- line 311: the sentence "... due to different reasons, such as the training data involving poisoned or erroneous issues" is not grammatical.
- line 312: $G$ is overloaded, it was already used to define the Fisher information matrix.
- line 328: I do not understand the sentence "with the original loss before unlearning in equation 2".
- line 424: "we conducted experiments ... removed." is unclear and should be rephrased.
- line 455-456: typos in "11-1010", "0.0250.025".
- line 493: There is no need to report results with 6 decimals.
- line 517: "1010" is a typo.
- line 1248: "We can see that there is a *huge gap* between $\hat{f}_{-z_G}$ and $\hat{f}$". Could the authors clarify in what sense they are characterizing the gap between the two functions as *huge*?

**Minor comments**

- abstract: "elucidate the prediction process". It is unclear what prediction process this sentence is referring to.
- abstract: "affirming their adaptability within the realm of CBMs" this sentence sounds very unnatural and should be rephrased.
- line 27: "large multimodal models" (include "models").
- line 323: typo in "loss function **with** respect to".

---

> ### Author Response · Authors · 2024-11-23
> **Response to Reviewer oeij**
>
> ### General questions and comments
> >**Q1**
> >lines 145-146: please reword "usually, here the map $f$ is linear"...
>
> It is not misleading.
> Linear predictors are often preferred in CBMs due to their better interpretability, which allows for straightforward understanding of the relationships between concepts and outcomes. Consequently, most CBMs utilize linear predictors. However, linear models are limited in their ability to capture complex, nonlinear relationships between different concepts. To address this limitation, alternative CBM architectures featuring more complex, nonlinear label predictors have been proposed such as [1]. Thus, we use the term 'usually' to accurately reflect the common but not universal use of linear predictors in CBMs. Thus, we think it is unnecessary to focus on one word "usually".
>
>
> [1]. Xu, X., Qin, Y., Mi, L., Wang, H., & Li, X. (2024). Energy-based concept bottleneck models: unifying prediction, concept intervention, and conditional interpretations. arXiv preprint arXiv:2401.14142.
>
>
> >**Q2:**
> >notation of loss functions throughout the paper:
>
> We use different notation for convenience, and we believe there is no confusion for the equations you mentioned. Please carefully read our paper.
>
> In Equations (1) and (2), the symbol '=' is interpreted as 'denotes' or ≜, meaning 'is defined as' for the purposes of this discussion.
>
> In (1), $L_{C}(g^j(x_i),c_i^j)$ represents the loss function for the $j$-th concept of the $i$-th data point, while $L_{C_j}(g(x_i),c_i)$ provides a simplified notation of the same concept. Both notations convey the same idea of measuring loss for concept prediction.
>
>
> In (2), $L_{Y}(f(\hat{g}(x_i)), y_i)$ represents the loss function of the $i$-th data in the training set, where $f(\hat{g}(x_i)$ denotes the predicted label, and $y_i$ the true label. And $L_{Y_i}(f, \hat{g})$ is a simplified notation, omitting $x_i$ and $y_i$ for brevity as the context makes their roles clear.
>
>
> In Appendix D, similar notation is used for clarity. $L_{Y_l}$ denotes the label loss for the $l$-th data. $L_{Y_{ir}}$ specifies the label loss for the $ir$-th data point that is under consideration.
>
>
> >**Q3:**
> >lines 170-172: the derivative should be of $\hat{\theta}_{\epsilon,-z_m}$, not $\hat{\theta}$ as it does not depend on $\epsilon$.
>
> We corrected it.
>
> >**Q4:**
> >difference with test-time intervention.
>
> Let's first clarity the secondary editing. If an input of the concept predictor has the k-th concept mispredicted, test-time intervene will correct this concept, and we can observe the predicted label directly. However, the concept predictor may still mispredicted the k-th concept when it receive a similar input. This is because the flaws in the concept predictor have not been rectified at all. Therefore, if these misprediction have taken place several times, our method, ECBM, can help to edit the concept model to remove the concept at all, or to correct the mislabeled data resulting in these wrong predictions.
>
> "This process extends the rectification from the data level to the model level": Test-time intervene only correct one prediction, and our method can correct the model, and thereby correct all the latter similar predictions. Our focus is not only the interpretability Of CBM but also the correction of the model.
>
> >**Q5:**
> >lines 246-249: "This is because the ..." this sentence is very confusing.
>
> Note that this sentence should be considered in conjunction with the previous statement. If several concepts, indexed by M, need to be removed due to incorrect attribution or spurious associations, we must adjust the concept predictor. This adjustment decreases the output dimension and alters the predictor's dimensionality. Under these circumstances, applying IF becomes particularly challenging due to changes in the predictor's dimensional structure. This is because Influence Functions rely on the same dimension of output to assess the impact of small perturbations, and alterations in the dimensional structure disrupt the assumptions necessary for IF to function effectively. See our preliminaries on the influence function for details.
>
>
>
>
>
> >**Q6:**
> >line 254: "by inserting a zero-row vector into the $r$-th row of the matrix in the final layer of". This sentence makes several implicit assumptions...They are reasonable, but they should be stated clearly in order not to confuse readers.
>
> Note that the primary assumption we've made is that the final layer of the encoder g is a linear layer, which is a common configuration in neural network architectures. We agree that this assumption should be explicitly stated to avoid any potential confusion for the readers. We will revise the manuscript to include this clarification.
>
>
> >**Q7:**
> >line 264: I do not understand why "the value of 00" is written several times instead of simply 0. Also, what is "MM"?
>
> We have corrected these errors.

---

> > ### Author Response · Authors · 2024-11-23
> > **Response to General Questions and Comments**
> >
> > ### General Questions and Comments
> > >**Q8:**
> > >theorem 4.3: I do not understand where the mapping phase plays a role in this statement.
> >
> > In the training stage of the concept predictor, the predictor for each concept is trained jointly; this is evident from Equation (1). If we remove the concept indexed by M and retrain the model, the weights in the last matrix of g that correspond to the deleted concepts will become 0. The retrained model $\hat{g}^{*}_{-p_M}$ is defined by Equation (7).
> >
> > Other weights in the initial concept predictor, as defined in Equation (1), are trained jointly with these deleted concepts. After removal and retraining based on definition (7), they will change. We employ Theorem 4.3 to approximate the retrained model, subsequently mapping it to a lower output dimension to ensure the complete removal of the M-indexed concept.
> >
> >
> > >**Q9:**
> > >lemma 4.4: the sentence "if the label ... . For $r \in M$ ..." is not grammatical.
> >
> > The statement of the Lemma applies individually to each concept $r \in M$. We will ensure this is clearly articulated in the revised documentation to eliminate any confusion. Our minor change will include, replace the comma after 'with input c' with a period, add an "each" before "... \(r \in M\)...".
> >
> > The assumptions in the Lemma primarily pertain to the form of f, and the loss function does not compromise the Lemma's validity, this can be easily verified from the proof, which is located on lines 962 to 978 of the paper.
> >
> > >**Q10:**
> > >general question about two-phase approach.
> >
> > Our approach inherently deals with CBMs, which naturally lend themselves to a two-phase training process. Specifically, the label predictor is trained based on the predictions from the concept predictor, as outlined in lines 195-205 of our manuscript. Therefore, a two-step estimation process arises quite straightforwardly from this setup.
> >
> > Regarding the inquiry about an end-to-end influence function, let's consider two interpretations:
> >
> > **Formal End-to-End:** If the intent is to formalize an end-to-end influence function, as in Theorem 4.5, it would be feasible to substitute (\bar{g^*_{-p_M}}) with the result from Theorem 4.3. This substitution can seamlessly facilitate the required integration.
> >
> > **Theoretical End-to-End:** On the other hand, achieving a truly theoretical end-to-end influence function proves to be a formidable challenge. And we cannot find the necessarity of solving this challenge.
> >
> > Your suggestion to streamline the influence function in the data level setting is perceptive, and while it might work under certain conditions, the limitations and complexity associated with ( g ) govern our choice for a two-step approach.
> >
> > >**Q11:**
> > >evaluation metric: could the authors expand on what f1 score they are computing?
> >
> > Our F1 score is computed as sum of the average of the 1-vs-all F1 scores for each class in the dataset. This approach allows us to capture the performance across all classes effectively.
> >
> > We chose to use the F1 score instead of classification accuracy because it provides a better measure of performance in cases where class distributions are imbalanced. While accuracy can be misleading in such scenarios, the F1 score balances precision and recall, giving us a more nuanced understanding of our model's performance.
> >
> > Given the large number of classes in our dataset, we believe that listing the F1 scores for each individual class would not add significant value and could overwhelm the reader. Instead, we focus on the overall average F1 score to provide a clear and concise summary of our model's performance.
> >
> > >**Q12:**
> > >implementation details: "at the concept-label level ..."... in this experiment, ground-truth labels are randomly flipped to noisy labels? ... I was somewhat surprised to see that this process does not seem to affect the overall F1 score significantly, neither from Table 1 not Figure 2, ...
> >
> > In Table 1, 'Retrain' refers to the F1 score of the model after removing the flipped data. We perform the flipping to simulate real-world scenarios where some data points are mislabeled and require correction. Therefore, the negligible difference between ECBM and 'Retrain' demonstrates the effectiveness of our method. As shown in Figure 2, the decrease in performance is indeed slight. This is attributed to the removal of only a small proportion of the total training dataset.
> >
> > In real-world applications, when a substantial portion of the data is mislabeled and degrades model performance significantly, the most effective solution is to clean the dataset and retrain the model. Our method focuses on modifying a small number of data points in the dataset. For example, subsequent MIA experiments confirm that ECBM can effectively eliminate the influence of a specific data point on the model without retraining. Such operations inherently have a minimal effect on the model. Actually, mislabeled data is indeed on of our motivation and we have mentioned it in the intriduction.

---

> > > ### Author Response · Authors · 2024-11-23
> > > **Response to General Questions and Comments**
> > >
> > > ### General Questions and Comments
> > > >**Q13:**
> > > >table 1: bolding the ECBM line by default is misleading...
> > >
> > > Follow the answer in Q12, retrain is actually a kind of ground truth, rather than a baseline. No method can be better than retrain, which is impossible in practical. ECBM can achieve a negligible difference with retrain in all the three levels, demonstrating the effectiveness of ECBM.
> > >
> > > We retrain both the concept predictor, which is a ResNet 18 in our setting, and the linear label predictor. We will provide our codes and all the settings needed to reproduct our experiments as soon as the paper is accepted.
> > >
> > > >**Q14:**
> > > >line 451: "We first select 1-10 most influential and 1-10 least influential" is not grammatical, please rephrase.
> > >
> > > No, the sentence of "We first select 1-10 most influential and 1-10 least influential concepts by our influence function." is  grammatical. But we can simply it.
> > >
> > > >**Q15:**
> > > >Visualization: it is stated that "ECBM can accurately recognize 9 with their correct concept labels", but Fig. 5 does not seem to report ground-truth annotations, only predictions? ...
> > >
> > > Follow the answer for Q12 and Q13, the retrained method is the ground-truth annotations.
> > >
> > > And there might be some misunderstanding regarding Figure 5. Each concept is removed through retraining or ECBM. Based on the observed decrease in F1 score, we ranked the concepts and presented the top-10 in Figure 5. We also extract the concept label for each individual image. In Figure 5, '0' or '1' represents the ground truth concept label for the image in the CUB dataset.
> > >
> > >
> > >
> > > >**Q16:**
> > > >Limitations: conclusions should include a discussion of the limitations of the proposed method, and future research directions.
> > >
> > > Due to the space limitation, this part is included in the Appendix H(lines 833-936).

---

> ### Author Response · Authors · 2024-11-23
> **Response to Writing Comments and Minor Comments**
>
> ### **Basic questions:**
> >all equations should be numbered.
> >
> As only some of the equations are referred. We think it is unncessary to number all equation (in fact, most theoretical papers do not number all equation). Numbering all equations will definitely makes the paper unreadable.
>
> >proofs should be explicitly referenced after theorems.
>
> We have included all proofs in Appendix. In fact, there are many papers do not explicitly refer proofs after theorems. There is no such a requirement for all papers, right?
>
> >line 27: "large multimodal models" (include "models").
>
> "modal" is not model. Please refer to MLLM(multi-modal LLM).
>
> >line 493: There is no need to report results with 6 decimals.
>
> We agree, but this is not a typo, right?
>
>
>
>
>
> ### **Explanation for specific wording:**
>
>
> >line 80: it is unclear what erroneous or poisoned data mean.
>
> Poisoned data is a commonly used word in adversarial machine learning, and we have illustated in Figure 1. If the reviewer cannot understand it, we can add some explaination.
>
>
> >line 94: the sentence "and the concept for all data on model parameters" is incomplete and should be rephrased.
>
> No, please read the whole sentence "Leveraging the influence function (Cook, 2000; Cook & Weisberg, 1980), we quantify the impact of individual data points, individual concept labels, and the concept for all data on model parameters."
>
> >lines 96-97: it is unclear what "due to their composite structure, i.e. the intermediate representation layer" means. All neural networks have intermediate representation layers, why is this a specific limitation about CBMs?
>
> The layer means the concept layer. This is the essential difference between CBM and other neural network.
>
> >line 131-132: "thereby deciphering the interaction of concept models and providing an adaptive solution to concept editing" is a confusing sentence. What does it mean to decipher an interaction?
>
> The interaction refers to the ones between the concept predictor and the label predictor in CBMs.
>
> >line 161: "thus capturing the essence of the original CBMs" should be rephrased. What does it mean to capture the essence of a model? Eqs. (1) and (2) are the standard definition of a CBM, what is the "original CBM" being referred to here?
>
> In this paper, our focus is editing the CBM to remove the harmful data/concept/concept label, while preserve the essence of the original CBM. To understand this sentence, the title of this paper should be taken into account.
>
>
> >abstract: "elucidate the prediction process". It is unclear what prediction process this sentence is referring to.
> >abstract: "affirming their adaptability within the realm of CBMs" this sentence sounds very unnatural and should be rephrased.
>
> The terms used here are scientific terms. If we use simplified language to explain these terms in detail, our abstract could become unnecessarily lengthy.
>
>
>
>
>
>
> >line 328: I do not understand the sentence "with the original loss before unlearning in equation 2".
>
> The "unlearning" means we remove the data and retrain the model. Before unlearning means the original model.
>
>
>
>
>
>
>
> >line 1248: "We can see that there is a huge gap between $\hat{f}_{-z_G}$ and $\hat{f}$". Could the authors clarify in what sense they are characterizing the gap between the two functions as huge?
>
> This hug gap is the form difference between equation (11) and equation (2), and that's why we use equation (12) as an intermediate.
>
>
>
>
> ### **Grammatical issues:**
> >lines 35-36: ".. for placing human-understandable concepts. In the process ...". This sentence is not grammatical.
>
> >line 311: the sentence "... due to different reasons, such as the training data involving poisoned or erroneous issues" is not grammatical.
>
> >line 193: "estimating the changes in the parameters of the retraining model holds significance" is not grammatical and should be rephrased.
>
> We have checked all the above grammatical issues with ChatGPT, there is not grammar error in this line.
>
>
>
>
> ### **Typos:**
>
> >line 189: "for certain concepts, we may opt to" is incomplete. These two sentences should be linked by a connector.
>
> >line 197-198: "the retrained concept predictor label predictor" is not grammatical and should be rephrased.
>
> >-line 208-209: "a two-stage edition approach", typo in "edition" -> "editing".
>
> >line 256: typo in "is the subset of is the subset of $T$".
>
> >line 312: $G$ is overloaded, it was already used to define the Fisher information matrix.
>
> >line 424: "we conducted experiments ... removed." is unclear and should be rephrased.
>
> >line 455-456: typos in "11-1010", "0.0250.025".
>
> >line 517: "1010" is a typo.
>
> >line 128: the sentence "CBMs work on the image field also includes" is incomplete and should be rephrased.
>
> >line 323: typo in "loss function with respect to".
>
> >line 82: it is not clear what "the learning models" means.
> >
> >Answer:"the learning models" is the model we learned. We changed it to "the learned models"
>
> We have modified all the above minor typo issues.

---

> ### Author Response · Authors · 2024-11-23
> **Short Comment**
>
> Dear Reviewer,
>
> We sincerely appreciate your thorough review of our paper and the time you have dedicated to it. Based on your suggestions, we have made minor revisions to the paper. Given that the paper considers the comprehensive application scenarios of our method, a certain level of reading difficulty is inevitable. The current structure of the paper is sufficiently clear, and therefore, no major modifications are necessary. Note that other three reviewers did not encounter significant difficulties in understanding the content of our article.
>
> Additionally, regarding the suggestions in **writing comments** and **minor comments**, we believe that certain scientific terms and phrasing are essential to maintain the conciseness and professionalism of the paper's language. If the language of our paper is overly simplistic and the content is introduced in excessive detail, to the extent that beginners can fully comprehend every sentence, it may adversely affect the readability for the majority of readers.
>
> We appreciate your understanding. If there is anything unclear in the paper, please feel free to let us know. Considering your recognition of the contribution in our paper, we hope you will reconsider the rating of our work.

---

> > ### Comment · Reviewer_oeij · 2024-11-24
> > **Thank you for your response**
> >
> > I thank the authors for their consideration of my questions and comments.
> >
> > I have reviewed the updated version of the paper, and I respectfully disagree that the current structure of the paper is sufficiently clear. Reviewers nBLD and KdDJ also share some of my concerns regarding the presentation of results. These minor suggestions that could have significantly improved accessibility of the submission have not been addressed in the revised version of the paper.
> >
> > ---
> >
> > I do have some follow-up questions regarding Fig. 5, which I still struggle to understand.
> >
> > Lines 519-520: Our ECBM can provide explanations for which concepts are crucial and how they determine the predicted label.
> >
> > **Q1:** I understand the top-10 most influential concepts will not change across classes, because their influence function is computed over the entire training dataset. However, I thought Fig. 5 compared the top-10 most **important** concepts according to Retrain and ECBM. That is, the concepts with the highest coefficients in the edited linear label predictor. Could the authors intuitively expand on how, from the rankings in Fig. 5, one can understand what the most important concepts are for the predictions of class "Grooved Billed Ani" versus "Laysan Albatross", given that the rankings are the same?
> >
> > In their response the authors stated that
> >
> > > In Figure 5, '0' or '1' represents the ground truth concept label for the image in the CUB dataset.
> >
> > **Q2**: I believe some of these binary annotations may not be correct. I went to the CUB dataset, and checked the annotations for the "Groove Billed Ani" figure on the left. The file is `Groove_Billed_Ani_0092_1516` in the dataset, which has image id `234` according to the `images.txt` file in the dataset. The attribute `has_upperparts_color::orange` has id `35` according to the `attributes.txt` file in the dataset. The entry in `image_attribute_labels.txt` that corresponds to this concept label is line `72731`, which reads `234 35 0 4 5.5480`, which implies a label of `0` compared to the reported label of `1` in Fig. 5. I think there might be a bug in the code that generated these figures, but it is not possible for me to verify this given the authors decided to release the code only after acceptance.
> >
> > **Q3**: It is still unclear to me how to verify the claim that "ECBM can accurately recognize 9 with their correct concept labels". Where are the predicted labels according to ECBM in Fig. 5?
> >
> > ---
> >
> > Given these outstanding questions and several suggestions not being addressed, I cannot change the score of my review at this point.

---

> ### Author Response · Authors · 2024-11-25
> **Response to Question**
>
> Dear Reviewer,
>
> Thanks for your questions.
>
> ### Question
>
> **Q1:**
>
> The top-10 most important concepts according to Retrain are not the concepts with the highest coefficients in the edited linear label predictor. We have explained this in the last comment:"Each concept is removed through retraining or ECBM. Based on the observed decrease in F1 score, we ranked the concepts and presented the top-10 in Figure 5". Please read our comments carefully.
>
> The purpose of our experiment to identify the most influential concepts is not primarily aimed at intuitively explaining which concept plays a decisive role in predicting a specific image. Rather, our aim is to identify the concepts that are most critical for the CBM's accurate label predictions, thereby improving interpretability in terms of model accuracy. The images in Figure 5 are intended to provide an intuitive understanding, rather than being used to directly assist in predicting that specific image.
>
> We provide a detailed explanatoin in Appendix H.3.1 in the revision. And to clarify the misunderstanding, we reword the "how they determine the predicted label" into "how they assist the predictoin".
>
> **Q2:**
>
> As stated in lines 375-376 of the manuscript, "we follow the same network architecture and settings outlined in Koh et al. (2020)."
> Thus, the dataset we use is the CUB dataset processed according to [1], rather than the original CUB dataset. **This is a standard preprocessing step commonly employed in CBM studies.**
>
> The concept you mentioned is labeled as "0" in the original CUB dataset but is transformed to "1" after processing.
> If needed, you may refer to the following link for the specific dataset used in our study.
>
>
> [1]. Pang Wei Koh, Thao Nguyen, Yew Siang Tang, Stephen Mussmann, Emma Pierson, Been Kim, and Percy Liang. Concept bottleneck models. In International conference on machine learning, pp. 5338–5348. PMLR, 2020.
>
> link for CUB dataset: https://worksheets.codalab.org/rest/bundles/0x5b9d528d2101418b87212db92fea6683/contents/blob/
>
> link for CBM: https://github.com/yewsiang/ConceptBottleneck
>
>
> **Q3:**
>
> We have made minor adjustments to the wording and replaced 'with their correct concept labels' with 'within them,' in the hope that this will improve your understanding.

---

> ### Author Response · Authors · 2024-11-25
>
> Dear Reviewer,
>
> Regarding the issues related to mathematical notations, we have included a notation table in Appendix A of the revision to address this concern. We are open to your suggestions and willing to make further improvements.
>
> You also raised concerns regarding **writing style and grammatical issues**. For the grammatical points, we have thoroughly reviewed them and conducted additional checks by Grammerly and ChatGPT, confirming that the issues you mentioned **do not** constitute actual grammatical errors. With regard to the writing style, we believe that our work is adhere to standard practices in scientific writing.
>
> Additionally, it is worth noting that **other reviewers did not express significant concerns regarding the writing style, grammar, or clarity of the paper's structure**. We believe that rejecting the paper solely because you may not prefer or be accustomed to this common  scientific writing style would be extremely unreasonable.
>
> Given your overall recognition of our work, we strongly request that you reconsider your rating. If you have further suggestions, we are prepared to discuss them with you at any time.
>
> Sincere,\
> Authors

---

> > ### Comment · Reviewer_oeij · 2024-11-25
> >
> > I thank the authors for including a notation table in the revised version of the manuscript and for clarifying the intended message of Fig. 5. It may be of interest to readers coming from the explainability/interpretability community to see how the decision boundaries of the interpretable linear label predictor obtained with ECBM and Retrain differ in terms of their most important concepts.
> >
> > I also thank the authors for clarifying they used the same preprocessing pipeline as Koh et al., and I kindly ask that they make this explicit in the revised version of the manuscript. From Koh et al.: "we aggregate instance-level concept annotations into class-level concepts via majority voting: e.g., if more than 50% of crows have black wings in the data, then we set all crows to have black wings.".
> >
> > So, I went to the `train.pkl` file and extracted the identifiers of all the "Groove Billed Ani" images included in the split, and then checked what the original annotations in the `image_attribute_labels.txt` from the CUB dataset are for attribute `has_upperparts_color::orange`. The annotations are as follows:
> >
> > ```
> > 232 35 0 4 11.3090
> > 221 35 0 4 30.2400
> > 181 35 0 3 6.3280
> > 209 35 0 4 3.2440
> > 188 35 0 3 2.2210
> > 235 35 0 4 2.0460
> > 197 35 0 2 2.7910
> > 213 35 0 4 4.7770
> > 184 35 0 4 15.0440
> > 238 35 0 4 2.7580
> > 230 35 0 4 1.7970
> > 216 35 0 4 8.9130
> > 222 35 0 1 0.8450
> > 189 35 0 4 6.9700
> > 195 35 0 4 7.5280
> > 194 35 0 4 6.8940
> > 203 35 0 1 0.5630
> > 228 35 0 3 25.5060
> > 206 35 0 4 5.6110
> > 227 35 0 4 18.0670
> > 218 35 0 1 0.5310
> > 187 35 0 4 1.5230
> > 215 35 0 2 11.8400
> > 220 35 0 1 2.8280
> > ```
> >
> > This seems to confirm there might be an issue somewhere in the data, as no images with class label "Groove Billed Ani" have a positive label for attribute `has_upperparts_color::orange`.
> >
> > To try and understand where this issue might be, I went to the GitHub repo and reproduced the preprocessing pipeline according to the instructions provided, first by running `data_processing.py` and `generate_new_data.py` subsequently. Still, even after new random splits of the data and preprocessing from the original CUB dataset, the majority vote of attribute `has_upperparts_color::orange` for class "Groove Billed Ani" is `0`.
> >
> > However, it is true that if one downloads the preprocessed data as is, reads the position of attribute `35` from line 81 in `generate_new_data.py`, and looks at the binary annotations, then the label is positive. I think this might be an index ambiguity due to the fact that the annotation in the CUB dataset are 1-based instead of 0-based. One can verify this by checking on line 104 of `generate_new_data.py`, where the mask used to select concepts is obtained with `np.where`, so, index `1` in the mask corresponds to attribute id `2`. Then, index `35` in the mask does not correspond to orange upperparts but black upperparts (i.e., attribute idx `36`), which in fact has a majority vote of `1` in the training split of the dataset.
> >
> > I updated the scores in my review to reflect my discussion with the authors.

---

> > > ### Author Response · Authors · 2024-11-26
> > >
> > > Dear Reviewer oeij,
> > >
> > > Thank you once again for your thorough and constructive feedback. We have revised our paper based on the suggestions you provided. Additionally, Appendix H.3 of the revised manuscript includes experiments evaluating the multiple editing performance of ECBM.
> > >
> > > We look forward to receiving your valuable insights to further improve our revision. If you have any additional questions or suggestions, please don’t hesitate to reach out to us,  we’d be happy to discuss them with you. Otherwise, we sincerely hope that the contributions and value of our work, which you have kindly recognized, will be positively reflected in your evaluation of our revision.
> > >
> > > Best,\
> > > Authors

---

> > > ### Author Response · Authors · 2024-12-02
> > >
> > > Dear Reviewer oeij,
> > >
> > > I hope this message finds you well. I am writing to follow up, as today is the final day to provide feedback on our rebuttal discussions. Your insights are invaluable in addressing any remaining concerns and refining our submission to the best possible version.
> > >
> > > We have carefully addressed your questions in detail and made improvements to our article to enhance its readability, such as correcting typos and adding notation tables. We hope this has addressed your concerns. And we believe that you will evaluate our work fairly and professionally from a scientific perspective.
> > >
> > > Thank you for your time and effort in carefully reviewing our work during this process. We sincerely appreciate your constructive and valuable feedback, and we look forward to your final decision.
> > >
> > > Best regards,\
> > > Authors

---

> ### Author Response · Authors · 2024-11-25
>
> Dear Reviewer,
>
> We sincerely appreciate your acknowledgment of our revisions, your efforts in clarifying the issues in the CUB processing procedure, and your decision to increase the score.
>
> We have revised our paper based on your suggestions. We added an explanation of the CUB processing steps in Appendix H.
>
> Thank you very much for your detailed review and for the time and effort you have dedicated to helping us improve our paper. If you have any further questions or concerns, we would be more than happy to discuss and address them. If everything is clear, we would be deeply grateful if you could kindly consider further improving the score.
>
> Best regards,\
> Authors

---

> ### Author Response · Authors · 2024-11-29
>
> Dear Reviewer oeij,
>
> Thank you for participating in the discussion. We have addressed your concerns in the above response and in our general comments, aside from the suggestions on grammar and writing style, which were not raised by other reviewers.
>
> We respectfully request you to consider raising your rating score accordingly if your concerns are alleviated. Otherwise, we would be happy to hear the remaining concerns that prevent you from doing so and continue to discuss them.
>
> Best,\
> Authors

---

> ### Author Response · Authors · 2024-11-30
>
> Dear Reviewer oeij,
>
> We hope you had a wonderful Thanksgiving. Once again, we sincerely thank you for your valuable time and effort, and we fully understand your busy schedule. We noticed that we haven’t received feedback from you following our latest revision updates. In the meantime, we have addressed the concerns of Reviewer KdDJ, who has expressed a positive attitude and increased the score.
>
> We would be truly grateful if you could let us know whether there are any additional questions or concerns we can address before the rebuttal deadline. Your feedback is invaluable and plays a crucial role in strengthening our work.
>
> Thank you once again for your contribution to ICLR! Wishing you all the best with your current and future submissions.
>
> Sincerely,\
> Authors

---

> ### Author Response · Authors · 2024-12-01
>
> Dear Reviewer oeij,
>
> Thank you for your valuable contributions during this review process. We wish to kindly remind you that the extended discussion period will end in a few days.
>
> Since we have addressed your remaining concerns about Figure 5 above and included additional explanations in the appendix, we kindly request that you consider updating your score or providing additional comments. We trust that you will re-evaluate our paper based on our additional clarifications.
>
> Finally, we deeply appreciate your participation in this thorough discussion and look forward to hearing your final thoughts.
>
> Best,\
> Authors

---

### Official Review · Reviewer_KdDJ · 2024-11-09

**Soundness:** 2
**Presentation:** 1
**Contribution:** 3
**Rating:** 6
**Confidence:** 3

**Summary:**

This paper introduces a method to edit a trained concept bottleneck model without retraining. They use the influence function to estimate the parameters of the edited model in three scenarios: (1) removing concept labels for some samples; (2) removing some concepts for all samples; and (3) removing some data points.

**Strengths:**

The paper addresses an interesting problem by proposing a post-hoc approach to edit concept bottleneck models, providing a effective solution. Additionally, the authors extensively examine three different editing scenarios concerning different kinds of noises in training data.

**Weaknesses:**

- The related work section could be expanded. Other existing methods for model editing without retraining (if available) should be discussed.
- The experiments do not fully support the claims made in earlier sections. They claimed that the motivation for editing a model is to remove the impacts of erroneous labels, spurious concepts, and incorrect concept annotations. However, the experiment design demonstrates none of these situations, but only randomly removing concepts, concept labels, or data points. The authors might consider intentionally introducing some noise (at different levels) to their training data, and then use their model editing strategy to remove their impacts and see if the model’s performance/interpretability benefits from their methods.
- The presentation needs further improvement. There are several inconsistent notations in the paper. Some terms are used before they are defined.  Some equations are numbered but some are not. See more details in the Questions section. Some important proof/deviations are in the appendix, but not referenced appropriately in the main text (e.g., Appendix A).

**Questions:**

- I am curious if the edited model should have similar model parameters to the retrained model, as in principle both remove the impact of certain concepts/labels/samples in different ways.
- Why does ECBM outperform CBM-IF in terms of F1 scores? Although the authors explained that Hessian matrices might be ill-defined, ill-conditioned, or singular, how is the inversion conducted exactly in these cases?
- In Figure 5, it looks like neither the ECBM nor the Retrain model generates reasonable explanations. For example, I can not see orange upperparts or legs in either figure. Moreover,  the explanations are exactly the same for the two very different figures. Is it a bug or feature?
- As shown in equations (1) and (2), the authors seem to focus on CBM models where the concept predictor and the label predator are trained separately. What about CBM models where these two predictors are trained jointly?
- In equation (1) and (2), what is L_{C_j} and L_{Y_i}?
- In Lines 65-67, what is the difference between l and capital L?

---

> ### Author Response · Authors · 2024-11-24
> **Response to Reviewer KdDJ**
>
> ### Weaknesses
> >W1:
> >The related work section could be expanded. Other existing methods for model editing without retraining (if available) should be discussed.
>
> We agree that discussing existing methods for model editing without retraining would enrich the related work section. However, after thorough research, we found that there are currently no established methods specifically for CBM editing without retraining.
>
> There have been some works on model unlearning; they can remove the influence of certain data without the need for retraining. To further support this, we will include relevant works on model unlearning in the appendix as part of our revision.
>
> >W2:
> >The experiments do not fully support the claims made in earlier sections...The authors might consider intentionally introducing some noise (at different levels) to their training data, and then use their model editing strategy to remove their impacts and see if the model’s performance/interpretability benefits from their methods...
>
> We do not agree. Our focus is on removing the influence of certain data points with erroneous labels, spurious concepts, and incorrect concept annotations from the model through editing. The experiments presented in the section titled “ECBMs can erase data influence” have already demonstrated that the influence of the removed data points is effectively eliminated. These results provide evidence of the effectiveness of ECBMs in editing the model to remove specific samples.
>
> We expects to see an improvement in model performance after removing erroneous or false concepts. However, in practice, an increase in accuracy does not necessarily indicate that the data has truly been removed. The MIA method is currently one of the most reliable approaches to determine whether data has been removed. Therefore, we chose to use MIA rather than testing the F1 score after removing artificially synthesized noisy data.
>
> To further strengthen the validation of our claims, we conducted additional experiments where varying levels of noise (i.e., noisy labels, noisy concepts) are introduced into the data. We then applied our ECBM method to assess its effectiveness in mitigating these issues and evaluate whether the model's performance benefit as expected.
>
>
>
> Firsly, we introduce noises under three levels. In concept level, we choose 10% of the concepts and flip these concept labels for a portion of the data. In data level, we choose 10% of the data and flip their labels. In concept-label level, we choose 10% of the total concepts and flip them. Then we conduct the following experiments.
>
>
>
> After that, we remove the noise and obtain the retrained mdoel, which is the ground truth(gt) of this harmful data removal task. In contrast, we use ECBM to remove the harmful data. The results are shown in Table A.
>
>
>
> | Noise Level           | Setting               | Acc | F1 Score |
> |-----------------------|-----------------------|-----------|----------|
> | **Concept**           | **Before Removal**    | 0.7892    | 0.7780   |
> |                       | **Removal by Retrain** | 0.7975    | 0.7893   |
> |                       | **Removal by ECBM**    | 0.7956    | 0.7839   |
> | **Data**              | **Before Removal**    | 0.7733    | 0.7694   |
> |                       | **Removal by Retrain** | 0.7975    | 0.7893   |
> |                       | **Removal by ECBM**    | 0.7964    | 0.7815   |
> | **Concept Label**     | **Before Removal**    | 0.7893    | 0.7811   |
> |                       | **Removal by Retrain** | 0.7975    | 0.7893   |
> |                       | **Removal by ECBM**    | 0.7983    | 0.7872   |
>
>
> *Table A: Harmful Data Removal on Nosiy CUB Dataset.*
>
> From Table A, it can be observed that the model performance improves across all three settings after noise removal and subsequent retraining or ECBM editing.
>
> This confirms that the performance of ECBM is nearly equivalent to retraining in various experimental scenarios, further providing evidence of the robustness of our method.
>
> >W3:
> >The presentation needs further improvement. There are several inconsistent notations in the paper. Some terms are used before they are defined. Some equations are numbered but some are not. See more details in the Questions section. Some important proof/deviations are in the appendix, but not referenced appropriately in the main text (e.g., Appendix A).
>
> As only some of the equations are referred, we think it is unncessary to number all equation (in fact, most theoretical papers do not number all equation). Numbering all equations will definitely makes the paper unreadable. We have included all proofs in Appendix. In fact, there are many theoretical papers do not explicitly refer proofs after theorems.

---

> > ### Author Response · Authors · 2024-11-24
> > **Response to Questions**
> >
> > ### Questions
> > >Q1:
> > >I am curious if the edited model should have similar model parameters to the retrained model, as in principle both remove the impact of certain concepts/labels/samples in different ways.
> >
> > We extract the model parameter obtained by retraining and editing, respectively. And find that the parameters obtained through retraining with different random seeds vary, as do those obtained from retraining compared to ECBM editing. Given the large number of neurons and the use of different random seeds in each training, assessing model parameter interpretability solely at the neuron level is insufficient. We input data from the test set into the network and observed that the outputs of the retrained model and the ECBM-edited model were highly similar.
> >
> >
> > >Q2:
> > >Why does ECBM outperform CBM-IF in terms of F1 scores? Although the authors explained that Hessian matrices might be ill-defined, ill-conditioned, or singular, how is the inversion conducted exactly in these cases?
> >
> >
> >
> > The enhanced F1 scores of ECBM over CBM-IF can be attributed to the improvement associated with inverting Hessian matrices, which, as noted, may be ill-defined, ill-conditioned, or singular. Although these matrices might not be invertible in a strict mathematical sense, numerical computation allows to perform approximate inversion. However, such numerical inversion can lead to numerical instability, such as exploding values, which CBM-IF may face.
> >
> > To counteract this, CBM-IF includes constraints to limit parameter changes, ensuring stability. This constraint effectively mitigates the risk of numerical explosion. In ECBM, we utilize EK-FAC to approximate the Hessian. This method first assumes that parameters across different layers are independent. For each layer, a specially designed regularization term is added to avoid singularity, thus making the entire Hessian matrix invertible.
> >
> > Thus, the improved F1 scores of ECBM over CBM-IF arise from differing methods of addressing Hessian singularity.
> >
> >
> > >Q3:
> > In Figure 5, it looks like neither the ECBM nor the Retrain model generates reasonable explanations. For example, I can not see orange upperparts or legs in either figure. Moreover, the explanations are exactly the same for the two very different figures. Is it a bug or feature?
> >
> > There might be some misunderstanding regarding Figure 5. Each concept is removed through retraining or ECBM. Based on the observed decrease in F1 score, we ranked the concepts and presented the top-10 in Figure 5. We also extract the concept label for each individual image. In Figure 5, '0' or '1' represents the ground truth concept label for the image in the CUB dataset. If visually the concept labels are inconsistent with the corresponding images, this is not an issue with ECBM or retraining.
> >
> > >Q4:
> > As shown in equations (1) and (2), the authors seem to focus on CBM models where the concept predictor and the label predator are trained separately. What about CBM models where these two predictors are trained jointly?
> >
> > Although our ECBM is based on a theoretical foundation for ensuring the reliability of model editing in sequential CBMs, ECBM can still be employed to perform model editing for jointly trained CBMs.
> >
> > This is because, although the concept predictor and the label predictor are jointly trained, their prediction processes are conducted sequentially. Therefore, we can use ECBM to edit them separately.
> >
> > >Q5:
> > In equation (1) and (2), what is L_{C_j} and L_{Y_i}?
> >
> > In (1), $L_{C}(g^j(x_i),c_i^j)$ represents the loss function for the $j$-th concept of the $i$-th data point, while $L_{C_j}(g(x_i),c_i)$ provides a simplified notation of the same concept. Both notations convey the same idea of measuring loss for concept prediction.
> >
> >
> > In (2), $L_{Y}(f(\hat{g}(x_i)), y_i)$ represents the loss function of the $i$-th data in the training set, where $f(\hat{g}(x_i)$ denotes the predicted label, and $y_i$ the true label. And $L_{Y_i}(f, \hat{g})$ is a simplified notation, omitting $x_i$ and $y_i$ for brevity as the context makes their roles clear.
> >
> > >Q6:
> > In Lines 65-67, what is the difference between l and capital L?
> >
> > We carefully reviewed Lines 65-67 but were unable to locate either 'l' or 'L' in the specified lines.

---

> > ### Comment · Reviewer_KdDJ · 2024-11-24
> >
> > Thank you for your reply. Most if my concerns about the experimental setting have been resolved, and I have updated my score to 6.
> >
> > I do, however, have some additional questions and suggestions:
> >
> > - In Figure 5, are the top 10 important concepts shown for the entire model rather than specific classes or individual samples? The figure is somewhat confusing because it appears that the listed concepts are relevant to the specific image displayed on the left. The authors might consider clarifying this in the caption or revising the figure to avoid potential misinterpretation.
> >
> > - The use of undefined notations or conventions impacts the paper's readability to some extent. For instance, $L_C (g^j(x_i), c_i^j)$ and $L_{C_j}(g(x_i), c_i)$ seem to represent the same thing, but this is not clarified when the notation first appears. Readers should not be expected to infer this automatically.

---

> > > ### Author Response · Authors · 2024-11-25
> > >
> > > Dear Reviewer,
> > >
> > > Thank you very much for taking the time to review our responses and reassess our work.
> > >
> > > You are correct in your understanding. The top 10 most influential concepts shown in Figure 5 are identified for the entire model. We have modified the caption and included a brief explanation in Appendix H.1 of the revision.\
> > > Furthermore, to enhance readability, we have included a notation table in Appendix A of the  revision.
> > >
> > > Thank you once again for your invaluable suggestions to improve our paper!
> > >
> > > We hope these revisions address your concerns and contribute to a favorable reassessment of our work.
> > >
> > > Best regards,\
> > > Authors

---

> > > ### Author Response · Authors · 2024-11-26
> > >
> > > Dear Reviewer KdDJ,
> > >
> > > Thank you again for your detailed and constructive review comments. We have revised our paper to incorporate the additional questions and suggestions you raised. We would greatly appreciate it if you could review these changes to ensure they address your concerns. We look forward to receiving your further feedback on our article.
> > >
> > > Best,\
> > > Authors

---

> > > ### Author Response · Authors · 2024-12-02
> > >
> > > Dear Reviewer KdDJ,
> > >
> > > I hope this message finds you well. I am writing to follow up, as today is the final day to provide feedback on our rebuttal discussions. Your insights are invaluable in addressing any remaining concerns and refining our submission to the best possible version.
> > >
> > > We have included detailed explanations for Figure 5 and added a notation table to directly address your feedback. We would greatly appreciate your review and feedback on the updated explanations and the notation table. If any aspects of the rebuttal require further clarification, please let us know, and we will respond promptly. If your concerns have been addressed, we kindly ask that you consider this when revising your review score.
> > >
> > > Thank you for your time and effort during this review process. We sincerely appreciate your valuable feedback and look forward to hearing your thoughts.
> > >
> > > Best,\
> > > Authors

---

> ### Author Response · Authors · 2024-11-29
>
> Dear Reviewer KdDJ,
>
> Thank you again for your thoughtful participation and feedback in this discussion. We have carefully addressed your primary concern as well as the additional points you raised. We kindly request your review of these changes to confirm that they effectively address your concerns. We would also appreciate it if you could reflect this in your evaluation if your concerns have been addressed. If there are any remaining concerns preventing this, we would appreciate further clarification and are happy to continue addressing them.
>
> Best,\
> Authors

---

> ### Author Response · Authors · 2024-11-30
>
> Dear Reviewer KdDJ,
>
> We hope you had a wonderful Thanksgiving. Once again, we sincerely thank you for your valuable time and effort, and we fully understand your busy schedule. We noticed that we haven’t received feedback from you following our latest revision updates in which we added explanations for figure 5 and notations. In the meantime, we have addressed the concerns of Reviewer oeij, who has expressed a positive attitude and increased the score.
>
> We would be truly grateful if you could let us know whether there are any additional questions or concerns we can address before the rebuttal deadline. Your feedback is invaluable and plays a crucial role in strengthening our work.
>
> Thank you once again for your contribution to ICLR! Wishing you all the best with your current and future submissions.
>
> Sincerely,\
> Authors

---

> ### Author Response · Authors · 2024-12-01
>
> Dear Reviewer KdDJ,
>
> Thank you for your valuable contributions during this review process. We wish to kindly remind you that the extended discussion period will end in a few days.
>
> Since we have addressed your remaining concerns about Figure 5 and notations above and included additional explanations in the appendix, we would be happy if you could read them and update your score or leave additional comments. We trust that you will re-evaluate our paper based on our additional clarifications.
>
> Finally, we deeply appreciate your participation in this thorough discussion and look forward to hearing your final decision.
>
> Best,\
> Authors

---

### Official Review · Reviewer_nBLD · 2024-11-11

**Soundness:** 3
**Presentation:** 2
**Contribution:** 3
**Rating:** 5
**Confidence:** 3

**Summary:**

The paper proposes editable concept bottleneck models (ECBMs), that allow a CBM model to be updated efficiently without a full retraining. They consider three different types of editing: concept-level, data-level, and concept-label level, which cover practical scenarios such as data mis-labeling, spurious concepts, and concept annotation errors.

Using the idea of influence functions, they propose closed-form approximations to update the CBM parameters (both concept predictor and label predictor) under the three types of editing scenarios. They utilize the EK-FAC method to accelerate the computation of Hessian inverse and non-convex loss functions.

**Strengths:**

Being able to edit or adapt a CBM model under the scenarios described in the paper is an important practical problem, with applications in domains such as medical imaging. I like how the paper formally addresses the three scenarios of concept-level, data-level, and concept label-level editing in a theoretical way.

Addressing data removal issues in a CBM, due to mis-labeling, poisoning, or privacy issues and the adoption of influence functions for this is a novel contribution. Using ECBMs to effectively edit the model’s knowledge about specific samples, and the connection to membership inference attacks is interesting.

**Weaknesses:**

1. Lots of issues with inconsistent and non-intuitive notation which makes it hard to follow the theory. I think it needs to be proof-read carefully and if possible use simpler notations. The loss functions should be defined so that the method can be understood better (are the loss functions convex?). In multiple places, it is not clear how the gradient and Hessian are defined with respect to a function, e.g., is $H_{\hat{g}}$ (last line of page 4) the Hessian of the function w.r.t the parameters of $\hat{g}$ ?

1. The theorems need some explanations to convey the implications. The paper is currently like theorem after theorem and it may be hard for readers to see the connection. In general, the presentation of ideas could be improved.

1. In the experiments, it is not clear what is the CBM architecture and the number of model parameters? How well does the proposed method scale to a large number of parameters, given that Hessian matrices or their approximations are involved?

1. Limitations of the proposed work are not addressed.

**Questions:**

1. How frequently are such editing updates expected to be made in a practical setting? After training the CBM, one may find some concept labeling errors and/or the need to remove some training samples or concepts. So applying the ECBM method once after training makes sense, but are there scenarios where the editing is done periodically? In situations where the distribution of the test data can change over time, it makes sense to adapt the CBM efficiently, instead of retraining.

1. Referring to Figure 2, the F1 score decreases as more edits are made to the CBM. This could be because the experiments consider a contrived scenario where a random subset of concepts or data points are removed. In reality, if there are concept/data labeling errors or some spurious concepts, then I think the performance of the (edited) CBM is expected to improve?

1. In Table 1, all the values corresponding to the ECBM method are bold-faced. It is more useful to highlight the best (and perhaps second-best) performing methods.

1. Lines 455–456: note the errors or typos in the values. On line 517, it says the top 1010 most influential concepts. I think it should be top 10.

---

> ### Author Response · Authors · 2024-11-23
> **Response to Reviewer nBLD**
>
> ### Weakness
> >**W1**:
> ...think it needs to be proof-read carefully and if possible use simpler notations. The loss functions should be defined so that the method can be understood better ...In multiple places, it is not clear how the gradient and Hessian are defined with respect to a function, e.g., is $H_{\hat{g}}$ (last line of page 4) the Hessian of the function w.r.t the parameters of $\hat{g}$ ?
>
> The detailed loss function is shown in lines 106-107. Here, both $L_C$ and $L_Y$ are chosen as cross-entropy loss, which is commonly used in classification tasks.
>
> The loss function may initially be non-convex. while the classical influence function is most straightforwardly applied to convex functions, its principles can be adapted to non-convex functions through regularization techniques. Moreover, the influence function has been successfully applied to modern neural networks including large language models (LLMs) and diffusion models [1-3].
>
> $g$ and $f$ refer to model parameters, rather than functions. $\nabla_{\hat{g}}$ represents the derivative of the loss function with respect to the parameter of the concept predictor, denoted by $g$. Therefore, $\nabla_{\hat{g}}$ shares the same shape as $g$. The Hessian $H_{\hat{g}}$ is defined similarly.
>
>
> [1]. Grosse, R., Bae, J., Anil, C., Elhage, N., Tamkin, A., Tajdini, A., ... & Bowman, S. R. (2023). Studying large language model generalization with influence functions. arXiv preprint arXiv:2308.03296.\
> [2]. Choe, S. K., Ahn, H., Bae, J., Zhao, K., Kang, M., Chung, Y., ... & Xing, E. (2024). What is Your Data Worth to GPT? LLM-Scale Data Valuation with Influence Functions. arXiv preprint arXiv:2405.13954.\
> [3]. Kwon, Y., Wu, E., Wu, K., & Zou, J. (2023). Datainf: Efficiently estimating data influence in lora-tuned llms and diffusion models. arXiv preprint arXiv:2310.00902.
> >**W2**:
> >The theorems need some explanations to convey the implications. The paper is currently like theorem after theorem and it may be hard for readers to see the connection. In general, the presentation of ideas could be improved.
>
> While the paper includes 6 theorems, these theorems are systematically categorized into three distinct parts—data-level, concept-level, and concept-label-level, each representing a specific aspect of our model editing framework. In each part, two theorems are presented, with one focusing on the editing of the concept predictor and the other on the label predictor within that level.
>
> This is also one of the advantages of our paper. We comprehensively consider all potential editing needs that CBM may face in practical applications, modeling them and deriving theorems to provide corresponding methods for editing models. The challenge of this paper lies in the fact that the model editing methods differ significantly at each level, requiring the use of different technical means to derive the model editing theorems. Therefore, although our paper contains many theorems and may appear complex, this is necessary to comprehensively cover the scenarios of CBM editing.
>
> >**W3**:
> >In the experiments, it is not clear what is the CBM architecture and the number of model parameters? How well does the proposed method scale to a large number of parameters, given that Hessian matrices or their approximations are involved?
>
> In the experiments, the CBM architecture is the same as [1]. And for the concept predictor, we use ResNet-18.
>
> Our method has been designed to accommodate the majority of current CBMs, which typically do not involve very large-scale models, such as VLMs with millions of parameters. By utilizing EK-FAC to accelerate, our approach can effectively manage the computational requirements associated with these CBMs.
>
> In [2-4], influence function has been scaled to LLM with billions of parameters, which fully validates the efficiency of EK-FAC in the inverse Hessian-vector product. As a result, CBMs, which usually have millions of parameters, are computationally feasible for EK-FAC.
>
>
> [1]. Koh, P. W., Nguyen, T., Tang, Y. S., Mussmann, S., Pierson, E., Kim, B., & Liang, P. (2020, November). Concept bottleneck models. In International conference on machine learning (pp. 5338-5348). PMLR.\
> [2]. Choe, S. K., Ahn, H., Bae, J., Zhao, K., Kang, M., Chung, Y., ... & Xing, E. (2024). What is Your Data Worth to GPT? LLM-Scale Data Valuation with Influence Functions. arXiv preprint arXiv:2405.13954.\
> [3]. Kwon, Y., Wu, E., Wu, K., & Zou, J. (2023). Datainf: Efficiently estimating data influence in lora-tuned llms and diffusion models. arXiv preprint arXiv:2310.00902.\
> [4]. Grosse, R., Bae, J., Anil, C., Elhage, N., Tamkin, A., Tajdini, A., ... & Bowman, S. R. (2023). Studying large language model generalization with influence functions. arXiv preprint arXiv:2308.03296.
>
> >**W4**:
> >Limitations of the proposed work are not addressed.
>
> Due to the space limitation, this part is included in the Appendix I in the revision.

---

> ### Author Response · Authors · 2024-11-23
> **Response to Questions**
>
> ### Questions
> >**Q1**:
> >How frequently are such editing updates expected to be made in a practical setting? After training the CBM, one may find some concept labeling errors and/or the need to remove some training samples or concepts. So applying the ECBM method once after training makes sense, but are there scenarios where the editing is done periodically? In situations where the distribution of the test data can change over time, it makes sense to adapt the CBM efficiently, instead of retraining.
>
> Our method can perform periodic editing. However, as the number of edits is very large, our method becomes less accurate. Actually, this is unavoidable for all IF methods. However, our method's ability to handle multiple edits, when the edit number is relatively large, we conducted multiple editing experiments.
>
> Firsly, we introduce noises under three levels. In concept level, we choose 10% of the concepts and flip these concept labels for a portion of the data. In data level, we choose 10% of the data and flip their labels. In concept-label level, we choose 10% of the total concepts and flip them. Then we conduct the following experiments.
>
> In the concept level, we firstly remove 1% of the concepts, then retrain or use ECBM to edit and repeat. In the data level, we firstly remove 1% of the data, then retrain or use ECBM to edit. In the concept label level, we firstly remove one concept label from 1% of the data, then retrain or use ECBM to edit. Note that when remove the next 1% of the concepts, **ECBM edit the model based on the last editing result**.
>
> The results are listed in Table A, B and C, where the first column refers to the portion of mislabeled concept/data/concept label left to be removed.
>
> | Concept (%) | Retrain Acc | ECBM Acc | Retrain F1 | ECBM F1 |
> |-------------|-------------|----------|------------|---------|
> | 10%         | 0.7892      | 0.7892   | 0.7780     | 0.7780  |
> | 9%          | 0.7908      | 0.7921   | 0.7815     | 0.7794  |
> | 8%          | 0.7917      | 0.7903   | 0.7797     | 0.7765  |
> | 7%          | 0.7921      | 0.7905   | 0.7814     | 0.7783  |
> | 6%          | 0.7919      | 0.7899   | 0.7833     | 0.7809  |
> | 5%          | 0.7930      | 0.7918   | 0.7844     | 0.7833  |
> | 4%          | 0.7942      | 0.7926   | 0.7836     | 0.7812  |
> | 3%          | 0.7952      | 0.7913   | 0.7859     | 0.7841  |
> | 2%          | 0.7974      | 0.7965   | 0.7843     | 0.7834  |
> | 1%          | 0.7987      | 0.7958   | 0.7898     | 0.7847  |
> | 0%          | 0.7975      | 0.7972   | 0.7893     | 0.7853  |
>
> *Table A: Multiple Editing under Concept Level.*
>
>
> | Data (%)   | Retrain Acc | ECBM Acc | Retrain F1 | ECBM F1 |
> |------------|-------------|----------|------------|---------|
> | 10%        | 0.7733      | 0.7733   | 0.7694     | 0.7694  |
> | 9%         | 0.7740      | 0.7738   | 0.7748     | 0.7704  |
> | 8%         | 0.7748      | 0.7702   | 0.7712     | 0.7675  |
> | 7%         | 0.7763      | 0.7743   | 0.7718     | 0.7693  |
> | 6%         | 0.7747      | 0.7755   | 0.7723     | 0.7685  |
> | 5%         | 0.7783      | 0.7758   | 0.7724     | 0.7681  |
> | 4%         | 0.7808      | 0.7786   | 0.7751     | 0.7713  |
> | 3%         | 0.7862      | 0.7843   | 0.7814     | 0.7765  |
> | 2%         | 0.7923      | 0.7889   | 0.7888     | 0.7843  |
> | 1%         | 0.7955      | 0.7938   | 0.7904     | 0.7881  |
> | 0%         | 0.7975      | 0.7961   | 0.7893     | 0.7887  |
>
> *Table B: Multiple Editing under Data Level.*
>
> | Concept Label(%) | Retrain Acc | ECBM Acc | Retrain F1 | ECBM F1 |
> |-------------------|-------------|----------|------------|---------|
> | 10%              | 0.7893      | 0.7893   | 0.7811     | 0.7811  |
> | 9%               | 0.7888      | 0.7881   | 0.7845     | 0.7796  |
> | 8%               | 0.7913      | 0.7890   | 0.7819     | 0.7805  |
> | 7%               | 0.7911      | 0.7895   | 0.7820     | 0.7794  |
> | 6%               | 0.7935      | 0.7904   | 0.7842     | 0.7814  |
> | 5%               | 0.7947      | 0.7896   | 0.7841     | 0.7803  |
> | 4%               | 0.7930      | 0.7921   | 0.7822     | 0.7791  |
> | 3%               | 0.7955      | 0.7907   | 0.7862     | 0.7834  |
> | 2%               | 0.7963      | 0.7916   | 0.7788     | 0.7793  |
> | 1%               | 0.7979      | 0.7964   | 0.7898     | 0.7843  |
> | 0%               | 0.7975      | 0.7967   | 0.7893     | 0.7850  |
>
>
> *Table C: Multiple Editing under Concept-label Level.*
>
> From the above three levels, we can find that with the mislabeled information removed, the retrained model achieves better performance in both accuracy and F1 score than the initial model. Furthermore, the performance of the ECBM-edited model is similar to that of the retrained model, even **after 10 rounds editing**, which demonstrates the ability of our ECBM method to handle multiple edits.

---

> > ### Author Response · Authors · 2024-11-23
> > **Response to Questions**
> >
> > >**Q2:**
> > >Referring to Figure 2, the F1 score decreases as more edits are made to the CBM. This could be because the experiments consider a contrived scenario where a random subset of concepts or data points are removed. In reality, if there are concept/data labeling errors or some spurious concepts, then I think the performance of the (edited) CBM is expected to improve?
> >
> >
> >
> > To verify your hypothesis, we have conducted additional experiments on datasets with synthetically introduced noisy concepts and datasets with synthetic noisy labels.
> >
> > Follow the setting in response to Q1, we introduce noises into the three levels and train the model. After that, we remove the noise and obtain the retrained mdoel, which is the ground truth(gt) of this harmful data removal task. In contrast, we use ECBM to remove the harmful data. The results are shown in Table D.
> >
> >
> >
> > | Noise Level           | Setting               | Acc | F1 Score |
> > |-----------------------|-----------------------|-----------|----------|
> > | **Concept**           | **Before Removal**    | 0.7892    | 0.7780   |
> > |                       | **Removal by Retrain** | 0.7975    | 0.7893   |
> > |                       | **Removal by ECBM**    | 0.7956    | 0.7839   |
> > | **Data**              | **Before Removal**    | 0.7733    | 0.7694   |
> > |                       | **Removal by Retrain** | 0.7975    | 0.7893   |
> > |                       | **Removal by ECBM**    | 0.7964    | 0.7815   |
> > | **Concept Label**     | **Before Removal**    | 0.7893    | 0.7811   |
> > |                       | **Removal by Retrain** | 0.7975    | 0.7893   |
> > |                       | **Removal by ECBM**    | 0.7983    | 0.7872   |
> >
> >
> > *Table D: Harmful Data Removal on Nosiy CUB Dataset.*
> >
> > From Table D, it can be observed that the model performance improves across all three settings after noise removal and subsequent retraining or ECBM editing.
> >
> >
> > This confirms that the performance of ECBM is nearly equivalent to retraining in various experimental scenarios, further providing evidence of the robustness of our method.
> >
> > >**Q3:**
> > >In Table 1, all the values corresponding to the ECBM method are bold-faced. It is more useful to highlight the best (and perhaps second-best) performing methods.
> >
> > We understand the importance of clearly highlighting the best performing methods for better comparison. However, currently, the ECBM method values are bold-faced because our focus was on demonstrating the effectiveness of our proposed ECBM method, which serves as the primary baseline for model editing without retraining.
> >
> > The "Retrain" method is included as it represents the ground truth in attributing data in model parameters. No method can outperform retrain. However, it is the ideal case and is inefficient when the model size is large.
> >
> > >**Q4**:
> > >Lines 455–456: note the errors or typos in the values. On line 517, it says the top 1010 most influential concepts. I think it should be top 10.
> >
> > We corrected it in our revision.

---

> ### Author Response · Authors · 2024-11-25
>
> Dear Reviewer nBLD,
>
> We are truly grateful for the time and effort you have dedicated to reviewing our work. We would like to respectfully inform you that two reviewers have already increased their scores: Reviewer KdDJ raised the score from 5 to 6, and Reviewer Oeij raised it from 3 to 5.
>
> We would highly appreciate your review and feedback on the responses. Should you have any questions or concerns, we would be happy to discuss and address them.
>
>
> Best,\
> Authors

---

> > ### Author Response · Authors · 2024-11-26
> >
> > Dear Reviewer nBLD,
> >
> > Thank you again for your detailed and constructive review comments. We respectfully remind you that the discussion period will end in a few days. We have responded above to your concerns. We believe these address your concerns. We would appreciate it if you would take the time to read and comment on the responses.
> >
> > Best,\
> > Authors

---

> ### Author Response · Authors · 2024-11-27
>
> Dear Reviewer nBLD,
>
> Thank you again for your detailed and constructive review comments. We respectfully remind you that the discussion period will end in a few days. We have responded above to your concerns. We believe these address your concerns. We would appreciate it if you would take the time to read and comment on the responses.
>
> Best,
> Authors

---

> ### Author Response · Authors · 2024-11-29
>
> Dear Reviewer nBLD,
>
> Thank you again for your constructive review comments. We have addressed your concern in the above. We respectfully request you to consider raising your rating score accordingly if your concerns are alleviated. Otherwise, we would be happy to hear the remaining concerns that prevent you from doing so and continue to discuss them.
>
>
> Best,\
> Authors

---

> ### Author Response · Authors · 2024-11-30
>
> Dear Reviewer nBLD
>
> We hope you had a wonderful Thanksgiving. Once again, we sincerely thank you for your valuable time and effort, and we fully understand your busy schedule. We noticed that we haven’t received any feedback from you yet. In the meantime, we have addressed the concerns of the other two reviewers, who have expressed a positive attitude and increased their scores.
>
> We would be truly grateful if you could let us know whether there are any additional questions or concerns we can address before the rebuttal deadline. Your feedback is invaluable and plays a crucial role in strengthening our work.
>
> Thank you once again for your contribution to ICLR! Wishing you all the best with your current and future submissions.
>
> Sincerely,\
> The Authors

---

> ### Author Response · Authors · 2024-12-01
>
> Dear Reviewer nBLD,
>
> Thank you for your valuable contributions to ICLR. We kindly remind you that the extended discussion period will end in a few days.
>
> Since we have addressed the concerns you raised, we kindly request that you review our clarifications and consider providing additional comments or updating your score. We trust that you, as the reviewer, will re-evaluate our paper based on our clarifications.
>
> Thank you for your valuable contributions, and we kindly await your thoughtful final decision.
>
> Best,\
> Authors

---

> ### Author Response · Authors · 2024-12-02
>
> Dear Reviewer nBLD,
>
> I hope this message finds you well. I am writing to follow up as today marks the final day for providing feedback on our rebuttal discussions. Your insights are invaluable in addressing any remaining concerns and ensuring the best possible version of our submission.
>
> If there are any points in the rebuttal that require further clarification or discussion, I would be more than happy to address them promptly. I truly appreciate the time and effort you dedicate to this review process. Otherwise, we respectfully request you to consider raising your rating score accordingly if your concerns are alleviated.
>
> Thank you very much for your consideration, and I look forward to your feedback.
>
> Best regards,\
> Authors

---

> ### Comment · Reviewer_nBLD · 2024-12-03
> **Response to author rebuttal**
>
> Thank you for your responses and for running additional experiments to address the reviews. I apologize for responding late in the process.
>
> Most of my questions have been addressed, but I find that the revised paper is still lacking in clarity and I think a significant rewrite will help address this. Similar concerns have been raised by Reviewer `oeij` as well. Therefore, I will maintain my score of 5.
>
> I also think that the contributions of the paper are valuable and it addresses and important problem. Therefore, I am not opposed to acceptance, if supported by the other reviewers.

---

> > ### Author Response · Authors · 2024-12-03
> >
> > Dear Reviewer nBLD,
> >
> > Thanks for your response!
> >
> > The concerns about writing from Reviewer oeij are mainly about grammar and writing style. For the grammatical issues, we have checked each of them in detail using both Grammarly and ChatGPT, and we did not identify any significant errors. With regard to the writing style, we believe that our work adheres to standard practices in scientific writing.
> >
> > The other three reviewers rated us as 6 (marginally above the acceptance threshold), while we received a 5 (marginally below the acceptance threshold) from Reviewer oeij. The only concern raised by Reviewer oeij and yourself pertains to the writing style. All the reviewers acknowledge our contribution. If you find no strong objections to our work's acceptance, we would sincerely appreciate your consideration in raising the score from 5 to 6.
> >
> > Thank you for your contribution to ICLR 2025. Wishing you a wonderful day.
> >
> > Best regards,\
> > Authors

---

> > ### Comment · Reviewer_nBLD · 2024-12-03
> >
> > Below are a few issues that I would like to point out.
> >
> > Appendix J, which was added during the revision, does not really discuss the limitations of the work.
> >
> > > The detailed loss function is shown in lines 106-107. Here, both $L_C$ and $L_Y$ are chosen as cross-entropy loss, which is commonly used in classification tasks.
> >
> > The loss functions are not defined here, and it is not mentioned that they are chosen to be the cross-entropy loss anywhere in the paper.
> >
> > >  $g$ and $f$ refer to model parameters, rather than functions. $\nabla_{\hat{g}}$ represents the derivative of the loss function with respect to the parameter of the concept predictor, denoted by $g$.
> >
> > This is an example of the hand-wavy math in the paper, that unfortunately makes it hard to follow. On lines 144 and 145 in the paper, you have defined these as functions $g : R^m \mapsto R^k$ and $f : R^k \mapsto R^{d_z}$. However, here you state they are parameters, and take gradient and Hessian w.r.t. them.
> >
> > On Line 160, the predicted concept vector should be $\hat{c} = \hat{g}(x)$.
> >
> > While these are admittedly minor issues, addressing them could improve the quality and readability of the paper.

---

> > > ### Author Response · Authors · 2024-12-03
> > >
> > > Dear Reviewer nBLD,
> > >
> > > Thanks for your feedback.
> > >
> > > Sorry for misleading. The loss function is defined in lines 148-155.
> > > For other minor issues, we promise we will fix these in the camera-ready version.
> > >
> > >
> > > Best,\
> > > Authors

---

> > > ### Comment · Reviewer_nBLD · 2024-12-03
> > >
> > > > The concerns about writing from Reviewer oeij are mainly about grammar and writing style.
> > >
> > > I respectfully disagree about the concerns being mainly about grammar and writing style. There are several comments on mathematical issues and errors. I had similar observations, but did not carefully list them out.

---

> > > > ### Author Response · Authors · 2024-12-04
> > > >
> > > > Dear Reviewer nBLD,
> > > >
> > > > Thanks for your comment.
> > > >
> > > > >The loss functions are not defined here, and it is not mentioned that they are chosen to be the cross-entropy loss anywhere in the paper.
> > > > >>The detailed loss function is shown in lines 106-107. Here, both $L_C$ and $L_Y$ are chosen as cross-entropy loss, which is commonly used in classification tasks.
> > > >
> > > > We apologize for our typo in the response to your first weakness. The detailed loss function is defined in lines 148-155.
> > > > And we stated that we employ the same CBM setting as [1] in lines in 375-376, which includes the cross-entropy loss setting.
> > > >
> > > >
> > > > >This is an example of the hand-wavy math in the paper, that unfortunately makes it hard to follow. On lines 144 and 145 in the paper, you have defined these as functions $g : R^m \mapsto R^k$ and $f : R^k \mapsto R^{d_z}$. However, here you state they are parameters, and take gradient and Hessian w.r.t. them.
> > > > >>$g$ and $f$ refer to model parameters, rather than functions. $\nabla_{\hat{g}}$ represents the derivative of the loss function with respect to the parameter of the concept predictor, denoted by $g$.
> > > >
> > > > $g$ and $f$ are model parameters as well as a function. Let's take a matrix $A\in R^{m\times n}$ as an example. $A$ can serve as the model parameter in a linear model. $A$ can also be regarded as a linear transformation which is a function mapping a vector $x\in R^{n\times 1}$ to a vector $y\in R^{m\times 1}$ via matrix multiplication. This dual interpretation is a fundamental concept in many areas of applied mathematics, and has been used in many papers.
> > > >
> > > > >On Line 160, the predicted concept vector should be $\hat{c} = \hat{g}(x)$.
> > > >
> > > > Thanks for your correction. We will fix it in the camera-ready version.
> > > >
> > > > Unfortunately, we could not check the mathematical issues you mentioned due to insufficient details being provided. However, we have already resolved the mathematical issues identified in oeij's previous response, which were related to typographical errors and resolved through minor adjustments.
> > > >
> > > > Best regards,\
> > > > Authors
> > > >
> > > >
> > > > [1]. Pang Wei Koh, Thao Nguyen, Yew Siang Tang, Stephen Mussmann, Emma Pierson, Been Kim, and Percy Liang. Concept bottleneck models. In International conference on machine learning, pp. 5338–5348. PMLR, 2020.

---

### Author Response · Authors · 2024-11-24
**General Response (Revision Updates)**

Dear Reviewers and ACs,

Thank you for providing your detailed, insightful, and helpful review comments. We addressed your concerns in our responses to each reviewer and revised the paper based on your suggestions. We greatly appreciate how the reviewers' comments have strengthened our paper. Below, we summarize the key revisions made.

1. We add harmful removal experiments at the concept, data, and concept label levels in 'H.2 Improvement via Harmful Data Removal' in the appendix of the revision. In this experiment, retraining and ECBM both demonstrate improved model performance, further highlighting the effectiveness of ECBM.
2. We add experiment to evaluate the multiple editing ability of ECBM in the appendix 'H.3 Periodic Editing Performance'.
3. We expand the discussion on related work concerning machine unlearning in 'I. More Related Work' in the appendix.
4. We include a discussion of the limitations and broader impacts in 'J. Limitations and Broader Impacts' in the appendix.
5. We add explanation for Figure 5 in the appendix 'H 4.1. Explanation for Visualization Results'.
6. We revise the description of retraining, changing it from 'baseline' to 'ground truth,' in the Baselines and Evaluation Metric section.
7. We add notation table in the appendix A.

We noticed that reviewers may have some misunderstandings when comparing retrain and ECBM. For our model editing task, retrain serves as a ground truth rather than a baseline. Our proposed ECBM method is an efficient and accurate approximation of this ground truth. As shown in Table 1, the estimation results of ECBM are very similar to those of the retrain method, while the runtime is significantly reduced. Therefore, to avoid misinterpretation, we update the description of retraining from 'baseline' to 'ground truth' in the Baselines and Evaluation Metric part.

We sincerely hope that the responses and the revised paper assist in your re-evaluation of our paper.

---

### Author Response · Authors · 2024-12-03
**General Response**

Dear Reviewers and Area Chair,

Thank you for your valuable feedback and thoughtful comments. We addressed all the concerns raised in the reviews during the rebuttal process.

For issues related to grammar and clarity, we have ensured that these concerns have been fully resolved through thorough revisions. The only remaining point pertains to differences in writing style. We have followed established conventions in scientific writing and have incorporated feedback from reviewers to improve the overall quality and precision of the revision.

Thank you again for your time and effort in reviewing our work.

Best regards,\
Authors

---

### Meta-Review · Area_Chair_LH7X · 2024-12-21

**Metareview:**

This work studies an approach, based on approximate influence functions, to edit Concept Bottleneck Models (CBM) to accommodate for three levels of interventions: modifying a (potentially erroneous) concept annotated for a given sample, removing a concept from the model altogether, or removing the influence of a training sample. The approach is empirically effective while circumventing the need for retraining the CMB from scratch.

**Strengths**
* Innovative approach, and nice trick, of influence functions to edit CMBs.
* Experiments support the claims in the paper.
* The three levels of edits are interesting.

**Weaknesses**
* The annotation often is unclear, making it more difficult to understand the methodology fully.
* Lack of clarity in writing and typos.
* Despite the stress on the rigor of the (approximate) mathematical formulations, no guarantees on the level of approximations are given, nor evaluated empirically. Additional assumptions - like the independence of the network layers - are unrealistic and only mentioned in passing.

**Summary**

This paper received profuse discussions among reviewers and authors, resulting in a more comprehensive picture of the paper from the former and an improvement of the quality of the manuscript by the latter. All reviewers agree on the relevant nature of the problem, and on the innovative solution. At the same time, several reviewers noted that the quality of the presentation is far from where they would like it to be. After reviewing all of the comments, questions, and responses from the authors, I turned to review the paper myself. Unfortunately, I concur with the reviewers nBLD, KdDJ, and oeij, that the presentation of the content is still confusing and unclear. I won't list all of my concerns here, but a non-exhaustive list includes:
* line 158: ".. different columns as a vector $g^j(x_i)$", whereas notation indicates that this should be a scalar.
* line 167: loss L is undefined;
* The notation $m$ is used to denote the index of a sample as well as the dimension of the data,
* line 207: The sentence beginning "Observing.." doesn't have a subject nor predicate.

Moreover, while verifying the modifications done by the authors, I noted that they added a Limitations and Broader Impacts section, as suggested by nBLD. However, this section only has 2 simple sentences on the limitations of the approximation used; the rest of it is simply a repeated verbatim copy of the abstract. This is, naturally, not acceptable.

As a result, given the strengths and weaknesses above, as well as revising the discussions and revision, I cannot recommend acceptance. I encourage the authors to continue improving on this work to provide the best presentation for their innovative and nice ideas.

**Additional Comments On Reviewer Discussion:**

The discussion with the reviewers was extensive and productive.

**Rev nBLD** had concerns mainly on the lack of clarity of presentation, comments on the implications of the theorems, clarifications on the implementation of the model, and limitations. Most of these were resolved, but not all of them to satisfaction (see above).

**Rev KdDj** had questions on related works and details on the experimental settings and results. They also referred to the lack of clarity of presentation. The former points were all resolved by the authors.

**Rev oeij** had many questions and suggestions on both writing and typos aspects, as well as very precise technical observations, some of them including finding inconsistencies in the version of the dataset used by the authors. The authors addressed most of their comments, improving their work, but the reviewer remained dissatisfied with the final state.

**Rev sqUB** was happy about the contribution and mathematical notation, and had questions on the scalability of the approach - which were addressed by the authors. The reviewer also mentioned room for improvement in presentation.

**rev Ak63** was happy about the notation and mentioned the paper was easy to follow, with insightful results. Had questions on the applicability of influence functions for non-convex settings and questions on notation for the '='. These were resolved.

---

### Decision · Program_Chairs · 2025-01-22

Reject